# Dimension-Independent Convergence of Underdamped Langevin Monte Carlo in KL Divergence

**Shiyuan Zhang**[* 1]  **Qiwei Di**[* 1]  **Xuheng Li**[* 1]  **Quanquan Gu**[1]

## Abstract

Underdamped Langevin dynamics (ULD) is a widely-used sampler for Gibbs distributions $\pi \propto e^{-V}$, and is often empirically effective in high dimensions. However, existing non-asymptotic convergence guarantees for discretized ULD typically scale polynomially with the ambient dimension $d$, leading to vacuous bounds when $d$ is large. The main known dimension-free result concerns the randomized midpoint discretization in Wasserstein-2 distance (Liu et al., 2023), while dimension-independent guarantees for ULD discretizations in KL divergence have remained open. We close this gap by proving the first dimension-free KL divergence bounds for discretized ULD. Our analysis refines the KL local error framework (Altschuler et al., 2025) to a dimension-free setting and yields bounds that depend on $\mathrm{tr}(\mathbf{H})$, where $\mathbf{H}$ upper bounds the Hessian of $V$, rather than on $d$. As a consequence, we obtain improved iteration complexity for underdamped Langevin Monte Carlo relative to overdamped Langevin methods in regimes where $\mathrm{tr}(\mathbf{H}) \ll d$.

## 1. Introduction

Sampling from high-dimensional Gibbs distributions $\pi(\mathbf{x}) \propto \exp(-V(\mathbf{x}))$ is a core primitive in modern machine learning, underpinning Bayesian inference (Robert et al., 1999), diffusion-based generative modeling (Ho et al., 2020; Song et al., 2020), and exploration in reinforcement learning (Thompson, 1933; Zhang et al., 2020). Among practical samplers for smooth log-concave targets, Langevin-based Markov chain Monte Carlo methods are especially appealing: they require only first-order information ($\nabla V$), are simple to implement, and come with a rich body of non-

---
[*]Equal contribution  [1]Department of Computer Science, University of California, Los Angeles, California 90095, USA. Correspondence to: Quanquan Gu <qgu@cs.ucla.edu>.

*Proceedings of the 43$^{rd}$ International Conference on Machine Learning*, Seoul, South Korea. PMLR 306, 2026. Copyright 2026 by the author(s).

asymptotic convergence theory.

The classical *overdamped* Langevin diffusion (OLD)

$$\mathrm{d}\boldsymbol{X}_t^{\mathrm{OLD}} = -\nabla V(\boldsymbol{X}_t^{\mathrm{OLD}})\,\mathrm{d}t + \sqrt{2}\,\mathrm{d}\boldsymbol{B}_t \qquad (1.1)$$

converges to $\pi$ under mild conditions, and its Euler–Maruyama discretization yields the Langevin Monte Carlo (LMC) algorithm. Motivated by Hamiltonian dynamics, the *underdamped* Langevin diffusion (ULD) augments the state with a momentum variable and evolves on phase space:

$$\mathrm{d}\boldsymbol{X}_t = \boldsymbol{P}_t\,\mathrm{d}t,$$
$$\mathrm{d}\boldsymbol{P}_t = -\gamma\boldsymbol{P}_t\,\mathrm{d}t - \nabla V(\boldsymbol{X}_t)\,\mathrm{d}t + \sqrt{2\gamma}\,\mathrm{d}\boldsymbol{B}_t, \qquad (1.2)$$

where $\gamma > 0$ is the friction parameter. ULD has invariant distribution $\pi(\mathbf{x}, \mathbf{p}) \propto \exp\left(-V(\mathbf{x}) - \|\mathbf{p}\|^2/2\right)$, so its $\mathbf{x}$-marginal is the target $\pi(\mathbf{x})$. Discretizations of ULD (collectively, ULMC) are empirically competitive and can provably improve iteration complexity over overdamped methods in several regimes (see Table 1).

A key limitation of existing non-asymptotic theory is that many convergence bounds for Langevin discretizations scale polynomially in the ambient dimension $d$. Such dimension dependence can be pessimistic in high-dimensional applications where the geometry of $V$ is effectively low-dimensional (e.g., ridge-separable (Liu et al., 2023)). Recent work has shown that, in some cases, the relevant complexity is governed by spectral quantities of a Hessian upper bound $\mathbf{H} \succeq \nabla^2 V$, such as $\mathrm{tr}(\mathbf{H})$, leading to dimension-free guarantees for specific overdamped and Wasserstein-based underdamped schemes (Freund et al., 2022; Liu et al., 2023). However, *dimension-independent guarantees for discretized underdamped Langevin in KL divergence* have remained open. Notably, in the strongly log-concave setting, KL convergence is strictly stronger than convergence in Wasserstein distance or total variation, since it implies Wasserstein convergence via Talagrand's $T_2$ inequality (see, e.g., Section 1.4 in Chewi (2025)) and total variation convergence via Pinsker's inequality.

This paper resolves the above question by establishing the first *dimension-free* KL convergence rates for ULMC discretizations. Concretely, we show that both standard ULMC and the randomized midpoint discretization (RMD) admit

*Table 1.* We summarize the sample complexity results from the most important prior works for various discretization methods applied to overdamped Langevin dynamics (OLD) and underdamped Langevin dynamics (ULD). The comparisons include the underlying dynamics (OLD or ULD), the discretization scheme, the metric used to measure convergence, the problem setting, and whether the bound is dimension-free. Throughout the table, LMC denotes Langevin Monte Carlo, i.e., the Euler–Maruyama discretization of OLD, while ULMC denotes the corresponding discretization of ULD. RMD refers to variants of the randomized midpoint method introduced in Shen & Lee (2019). PLMC denotes the Poisson midpoint method proposed in Srinivasan & Nagaraj (2025), which achieves improved convergence rates in Wasserstein distance. For Composite, Freund et al. (2022) assume that the potential function $V$ admits a decomposition and interpret overdamped Langevin dynamics as a composite optimization problem. Regarding the parameters, in the $\alpha$-strongly convex setting, let $\kappa = \beta/\alpha$ be the condition number, $d$ be the ambient dimension, and $\mathbf{H}$ be an upper bound of the Hessian matrix $\nabla^2 V$. For uniformity of presentation, we replace any explicit dependence on $\alpha$ in the bounds by $\beta/\kappa$. In the general convex setting, let $W$ denote the Wasserstein distance between the initial distribution and the target distribution. In all results, we hide the logarithmic factors and omit the $\widetilde{O}$ notation for simplicity.

| Dynamics | Discretization | Metric | Strongly convex | General convex | Dim.-free? | Reference |
|---|---|---|---|---|---|---|
| **OLD** | LMC | $W_2$ | $\mathcal{O}(\kappa^3\beta^{-1}d/\epsilon^2)$ | – | ✗ | Dalalyan (2017a) |
| | LMC | KL | $\mathcal{O}(\kappa^2 d/\epsilon^2)$ | $\mathcal{O}(\beta^2 dW^4/\epsilon^6)$ | ✗ | Cheng & Bartlett (2018) |
| | LMC | KL | $\mathcal{O}(\kappa^2\beta^{-1}d/\epsilon^2)$ | $\mathcal{O}(\beta dW^2/\epsilon^4)$ | ✗ | Durmus et al. (2019) |
| | LMC | KL | $\mathcal{O}(\kappa^2 d/\epsilon^2)$ | $\mathcal{O}(\beta^2 dW^4/\epsilon^6)$ | ✗ | Altschuler & Chewi (2024b) |
| | RMD | KL | $\mathcal{O}(\kappa d^{1/2}/\epsilon)$ | $\mathcal{O}(\beta^{4/3}d^{1/3}W^{8/3}/\epsilon^{10/3})$ | ✗ | Altschuler & Chewi (2024b) |
| | PLMC | $W_2$ | $\mathcal{O}(\kappa^{4/3}\beta^{-1/3}d^{2/3}/\epsilon^{2/3})$ | – | ✗ | Srinivasan & Nagaraj (2025) |
| | Composite | KL | $\mathcal{O}(\kappa^2\beta^{-1}\operatorname{tr}(\mathbf{H})/\epsilon^2)$ | – | ✓ | Freund et al. (2022) |
| **ULD** | ULMC | $W_2$ | $\mathcal{O}(\kappa^{5/2}\beta^{-1/2}d^{1/2}/\epsilon)$ | – | ✗ | Cheng et al. (2018) |
| | ULMC | $W_2$ | $\mathcal{O}(\kappa^2\beta^{-1/2}d^{1/2}/\epsilon)$ | – | ✗ | Dalalyan & Riou-Durand (2018) |
| | RMD | $W_2$ | $\mathcal{O}(\kappa^{4/3}\beta^{-1/3}d^{1/3}/\epsilon^{2/3})$ | – | ✗ | Shen & Lee (2019) |
| | ULMC | KL | $\mathcal{O}(\kappa^{3/2}d^{1/2}/\epsilon)$ | $\mathcal{O}(\beta^{3/2}d^{1/2}W^3/\epsilon^4)$ | ✗ | Altschuler et al. (2025) |
| | RMD | KL | $\mathcal{O}(\kappa d^{1/3}/\epsilon^{2/3})$ | $\mathcal{O}(\beta^{5/4}d^{1/4}W^{5/2}/\epsilon^3)$ | ✗ | Altschuler et al. (2025) |
| | PLMC | $W_2$ | $\mathcal{O}(\kappa^{4/3}\beta^{-1/6}d^{1/3}/\epsilon^{1/3})$ | – | ✗ | Srinivasan & Nagaraj (2025) |
| | RMD | $W_2$ | $\mathcal{O}(\kappa^{5/3}\beta^{-2/3}[\operatorname{tr}(\mathbf{H})]^{1/3}/\epsilon^{2/3})$ | – | ✓ | Liu et al. (2023) |
| | ULMC | KL | $\mathcal{O}(\kappa^{3/2}\beta^{-1/2}[\operatorname{tr}(\mathbf{H})]^{1/2}/\epsilon)$ | $\mathcal{O}(\beta\operatorname{tr}(\mathbf{H})^{1/2}W^3/\epsilon^4)$ | ✓ | **Ours** |
| | RMD | KL | $\mathcal{O}(\kappa\beta^{-1/3}[\operatorname{tr}(\mathbf{H})]^{1/3}/\epsilon^{2/3})$ | $\mathcal{O}(\beta\operatorname{tr}(\mathbf{H})^{1/4}W^{5/2}/\epsilon^3)$ | ✓ | **Ours** |

KL iteration complexities depending on $\operatorname{tr}(\mathbf{H})$ rather than $d$. Our main contributions are:

- In the $\alpha$-strongly convex and $\beta$-smooth setting, we establish non-asymptotic convergence bounds in KL divergence for both standard ULMC and the randomized midpoint discretization. The resulting iteration complexity depends on $\operatorname{tr}(\mathbf{H})$ rather than explicitly on the ambient dimension $d$. In the strongly convex setting, our KL guarantee further implies convergence in Wasserstein distance via Talagrand's inequality, and we show that the resulting rate enjoys a strictly better dependence on the condition number $\kappa$ than that of Liu et al. (2023).

- In the general convex setting ($\alpha = 0$), prior work does not provide any *dimension-free* convergence rates for Langevin dynamics. In this work, we establish the first dimension-free KL convergence guarantees for ULMC and the randomized midpoint discretization (RMD), with complexity governed by $\operatorname{tr}(\mathbf{H})$ instead of $d$. Moreover, the dimension-free convergence rate of RMD is $\mathcal{O}(1/\epsilon^3)$, matching the state-of-the-art rate (Altschuler et al., 2025) in this setting.

- Technically, our improvement stems from two ideas (i) bounding the strong and weak local errors in a manner compatible with $\mathbf{H}$-weighted norms and (ii) controlling the change-of-measure terms without introducing explicit dimension dependence via crude Gaussian moment bounds. These two ingredients allow us to close a strictly tighter error recursion and ultimately yield the first dimension-free KL guarantees for underdamped Langevin discretizations.

**Notations.** We use lower-case boldface letters such as $\mathbf{x}, \mathbf{y}, \mathbf{z}$ to denote vectors, and upper-case boldface italic letters such as $\boldsymbol{X}, \boldsymbol{Y}, \boldsymbol{Z}$ to denote random vectors. We use upper-case boldface letters such as $\mathbf{X}, \mathbf{Y}, \mathbf{Z}$ to denote matrices. For any two positive semi-definite (PSD) matrices $\mathbf{X}, \mathbf{Y}$, we use $\mathbf{X} \preceq \mathbf{Y}$ ($\mathbf{X} \succeq \mathbf{Y}$) to indicate $\mathbf{Y} - \mathbf{X}$ ($\mathbf{X} - \mathbf{Y}$) is positive semi-definite, respectively. We use $\|\cdot\|$ to denote the standard Euclidean 2-norm, and $\|\cdot\|_{L_2}$ to denote the $L_2$ norm of a random vector, i.e., $\|\boldsymbol{X}\|_{L_2} = \sqrt{\mathbb{E}[\|\boldsymbol{X}\|^2]}$. For a PSD matrix $\mathbf{M} \in \mathbb{R}^{d\times d}$ and a vector $\mathbf{x} \in \mathbb{R}^d$, we denote $\|\mathbf{x}\|_{\mathbf{M}} = \sqrt{\mathbf{x}^\top \mathbf{M}\mathbf{x}}$. For any vector $\mathbf{x}$, we use $\delta_{\mathbf{x}}$ to denote the Dirac distribution at $\mathbf{x}$. For any random process with initial distribution $\mu$ and transition kernel $\mathcal{P}$, let $\mu\mathcal{P}$ be the distribution of the random process after applying the transition kernel. We use standard asymptotic notations $\mathcal{O}(\cdot)$ and $\Theta(\cdot)$, and use $\widetilde{\mathcal{O}}(\cdot)$ and $\widetilde{\Theta}(\cdot)$ to hide logarithmic factors. We use $a \lesssim b$, $a \simeq b$ and $a \gtrsim b$ to denote $a = \mathcal{O}(b)$, $a = \Theta(b)$ and $a = \Omega(b)$, respectively. For $a, b \in \mathbb{R}$, we use $a \wedge b$ for $\min\{a, b\}$ and $a \vee b$ for $\max\{a, b\}$.

## 2. Related Work

**Langevin Monte Carlo.** The idea of using Langevin dynamics for sampling from a target distribution dates back to Parisi (1981). More recently, Dalalyan (2017b) provided the first non-asymptotic convergence guarantees for approximating target distributions with log-concave and smooth densities via Langevin Monte Carlo. Since then, a large body of work has focused on developing faster sampling algorithms and establishing sharper convergence rates, covering a wide range of settings, including the strongly log-concave case (Durmus & Moulines, 2019; Dalalyan, 2017a; DURMUS & MOULINES, 2017; Li et al., 2021), the weakly log-concave case (Cheng & Bartlett, 2018; Mangoubi & Vishnoi, 2019), and certain non-log-concave distributions that satisfy some isoperimetric conditions (Ma et al., 2019b; Lee et al., 2018; Xu et al., 2018; Zou et al., 2019; 2021). There is also a line of work that studies convergence in KL or Rényi divergence under functional inequality assumptions (Raginsky et al., 2017; Erdogdu & Hosseinzadeh, 2021; Vempala & Wibisono, 2019; Ganesh & Talwar, 2020; Erdogdu et al., 2022; Mou et al., 2022; Chewi et al., 2025). Other related work includes studies of stochastic gradient Langevin dynamics (SGLD) (Zhang et al., 2017; Gao et al., 2022; Chen et al., 2020a; Deng et al., 2020), the Metropolis-adjusted Langevin algorithm (MALA) (Roberts & Tweedie, 1996; Dwivedi et al., 2019; Bou-Rabee & Hairer, 2013), and the Hamiltonian Monte Carlo (HMC) method (Neal et al., 2011; Durmus et al., 2017; Mangoubi & Vishnoi, 2018; 2019; Bou-Rabee et al., 2020; Chen et al., 2020b).

**Analysis of Underdamped Langevin.** Due to its faster convergence compared with overdamped Langevin, underdamped Langevin diffusion arouses huge research interest (Hérau & Nier, 2004; Villani, 2009; Eberle et al., 2019; Gorham et al., 2019; Baudoin, 2016; Bolley et al., 2010; Calogero, 2012; Dolbeault et al., 2015; Mischler & Mouhot, 2016; Bernard et al., 2022). Its discretization can be viewed as a form of Hamiltonian Monte Carlo, and explicit convergence rates for sampling from smooth and strongly log-concave distributions were first established in Cheng et al. (2018). Subsequently, Ma et al. (2019a) established a connection between underdamped Langevin dynamics and Nesterov's acceleration and proved non-asymptotic convergence guarantees for the underdamped Langevin discretization in KL divergence. Furthermore, Zhang et al. (2023) studied discretization errors in Rényi divergence. Strasman et al. (2025) studied Wasserstein convergence of critically damped Langevin dynamics, while Conforti et al. (2025) investigated convergence in KL divergence. Of particular relevance to our work, Shen & Lee (2019) proposed a randomized midpoint method and proved faster convergence rates in Wasserstein distance. In addition, a number of recent works have focused on developing alternative discretization schemes for underdamped Langevin dynamics (Foster et al.,

2021; Monmarché, 2021; Foster et al., 2024; Johnston et al., 2024; Yu et al., 2023; Srinivasan & Nagaraj, 2025).

**Dimension-free Sample Complexity.** The line of work on dimension-free sampling complexity originates from the connection between Langevin dynamics and optimization. The convergence guarantees of first-order optimization methods typically do not depend explicitly on the ambient dimension $d$ (Nesterov, 2013). In contrast, the convergence rates of Langevin-based sampling algorithms often exhibit an explicit dependence on $d$, stemming from the presence of isotropic Gaussian noise in the sampling dynamics. To bridge this gap, Freund et al. (2022) proved a dimension-free convergence rate for overdamped Langevin dynamics in two settings. When $V$ is $\alpha$-strongly convex and $L$-Lipschitz, they proved a dimension-free sample complexity of $\Theta(L^2/(\alpha^2\epsilon^2))$ in Wasserstein distance. When $V$ is $\alpha$-strongly convex and $\beta$-smooth, it characterized the sample complexity via an upper bound $\mathbf{H}$ of the Hessian matrix $\nabla^2 V$, with sample complexity $\Theta(\kappa^2\beta^{-1}\operatorname{tr}(\mathbf{H})/\epsilon^2)$ in KL divergence. In particular, when $V$ has a ridge separable structure with mild conditions, the sample complexity is independent of the dimension. Under the same setting and notation, Liu et al. (2023) analyzed underdamped Langevin sampling with a doubly randomized algorithm and showed that a sample complexity of order $\Theta\big(\kappa[\beta^{-1}\operatorname{tr}(\mathbf{H})]^{1/3}\epsilon^{-2/3}\big)$ is sufficient in Wasserstein distance. However, the corresponding sample complexity for underdamped Langevin dynamics in the KL divergence remains unexplored.

**Shifted Composition.** Altschuler & Chewi (2024b) proposed a KL local error framework that reduces the problem of establishing tight convergence bounds for sampling algorithms to the verification of local assumptions. This framework was first developed for overdamped Langevin dynamics in Altschuler & Chewi (2024b) and was later extended to the underdamped setting in Altschuler et al. (2025). A key technical ingredient is the construction of an auxiliary process interpolating between the laws of two stochastic processes, together with the use of a shifted composition rule; these ideas were introduced in Altschuler & Chewi (2024a) and Altschuler & Chewi (2025). More recently, this framework was further applied in Zhang (2025), where an improved cross-regularity analysis was developed, leading to faster convergence rates for a deterministic double midpoint method under higher-order differentiable assumptions.

## 3. Preliminaries

In this section, we introduce the ULMC discretization methods, including the standard ULMC and the randomized midpoint discretization (RMD, Shen & Lee 2019). We also state the assumptions on the invariant distribution $\pi$ used in this paper, and introduce several notions from numerical

analysis, such as the weak/strong KL local errors and the cross-regularity condition.

### 3.1. Underdamped Langevin Monte Carlo

The practical application of underdamped Langevin dynamics (ULD) requires the construction of a discrete-time sampling algorithm, named Underdamped Langevin Monte Carlo (ULMC). It produces a Markov chain $\{\mathbf{X}_{nh}^{\text{alg}}, \mathbf{P}_{nh}^{\text{alg}}\}_{n \in \mathbb{N}}$ through the use of an appropriate numerical discretization scheme. In this paper, we focus on two discretization methods, standard ULMC and the randomized midpoint, both of which are motivated by the equivalent integral representation of the ULD (1.2) given below:

$$
\mathbf{X}_t = \mathbf{X}_0 + (1 - e^{-\gamma t})/\gamma \cdot \mathbf{P}_0 + \boldsymbol{\xi}_{0,t}^{(1)}
$$
$$
- \int_0^t \frac{1 - e^{-\gamma(t-s)}}{\gamma} \nabla V(\mathbf{X}_s) \, \mathrm{d}s,
$$
$$
\mathbf{P}_t = e^{-\gamma t} \mathbf{P}_0 + \boldsymbol{\xi}_{0,t}^{(2)} - \int_0^t e^{-\gamma(t-s)} \nabla V(\mathbf{X}_s) \, \mathrm{d}s, \quad (3.1)
$$

where the random processes $\boldsymbol{\xi}_{s,t}^{(1)}$ and $\boldsymbol{\xi}_{s,t}^{(2)}$ are given by the Itô integral

$$
\boldsymbol{\xi}_{s,t}^{(1)} := \sqrt{2\gamma} \int_s^{s+t} \frac{1 - e^{-\gamma(s+t-u)}}{\gamma} \, \mathrm{d}\mathbf{B}_u,
$$
$$
\boldsymbol{\xi}_{s,t}^{(2)} := \sqrt{2\gamma} \int_s^{s+t} e^{-\gamma(s+t-u)} \, \mathrm{d}\mathbf{B}_u.
$$

The discretization methods are essentially numerical approximations of the intractable integral terms in (3.1).

**Standard ULMC.** The standard ULMC is arguably the simplest discretization method. With a given step size $h$, ULMC approximates $\nabla V(\mathbf{X}_t)$ in (1.2) with $\nabla V(\mathbf{X}_{nh}^{\text{ULMC}})$ for $t \in [nh, (n+1)h)$. The integral terms in (3.1) thus have closed-form solutions with $\nabla V(\mathbf{X}_{nh}^{\text{ULMC}})$ being a constant vector. Therefore, the standard ULMC proceeds with the following iterations:

$$
\mathbf{X}_{(n+1)h}^{\text{ULMC}} = \mathbf{X}_{nh} + (1 - e^{-\gamma h})/\gamma \cdot \mathbf{P}_{nh}^{\text{ULMC}} + \boldsymbol{\xi}_{nh,h}^{(1)}
$$
$$
- \frac{1}{\gamma}\Big(h - \frac{1 - e^{-\gamma h}}{\gamma}\Big) \nabla V(\mathbf{X}_{nh}^{\text{ULMC}}),
$$
$$
\mathbf{P}_{(n+1)h}^{\text{ULMC}} = e^{-\gamma h} \mathbf{P}_{nh}^{\text{ULMC}} + \boldsymbol{\xi}_{nh,h}^{(2)}
$$
$$
- (1 - e^{-\gamma h})/\gamma \cdot \nabla V(\mathbf{X}_{nh}^{\text{ULMC}}). \quad (3.2)
$$

**Randomized Midpoint Discretization (RMD).** The randomized midpoint discretization, first proposed in Shen & Lee (2019), aims to provide a more accurate estimation of the integral terms in (3.1) by replacing the integral with the expectation over a randomized stepsize. In this paper, we follow the doubly randomized implementation in Altschuler et al. (2025). In detail, let $\{(u_n, v_n)\}_{n \in \mathbb{N}}$ be i.i.d. random

variables on $[0,1]^2$, independent of the Brownian motion, with distribution

$$
\mathbb{P}(u_n \in A) = \int_{A \cap [0,1]} \frac{h(1 - e^{-\gamma(1-u)h})}{h - (1 - e^{-\gamma h})/\gamma} \, \mathrm{d}u,
$$
$$
\mathbb{P}(v_n \in A) = \int_{A \cap [0,1]} \frac{h\gamma e^{-\gamma(1-v)h}}{1 - e^{-\gamma h}} \, \mathrm{d}v. \quad (3.3)
$$

The numerical scheme aims to replace the first integral term with $[h - (1 - e^{-\gamma h})/\gamma] \nabla V(\mathbf{X}_{(n+u_n)h})/\gamma$. Observe that the expectation over $u_n$ satisfies

$$
\frac{1}{\gamma}\Big(h - \frac{1 - e^{-\gamma h}}{\gamma}\Big) \mathbb{E}_{u_n}[\nabla V(\mathbf{X}_{(n+u_n)h})]
$$
$$
= \int_{nh}^{(n+1)h} \frac{e^{-\gamma((n+1)h-s)}}{\gamma} \nabla V(\mathbf{X}_s) \, \mathrm{d}s.
$$

Thus, the expectation above is an unbiased estimation of the first integral term. A similar property holds for the integral term in the equation of the momentum. However, the "randomized midpoint" vectors $\mathbf{X}_{(n+u_n)h}$ and $\mathbf{X}_{(n+v_n)h}$ are unavailable in general. Instead, the numerical scheme approximates them with auxiliary vectors $\widehat{\mathbf{X}}_{(n+u_n)h}^+$ and $\widehat{\mathbf{X}}_{(n+v_n)h}^{++}$, respectively, both obtained using standard ULMC starting from $(\mathbf{X}_{nh}^{\text{RM}}, \mathbf{P}_{nh}^{\text{RM}})$. In summary, the randomized midpoint discretization calculates

$$
\widehat{\mathbf{X}}_{(n+u_n)h}^+ = \mathbf{X}_{nh}^{\text{RM}} + (1 - e^{-\gamma u_n h})/\gamma \cdot \mathbf{P}_{nh}^{\text{RM}} + \boldsymbol{\xi}_{nh,u_n h}^{(1)}
$$
$$
- \frac{1}{\gamma}\Big(u_n h - \frac{1 - e^{-\gamma u_n h}}{\gamma}\Big) \nabla V(\mathbf{X}_{nh}^{\text{RM}}),
$$
$$
\widehat{\mathbf{X}}_{(n+v_n)h}^{++} = \mathbf{X}_{nh}^{\text{RM}} + (1 - e^{-\gamma v_n h})/\gamma \cdot \mathbf{P}_{nh}^{\text{RM}} + \boldsymbol{\xi}_{nh,v_n h}^{(1)}
$$
$$
- \frac{1}{\gamma}\Big(v_n h - \frac{1 - e^{-\gamma v_n h}}{\gamma}\Big) \nabla V(\mathbf{X}_{nh}^{\text{RM}}),
$$
$$
\mathbf{X}_{(n+1)h}^{\text{RM}} = \mathbf{X}_{nh}^{\text{RM}} + (1 - e^{-\gamma h})/\gamma \cdot \mathbf{P}_{nh}^{\text{RM}} + \boldsymbol{\xi}_{nh,h}^{(1)}
$$
$$
- \frac{1}{\gamma}\Big(h - \frac{1 - e^{-\gamma h}}{\gamma}\Big) \nabla V(\widehat{\mathbf{X}}_{(n+u_n)h}^+),
$$
$$
\mathbf{P}_{(n+1)h}^{\text{RM}} = e^{-\gamma h} \mathbf{P}_{nh}^{\text{RM}} + \boldsymbol{\xi}_{nh,h}^{(2)}
$$
$$
- (1 - e^{-\gamma h})/\gamma \cdot \nabla V(\widehat{\mathbf{X}}_{(n+v_n)h}^{++}). \quad (3.4)
$$

### 3.2. Assumptions

We make the following assumptions on the convexity and smoothness of the function $V$.

**Assumption 3.1.** The function $V(\cdot)$ is twice-differentiable, and $\beta$-smooth. Furthermore, there exists a constant $\alpha \geq 0$, such that the Hessian of $V$ satisfies,

$$
\alpha \mathbf{I} \preceq \nabla^2 V \preceq \mathbf{H} \preceq \beta \mathbf{I},
$$

where $\mathbf{H}$ is a known positive semi-definite matrix.

**Remark 3.2.** In prior work (Cheng et al., 2018), it is common to assume $V$ is strongly-convex, i.e., $\alpha > 0$. In this paper, we adopt a more general perspective and consider two cases separately: the strongly convex case ($\alpha > 0$) and the general convex case ($\alpha = 0$).

**Remark 3.3.** Compared with the traditional analyses of ULMC that characterize the curvature of $V$ using only the smoothness parameter $\beta$, our analysis depends on the instance-dependent Hessian upper bound $\mathbf{H}$, which better captures the relationships of the convergence of ULMC with the curvature. Furthermore, this is the core assumption that enables the **dimension-independent** analysis: The dependence on the ambient $d$ in the traditional analysis is caused by the term $\text{tr}(\nabla^2 V)$, which is bounded by $\beta d$ with only the assumption of the universal smoothness $\beta$. In our analysis, however, we have $\text{tr}(\nabla^2 V) \leq \text{tr}(\mathbf{H})$ provided that $\nabla^2 V \preceq \mathbf{H}$, where $\text{tr}(\mathbf{H})$ can be much smaller than $\beta d$.

We provide several concrete examples where $\text{tr}(\mathbf{H}) \ll \beta d$.

- When $V$ is ridge-separable, i.e., $V(\mathbf{x}) = \frac{1}{n} \sum_i \sigma(\mathbf{a}_i^\top \mathbf{x})$, where $\sigma''(z) \leq L$ and $\|\mathbf{a}_i\| \leq R$. In this case, $\nabla^2 V(\mathbf{x}) = \frac{1}{n} \sum_i \sigma''(\mathbf{a}_i^\top \mathbf{x}) \mathbf{a}_i \mathbf{a}_i^\top \preceq \frac{L}{n} \sum_i \mathbf{a}_i \mathbf{a}_i^\top := \mathbf{H}$, and $\text{tr}(\mathbf{H}) = \frac{L}{n} \sum_i \|\mathbf{a}_i\|^2 \leq LR^2$. There are cases where $\beta d \gg \text{tr}(\mathbf{H})$. For example, if $\mathbf{a}_i = \epsilon_i \boldsymbol{\mu}$ with $\|\boldsymbol{\mu}\| = R$ and $\epsilon_i \in \{\pm 1\}$, then $\mathbf{H} = L\boldsymbol{\mu}\boldsymbol{\mu}^\top$, such that $\text{tr}(\mathbf{H}) = LR^2$, while $\beta d = LR^2 d$.

- When the eigenvalues follow power-law decay, i.e., $\mathbf{H} = \text{diag}(\{\lambda_i\})$ where $\lambda_i \simeq i^{-a}$, $\text{tr}(\mathbf{H}) = \sum_i \lambda_i = O(1)$ if $a > 1$, while $\beta d = \Theta(d)$. The power-law decay is verified in a number of works in the kernel regime (Bietti & Mairal, 2019; Bordelon et al., 2020, e.g.).

We focus on the Underdamped Langevin Dynamics (ULD), which evolves in the phase plane $(\boldsymbol{X}_t, \boldsymbol{P}_t)$ of the displacement $\boldsymbol{X}_t$ and the momentum $\boldsymbol{P}_t$.

**Error of one-step discretization.** We first introduce the notation for the one-step discretization error, distinguishing between two types: weak and strong errors. This terminology is adopted from the classical theory of weak and strong convergence in numerical analysis (see, e.g., Section 9 in Kloeden & Platen (1992)).

**Definition 3.4.** Suppose that the initial conditions of the ULD and the numerical discretization method alg are $(\boldsymbol{X}_0, \boldsymbol{P}_0) = (\boldsymbol{X}_0^{\text{alg}}, \boldsymbol{P}_0^{\text{alg}}) = (\mathbf{x}, \mathbf{p})$. The one-step weak error $\mathcal{E}^w$ and strong error $\mathcal{E}^s$ are defined as

$$\mathcal{E}^w(\mathbf{x}, \mathbf{p}) := h^{-1}\|\mathbb{E}\boldsymbol{X}_h^{\text{alg}} - \mathbb{E}\boldsymbol{X}_h\| \vee \|\mathbb{E}\boldsymbol{P}_h^{\text{alg}} - \mathbb{E}\boldsymbol{P}_h\|;$$

$$\mathcal{E}^s(\mathbf{x}, \mathbf{p}) := h^{-1}\|\boldsymbol{X}_h^{\text{alg}} - \boldsymbol{X}_h\|_{L_2} \vee \|\boldsymbol{P}_h^{\text{alg}} - \boldsymbol{P}_h\|_{L_2}.$$

Using this definition, Altschuler et al. (2025) proposed a KL local error framework, which characterizes the convergence

of discretization methods with their one-step discretization errors, including the weak and strong errors. Furthermore, we need the following one-step cross-regularity condition.

Let $(\mathcal{P}_t)_{t \geq 0}$ denote the Markov semigroup associated with the underdamped Langevin (1.2), and let $\mathcal{P}_h$ be its time-$h$ transition operator. Let $\mathcal{P}_h^{\text{alg}}$ denote the Markov transition kernel induced by one step of the numerical integrator with step size $h$. Then $(\boldsymbol{X}_h, \boldsymbol{P}_h)$ and $(\boldsymbol{X}_h^{\text{alg}}, \boldsymbol{P}_h^{\text{alg}})$ satisfy

$$(\boldsymbol{X}_h, \boldsymbol{P}_h) \sim \delta_{\mathbf{x},\mathbf{p}}\mathcal{P}_h, \quad (\boldsymbol{X}_h^{\text{alg}}, \boldsymbol{P}_h^{\text{alg}}) \sim \delta_{\mathbf{x},\mathbf{p}}\mathcal{P}_h^{\text{alg}}.$$

The cross-regularity condition characterizes the divergence of two transition kernels $\mathcal{P}_h$ and $\widetilde{\mathcal{P}}_h$ starting from different initial conditions:

**Definition 3.5.** Transition kernels $\mathcal{P}$ and $\widetilde{\mathcal{P}}$ satisfy the *cross-regularity condition* with function $b(\mathbf{x}, \mathbf{p})$ if for any initial conditions $(\mathbf{x}, \mathbf{p}), (\bar{\mathbf{x}}, \bar{\mathbf{p}}) \in \mathbb{R}^{2d}$, the distributions $\delta_{\bar{\mathbf{x}}, \bar{\mathbf{p}}}\widetilde{\mathcal{P}}$ and $\delta_{\mathbf{x}, \mathbf{p}}\mathcal{P}$ satisfy

$$\text{KL}\big(\delta_{\mathbf{x},\mathbf{p}}\widetilde{\mathcal{P}}_h \| \delta_{\bar{\mathbf{x}},\bar{\mathbf{p}}}\mathcal{P}_h\big) \lesssim \frac{\|\mathbf{x} - \bar{\mathbf{x}}\|^2}{\gamma h^3} + \frac{\|\mathbf{p} - \bar{\mathbf{p}}\|^2}{\gamma h} + b^2(\mathbf{x}, \mathbf{p}).$$

Once the one-step errors of discretization methods are obtained, the KL local error framework provides the convergence rate in a plug-and-play way, for which we will provide a comprehensive guideline in Section 3.3.

### 3.3. KL local error framework

In this section, we give a detailed description of the KL local error framework proposed in Altschuler et al. (2025). We consider the ULD chain $(\boldsymbol{\psi}_n)_{n=0}^N$ with $N$ steps, defined as

$$\boldsymbol{\psi}_n|\boldsymbol{\psi}_{n-1} \sim \delta_{\boldsymbol{\psi}_{n-1}}\mathcal{P}_h, \qquad n = 1, 2, \ldots, N, \qquad (3.5)$$

with initial condition $\boldsymbol{\psi}_0 = (\boldsymbol{X}_0, \boldsymbol{P}_0) \sim \nu$. Similarly, we define the chain of numerical discretization $(\boldsymbol{\psi}_n^{\text{alg}})_{n=0}^N$ as

$$\boldsymbol{\psi}_n^{\text{alg}}|\boldsymbol{\psi}_{n-1}^{\text{alg}} \sim \delta_{\boldsymbol{\psi}_{n-1}^{\text{alg}}}\mathcal{P}_h^{\text{alg}}, \qquad n = 1, 2, \ldots, N, \quad (3.6)$$

with initial condition $\boldsymbol{\psi}_0^{\text{alg}} = (\boldsymbol{X}_0^{\text{alg}}, \boldsymbol{P}_0^{\text{alg}}) \sim \mu$.

At the center of the framework is the *shifted operator* defined as follows:

**Definition 3.6.** Given a random process $\boldsymbol{\psi} = (\boldsymbol{X}, \boldsymbol{P})$, the shifted process towards another target process $\widehat{\boldsymbol{\psi}} = (\widehat{\boldsymbol{X}}, \widehat{\boldsymbol{P}})$, with parameter $\eta^{\mathbf{x}}, \eta^{\mathbf{p}}$, is defined as

$$\mathcal{T}_{\eta^{\mathbf{x}}, \eta^{\mathbf{p}}}(\boldsymbol{\psi}, \widehat{\boldsymbol{\psi}}) = \big(\boldsymbol{X}, \boldsymbol{P} + \eta^{\mathbf{x}}(\boldsymbol{X} - \widehat{\boldsymbol{X}}) + \eta^{\mathbf{p}}(\boldsymbol{P} - \widehat{\boldsymbol{P}})\big).$$

Using the shifted operator, with a target process $\boldsymbol{\psi}_n^{\text{target}}$, we further define two processes $\boldsymbol{\psi}^{\text{aux}}, \boldsymbol{\psi}^{\text{sh}}$ iteratively:

$$\boldsymbol{\psi}_n^{\text{aux}}|\boldsymbol{\psi}_{n-1}^{\text{sh}} \sim \delta_{\boldsymbol{\psi}_{n-1}^{\text{sh}}}\mathcal{P}_h,$$

$$\psi_n^{\text{sh}} = \mathcal{T}_{\eta_n^{\mathbf{x}}, \eta_n^{\mathbf{P}}}(\psi_n^{\text{aux}}, \psi_n^{\text{target}}) =$$
$$\left(X_n^{\text{aux}}, P_n^{\text{aux}} + \eta_n^{\mathbf{x}}(X_n^{\text{target}} - X_n^{\text{aux}}) + \eta_n^{\mathbf{P}}(P_n^{\text{target}} - P_n^{\text{aux}})\right),$$
(3.7)

with initial condition $\psi_0^{\text{sh}} = (X_0, P_0) \sim \nu$ and $\eta_n^{\mathbf{x}}, \eta_n^{\mathbf{P}}$ as predetermined constants (The concrete value to be discussed in Appendix A). The auxiliary step $\psi_n^{\text{aux}}$ applies the ULD transition kernel $\mathcal{P}_h$ to the previous shifted state $\psi_{n-1}^{\text{sh}}$, while the shifting step modifies the momentum term of the auxiliary process by interpolating towards the target process. When the target process is selected as $\psi^{\text{alg}}$, Altschuler et al. (2025) proved the following theorem, by reducing the KL-divergence between the auxiliary process and the ULD process to the calculation of strong error and weak error.

**Theorem 3.7** (Theorem 4.1 in Altschuler et al. 2025). Assume $h \lesssim \gamma^{-1} \wedge \gamma/\beta$. Let $\psi^{\text{aux}}$ be the auxiliary process defined in (3.7). For $n \leq N - 1$, let $\nu_n^{\text{aux}}$ be the distribution of the auxiliary process $\psi_n^{\text{aux}}$, and $\nu_n$ be the distribution of $\psi_n$. We denote by $W^2 = \mathcal{W}_2^2(\mu, \nu)$ the squared Wasserstein-2 distance induced by the twisted norm

$$(\mathbf{x}, \mathbf{p}) \to \sqrt{\|\mathbf{x}\|^2 + (\gamma + 1/(Nh))^{-2}\|\mathbf{p}\|^2}.$$

Then the KL-divergence between $\nu_n^{\text{aux}}$ and $\nu_n$ satisfies

$$\text{KL}\left(\nu_n^{\text{aux}} \| \nu_n\right) \lesssim CW^2 + A_w\left(\bar{\mathcal{E}}^w\right)^2 + A_s\left(\bar{\mathcal{E}}^s\right)^2.$$

where we define $\bar{f} = \max_{1 \leq i \leq n-1} \|f\|_{L^2(\mu[\mathcal{P}^{\text{alg}}]^i)}$ for $f \in \{b, \mathcal{E}^w, \mathcal{E}^s\}$. Here $C, A_w, A_s$ are parameter-dependent constants, which we will provide a detailed discussion in Appendix A.

As a corollary, the KL-divergence between the distribution resulting from the composition of $N - 1$ steps of $\mathcal{P}^{\text{alg}}$, and one step of $\widetilde{\mathcal{P}}$, and the distribution from $N$ steps of $\mathcal{P}$, can be upper bounded as follows:

**Corollary 3.8.** With the same setting and notation as Theorem 3.7, the KL-divergence between the distribution resulting from the composition of $N - 1$ steps of $\mathcal{P}^{\text{alg}}$, and one step of $\widetilde{\mathcal{P}}$, and the distribution from $N$ steps of $\mathcal{P}$, can be upper bounded as follows:

$$\text{KL}\left(\mu(\mathcal{P}^{\text{alg}})^{N-1}\widetilde{\mathcal{P}} \| \nu\mathcal{P}^N\right) \lesssim C\mathcal{W}_2^2(\mu, \nu)$$
$$+ A_w\left(\bar{\mathcal{E}}^w\right)^2 + A_s\left(\bar{\mathcal{E}}^s\right)^2 + \bar{b}^2.$$

**Remark 3.9.** Note that the standard analysis of strong and weak local errors typically depends not only on constant terms but also on the initial state of the process itself (see e.g., Section 5 in Chewi (2025)). Substituting such bounds into Theorem 3.7 and Corollary 3.8 would therefore introduce terms of the form $\mathbb{E}_{\mu[\mathcal{P}^{\text{alg}}]^n}[\|\mathbf{p}\|^2]$ and $\mathbb{E}_{\mu[\mathcal{P}^{\text{alg}}]^n}[\|\nabla V(\mathbf{x})\|^2]$. These quantities can be controlled using a change-of-measure argument in terms of

the KL divergence between $\mu[\mathcal{P}^{\text{alg}}]^n$ and the invariant distribution $\pi$ via the Donsker–Varadhan variational formula. In particular, with an additional Lipschitz assumption (See Appendix A for details), we can further define $\widetilde{f} = \max_{1 \leq i \leq n-1} \|f\|_{L^2(\nu_i^{\text{aux}})}$ for $f \in \{b, \mathcal{E}^w, \mathcal{E}^s\}$ with respect to the auxiliary process, and replace $\bar{f}$ with $\widetilde{f}$ in Theorem 3.7 and Corollary 3.8. This enables a closed-form recursive error control argument, as detailed in Lemma 6.1.

## 4. Dimension-free Analysis of ULMC

To apply the KL local error framework described in Section 3.3, we first calculate the strong and weak error. Note that the standard calculation (e.g. Section 5 in Chewi (2025)) is not dimension-free. By refining the analysis, we establish a dimension-free version of these bounds that depend on $\text{tr}(\mathbf{H})$.

**Lemma 4.1** (Strong and weak error for ULMC, dimension-free). Let the strong error $\mathcal{E}^s$ and the weak error $\mathcal{E}^w$ be defined in Definition 3.4. Under Assumption 3.1, the strong and weak errors coincide and satisfy the following bounds:

$$\mathcal{E}^w(\mathbf{x}, \mathbf{p}) \vee \mathcal{E}^s(\mathbf{x}, \mathbf{p}) \lesssim \beta^{1/2}h^2\|\mathbf{p}\|_{\mathbf{H}} + \beta h^3\|\nabla V(\mathbf{x})\|$$
$$+ \beta^{1/2}\gamma^{1/2}h^{5/2}\sqrt{\text{tr}(\mathbf{H})}.$$

When $\mathbf{H} = \beta\mathbf{I}$, direct calculation shows that $\|\mathbf{p}\|_{\mathbf{H}} = \beta^{1/2}\|\mathbf{p}\|$, $\text{tr}(\mathbf{H}) = \beta d$. Then, this result reduces to Lemma 5.1 in Altschuler et al. (2025). Compared with prior results, our analysis offers improvements in two key directions. First, we replace the worst-case $\sqrt{d}$ dependence by a trace-dependent term. Second, we observe that using the standard Euclidean norm $\|\mathbf{p}\|$ is suboptimal for our purposes. Therefore, we consider the $\mathbf{H}$-norm. This refinement yields a tighter analysis and plays an essential role in establishing the final dimension-free bounds. We will discuss this with more details in Remark 6.2.

Applying similar considerations, we derive the dimension-free version of the cross-regularity assumption for ULMC below.

**Lemma 4.2** (Cross-regularity for ULMC, dimension-free). For $\mathcal{P}' = \mathcal{P}^{\text{ULMC}}$, $\mathcal{P}$ as the ULD transition kernel, we have:

$$\text{KL}(\delta_{\mathbf{x}, \mathbf{p}}\mathcal{P}' \| \delta_{\bar{\mathbf{x}}, \bar{\mathbf{p}}}\mathcal{P}) \lesssim \frac{\|\mathbf{x} - \bar{\mathbf{x}}\|^2}{\gamma h^3} + \frac{\|\mathbf{p} - \bar{\mathbf{p}}\|^2}{\gamma h}$$
$$+ \frac{\beta h^3}{\gamma}\|\mathbf{p}\|_{\mathbf{H}}^2 + \beta h^4\,\text{tr}(\mathbf{H}) + \frac{\beta^2 h^5}{\gamma}\|\nabla V(\mathbf{x})\|^2.$$

Thus, ULMC satisfies the cross-regularity condition (Definition 3.5) with

$$b^2(\mathbf{x}, \mathbf{p}) = \frac{\beta h^3}{\gamma}\|\mathbf{p}\|_{\mathbf{H}}^2 + \beta h^4\,\text{tr}(\mathbf{H}) + \frac{\beta^2 h^5}{\gamma}\|\nabla V(\mathbf{x})\|^2.$$

With these dimension-free calculations in place, we are now ready to establish a dimension-free sample complexity bound for ULMC.

**Strongly Convex.** We first consider the strongly convex setting ($\alpha > 0$) with $\gamma = \sqrt{32\beta}$.

**Theorem 4.3.** Suppose $\alpha > 0$ and $\gamma = \sqrt{32\beta}$. Under Assumption 3.1, let $\mathcal{P}' = \mathcal{P}^{\mathrm{ULMC}}$. Let $\pi$ be the invariant distribution of the underdamped Langevin dynamics (1.2), $\mu$ be the initial distribution of the algorithm. For any $0 < \epsilon \leq [\mathrm{tr}(\mathbf{H})]^{1/2}\beta^{-1/2}\kappa^{-1/2}$, if

$$h = \widetilde{\Theta}\left(\frac{\epsilon}{[\kappa\,\mathrm{tr}(\mathbf{H})]^{1/2}}\right),$$

$$N = \widetilde{\Theta}\left(\frac{\kappa^{3/2}\beta^{-1/2}[\mathrm{tr}(\mathbf{H})]^{1/2}}{\epsilon}\right),$$

the KL divergence between the law of the process with $N$ steps of ULMC and the invariant distribution $\pi$ can be upper bounded by

$$\mathrm{KL}\big(\mu(\mathcal{P}')^N\|\pi\big) \leq \epsilon^2.$$

Theorem 4.3 yields a dimension-free sample complexity of $\widetilde{\mathcal{O}}\big(\kappa^{3/2}\beta^{-1/2}[\mathrm{tr}(\mathbf{H})]^{1/2}/\epsilon\big)$, guaranteeing that the KL divergence is at most $\epsilon^2$. Moreover, when $\mathbf{H} = \beta\mathbf{I}$, our result matches Altschuler et al. (2025). It improves upon the dimension-free overdamped KL bound $\Theta(\kappa^2\beta^{-1}\,\mathrm{tr}(\mathbf{H})/\epsilon^2)$ in Freund et al. (2022). Moreover, using Talagrand's $T_2$ inequality, Theorem 4.3 implies a sample complexity of $\widetilde{\mathcal{O}}(\kappa^2\beta^{-1}[\mathrm{tr}(\mathbf{H})]^{1/2}/\epsilon)$ to guarantee that the Wasserstein-2 distance is at most $\epsilon$.

**General Convex.** We then consider the general convex setting ($\alpha = 0$) with $\gamma = \sqrt{32\beta}$.

**Theorem 4.4.** Suppose $\alpha = 0$ and $\gamma = \sqrt{32\beta}$. Under Assumption 3.1, let $\mathcal{P}' = \mathcal{P}^{\mathrm{ULMC}}$. Let $\pi$ be the invariant distribution of ULD (1.2), $\mu$ be the initial distribution of the algorithm. For any $0 < \epsilon \leq \beta^{1/2}W$, if

$$h = \Theta\left(\min\left\{\frac{\epsilon^2}{\beta^{1/2}\big(\mathrm{tr}(\mathbf{H})\big)^{1/2}W}, \frac{\epsilon^2}{\beta^{3/2}W^2}\right\}\right),$$

and

$$N = \Theta\left(\max\left\{\frac{\beta[\mathrm{tr}(\mathbf{H})]^{1/2}W^3}{\epsilon^4}, \frac{\beta^2W^4}{\epsilon^4}\right\}\right),$$

the KL divergence between the law of the process with $N$ steps of ULMC and the invariant distribution $\pi$ can be upper bounded by

$$\mathrm{KL}\big(\mu(\mathcal{P}')^N\|\pi\big) \leq \epsilon^2.$$

To the best of our knowledge, our work is the first to establish a dimension-free sample complexity bound for ULMC in the general convex setting. Moreover, our bound matches Altschuler et al. (2025) when $\mathbf{H} = \beta\mathbf{I}$.

## 5. Dimension-free Analysis of RMD

In this section, our goal is to develop a *dimension-free* analysis of the randomized midpoint discretization (RMD) introduced in (3.4). To this end, in order to apply the KL local framework, we first establish refined bounds on the strong and weak local errors.

**Lemma 5.1** (Strong and weak error for RMD, dimension-free). Let the strong error $\mathcal{E}^s$ and the weak error $\mathcal{E}^w$ be defined in Definition 3.4. Under Assumption 3.1, the following bounds hold:

$$\mathcal{E}^w(\mathbf{x}, \mathbf{p}) \lesssim \beta^{3/2}h^4\|\mathbf{p}\|_{\mathbf{H}} + \beta^2h^5\|\nabla V(\mathbf{x})\|$$
$$+ \beta^{3/2}\gamma^{1/2}h^{9/2}\sqrt{\mathrm{tr}(\mathbf{H})},$$
$$\mathcal{E}^s(\mathbf{x}, \mathbf{p}) \lesssim \beta^{1/2}h^2\|\mathbf{p}\|_{\mathbf{H}} + \beta h^3\|\nabla V(\mathbf{x})\|$$
$$+ \beta^{1/2}\gamma^{1/2}h^{5/2}\sqrt{\mathrm{tr}(\mathbf{H})},$$

where $\|\mathbf{p}\|_{\mathbf{H}} = \sqrt{\mathbf{p}^\top\mathbf{H}\mathbf{p}}$.

Analogous to Lemma 4.1, we replace the $\sqrt{d}$ term with an $\mathrm{tr}(\mathbf{H})$-dependent term and adopt the $\mathbf{H}$-norm. Moreover, as in Altschuler et al. (2025), the randomized midpoint discretization together with the specific choice of the randomized midpoint distribution in (3.3) yields an improved bound on the weak error, which in turn leads to a sharper rate in the final convergence bound. Equipped with Lemma 5.1, we establish theoretical guarantees for RMD in two distinct settings: strongly convex and generally convex.

**Strongly Convex.** We first consider the strongly convex setting ($\alpha > 0$) with $\gamma = \sqrt{32\beta}$.

**Theorem 5.2.** Suppose $\alpha > 0$ and $\gamma = \sqrt{32\beta}$. Under Assumption 3.1, let $\mathcal{P}^{\mathrm{alg}} = \mathcal{P}^{\mathrm{RM}}$, $\mathcal{P}' = \mathcal{P}^{\mathrm{ULMC}}$. Let $\pi$ be the invariant distribution of the underdamped Langevin dynamics (1.2), $\mu$ be the initial distribution of the algorithm. We denote by $W^2 := \mathcal{W}_2^2(\mu, \pi)$ the squared 2-Wasserstein distance induced by the twisted norm $(\mathbf{x}, \mathbf{p}) \to \sqrt{\|\mathbf{x}\|^2 + (\gamma + 1/(Nh))^{-2}\|\mathbf{p}\|^2}$.
For any $0 < \epsilon \leq [\mathrm{tr}(\mathbf{H})]^{1/2}\beta^{-3/2}\kappa^{-3/4}$, if

$$h = \widetilde{\Theta}\big(\beta^{-1/6}[\mathrm{tr}(\mathbf{H})]^{-1/3}\epsilon^{2/3}\big),$$
$$N = \widetilde{\Theta}\big(\kappa\big[\beta^{-1}\,\mathrm{tr}(\mathbf{H})\big]^{1/3}\epsilon^{-2/3}\big),$$

the KL divergence between the law of the process with $N-1$ steps of RMD and one step of ULMC and the invariant distribution $\pi$ can be upper bounded by

$$\mathrm{KL}\big(\mu(\mathcal{P}^{\mathrm{alg}})^{N-1}\mathcal{P}'\|\pi\big) \leq \epsilon^2.$$

Theorem 5.2 yields a dimension-free sample complexity of $\widetilde{\Theta}\big(\kappa[\beta^{-1}\,\mathrm{tr}(\mathbf{H})]^{1/3}\epsilon^{-2/3}\big)$ to guarantee that the KL divergence is at most $\epsilon^2$. When $\mathbf{H} = \beta\mathbf{I}$, the

sample complexity is reduced to $\widetilde{\Theta}(\kappa d^{1/3}\epsilon^{-2/3})$, which matches the result in Altschuler et al. (2025). In general, the dimension-free complexity can be substantially smaller than the direct dependence on $d$. Using Talagrand's $T_2$ inequality, Theorem 5.2 implies a sample complexity of $\widetilde{\mathcal{O}}(\kappa^{4/3}\beta^{-2/3}[\text{tr}(\mathbf{H})]^{1/3}\epsilon^{-2/3})$ to guarantee that the Wasserstein-2 distance is at most $\epsilon$. In contrast, Liu et al. (2023) proved a sample complexity of $\widetilde{\Theta}(\kappa^{5/3}\beta^{-2/3}[\text{tr}(\mathbf{H})]^{1/3}\epsilon^{-2/3})$ for a doubly randomized algorithm for underdamped Langevin dynamics in the Wasserstein distance. Thus, with the same dependence on $\text{tr}(\mathbf{H})$ and $1/\epsilon, \beta$, our result strictly improves the dependence on the condition number $\kappa$.

**Remark 5.3.** Following Altschuler et al. (2025), we modify the transition kernel at the final step of the algorithm to that of ULMC in order to apply the cross-regularity property (Lemma 4.2). This allows us to bypass the technical difficulty of establishing cross-regularity directly for the randomized midpoint method. Since our primary goal is to derive a dimension-free sample complexity bound, and since the same analysis can be easily adapted once cross-regularity is established for more general discretization schemes, we believe that this modification does not detract from the generality or significance of our results.

**General Convex.** We then consider the general convex setting ($\alpha = 0$) with $\gamma = \sqrt{32\beta}$.

**Theorem 5.4.** Suppose $\alpha = 0$ and $\gamma = \sqrt{32\beta}$. Under Assumption 3.1, let $\mathcal{P}^{\text{alg}} = \mathcal{P}^{\text{RM}}$, $\mathcal{P}' = \mathcal{P}^{\text{ULMC}}$. Let $\pi$ be the invariant distribution of the underdamped Langevin dynamics (1.2), $\mu$ be the initial distribution of the algorithm. For any $0 < \epsilon \leq \min\{\sqrt{\beta}W, [\text{tr}(\mathbf{H})]^{3/4}\beta^{-1}W^{-1/2}\}$, if

$$h = \widetilde{\Theta}\left(\frac{\epsilon}{\beta^{1/2}[\text{tr}(\mathbf{H})]^{1/4}W^{1/2}}\right),$$

$$N = \widetilde{\Theta}\left(\frac{\beta[\text{tr}(\mathbf{H})]^{1/4}W^{5/2}}{\epsilon^3}\right),$$

the KL divergence between the law of the process with $N-1$ steps of RMD and one step of ULMC and the invariant distribution $\pi$ can be upper bounded by

$$\text{KL}\big(\mu(\mathcal{P}^{\text{alg}})^{N-1}\mathcal{P}'\|\pi\big) \leq \epsilon^2.$$

Theorem 5.4 demonstrates a $\Theta(1/\epsilon^3)$ sample complexity, with polynomial dependence in $\beta, \text{tr}(\mathbf{H})$ and the Wasserstein distance $W^2$. Compared with Theorem 4.4, it shows that RMD achieves a substantial improvement over ULMC in efficiency, reducing the sampling complexity from $\Theta(1/\epsilon^4)$ to $\Theta(1/\epsilon^3)$. To the best of our knowledge, this is the first *dimension-free* sample complexity bound for RMD under the general convex setting. It remains an interesting open question whether the $\Theta(1/\epsilon^3)$ rate can be further improved in the general convex setting.

# 6. Overview of Proof

In this section, we use the analysis of RMD in the strongly convex setting (Theorem 5.2) to illustrate the proof strategy for our dimension-free results. Let $\pi$ be the invariant distribution of ULD (1.2). Using Corollary 3.8 with $\nu = \pi$, we can bound the KL divergence in Theorem 5.2 as follows:

$$\text{KL}\big(\mu(\mathcal{P}^{\text{alg}})^{N-1}\widetilde{\mathcal{P}}\|\pi\big) \lesssim C\mathcal{W}_2^2(\mu, \pi) \\ + A_w\big(\bar{\mathcal{E}}^w\big)^2 + A_s\big(\bar{\mathcal{E}}^s\big)^2 + \bar{b}^2. \quad (6.1)$$

Since the Wasserstein term is small when $Nh$ is large, we focus on the remaining terms. Substituting Lemma 5.1 and Lemma 4.2 into (6.1), we have

$$A_w\big(\bar{\mathcal{E}}^w\big)^2 + A_s\big(\bar{\mathcal{E}}^s\big)^2 + \bar{b}^2 \lesssim \beta h^4 \log\left(\frac{3\gamma}{\alpha h}\right)\text{tr}(\mathbf{H}) \\ + \beta^{1/2}h^3 \log\left(\frac{3\gamma}{\alpha h}\right)\left[\max_{1\leq i\leq n-1}\mathbb{E}_{\nu_i^{\text{aux}}}\big[\|\mathbf{p}\|_{\mathbf{H}}^2\big]\right] \\ + \beta^{3/2}h^5 \log\left(\frac{3\gamma}{\alpha h}\right)\left[\max_{1\leq i\leq n-1}\mathbb{E}_{\nu_i^{\text{aux}}}\big[\|\nabla V(\mathbf{x})\|^2\big]\right]. \quad (6.2)$$

To continue, we require the following dimension-free change-of-measure lemma, which allows us to control the state-dependent terms in the recursion without introducing any explicit dimension dependence.

**Lemma 6.1** (Change-of-measure, dimension-free). Consider a measure $\mu \in \mathbb{R}^d \times \mathbb{R}^d$, and $-\beta\mathbf{I} \preceq \nabla^2 V(\mathbf{x}) \preceq \mathbf{H} \preceq \beta\mathbf{I}$. With $\pi(\mathbf{x}, \mathbf{p}) \propto \exp(-V(\mathbf{x}) - \frac{1}{2}\|\mathbf{p}\|^2)$, we have:

$$\mathbb{E}_\mu[\|\nabla V(\mathbf{x})\|^2] \lesssim \text{tr}(\mathbf{H}) + \beta\text{KL}(\mu\|\pi),$$
$$\mathbb{E}_\mu[\mathbf{p}^\top\mathbf{H}\mathbf{p}] \lesssim \text{tr}(\mathbf{H}) + \beta\text{KL}(\mu\|\pi).$$

**Remark 6.2.** For the inequality regarding $\nabla V(\mathbf{x})$, we apply Donsker-Varadhan's variational formula to get

$$\mathbb{E}_\mu[\|\nabla V(\mathbf{x})\|^2] \lesssim \beta\text{KL}(\mu\|\pi) \\ + \log\mathbb{E}_\pi\left[\exp\big(\|\nabla V(\mathbf{x})\|^2/(4\beta)\big)\right].$$

A direct analysis of the logarithmic moment generating function on the right-hand side leads to an explicit dependence on the dimension $d$, which is not dimension-free. To overcome this issue, we instead apply a Taylor expansion of the exponential and bound the expectation of each order separately. This refined analysis yields a tighter bound that depends only on $\text{tr}(\mathbf{H})$.

For the inequality with regard to the momentum term, note that under the invariant distribution $\pi$, the marginal distribution of $\mathbf{p}$ is Gaussian, implying $\mathbb{E}_\pi[\|\mathbf{p}\|^2] \simeq d$. This observation is crucial: it demonstrates that if we do not introduce the $\mathbf{H}$-matrix norm in the local error analysis (Lemma 5.1), the final complexity bound would remain dimension-dependent regardless of remaining analysis. This underscores our modification, using the $\mathbf{H}$-norm in the momentum error analysis.

Using Lemma 6.1, we can convert the maximum expectation over auxiliary processes before step $n$ into a term involving $\max_{1 \le i \le n-1} \mathrm{KL}(\nu_i^{\mathrm{aux}} \| \pi)$. Thus, (6.2) becomes

$$A_w (\bar{\mathcal{E}}^w)^2 + A_s (\bar{\mathcal{E}}^s)^2 + \bar{b}^2 \lesssim \beta h^4 \log \left( \frac{3\gamma}{\alpha h} \right) \mathrm{tr}(\mathbf{H})$$
$$+ \beta^{3/2} h^3 \log \left( \frac{3\gamma}{\alpha h} \right) \left[ \max_{1 \le i \le n-1} \mathrm{KL}(\nu_i^{\mathrm{aux}} \| \pi) \right]. \quad (6.3)$$

Finally, we can apply Theorem 3.7 for any $n \le N - 1$ to obtain

$$\max_{1 \le i \le n} \mathrm{KL}(\nu_i^{\mathrm{aux}} \| \pi) \lesssim C \mathcal{W}_2^2(\mu, \pi) + \beta h^4 \log \left( \frac{3\gamma}{\alpha h} \right) \mathrm{tr}(\mathbf{H})$$
$$+ \beta^{3/2} h^3 \log \left( \frac{3\gamma}{\alpha h} \right) \max_{1 \le i \le n-1} \mathrm{KL}(\nu_i^{\mathrm{aux}} \| \pi).$$

Substituting into (6.3), we obtain the final KL bound for Theorem 5.2. We conclude by choosing $h$ small enough so that the KL divergence is at most $\epsilon^2$.

## 7. Conclusion

In this paper, we establish the first dimension-free KL convergence guarantees for discretizations of underdamped Langevin dynamics. Our bounds depend on $\mathrm{tr}(\mathbf{H})$, where $\mathbf{H}$ is an upper bound on the Hessian $\nabla^2 V$, rather than on the ambient dimension $d$, yielding improved rates in regimes where $\mathrm{tr}(\mathbf{H}) \ll d$. We show that both standard ULMC and the randomized midpoint discretization (RMD) enjoy dimension-free KL convergence, and our results cover both the strongly convex and the general convex settings.

Two natural extensions remain open. The first is the nonconvex setting. A possible route is to establish asymptotic convergence under functional inequalities such as the log-Sobolev inequality (LSI) or the Poincaré inequality (PI). However, obtaining such results in a dimension-independent form appears substantially more challenging. For example, the nonconvex analysis of Altschuler et al. (2025) under LSI, see Theorem 5.7 therein, leads to bounds that depend explicitly on the ambient dimension.

The second extension is the problem with stochastic gradients (noise for the observed gradient). Since the stochastic-gradient update still induces a Markov transition kernel, the KL local error framework in Theorem 3.7 remains applicable. However, when analyzing the one-step strong and weak errors, additional error terms caused by the stochastic gradient arise. Bounding these terms in a dimension-independent manner would require suitable tail, variance, or moment assumptions on the gradient estimator. We leave a systematic treatment of this setting to future work.

## Impact Statement

The goal of this paper is to achieve better understandings of the Underdamped Langevin Dynamics with discretizations. As a paper of a theoretical nature, this paper is believed to have no direct social impact, which we feel must be specifically highlighted here.

## Acknowledgements

We thank the anonymous reviewers for their helpful comments. SZ, QD, XL and QG are supported in part by the National Science Foundation DMS-2323113, CPS-2312094 and DMS-2502536, as well as the Sloan Research Fellowship. The views and conclusions contained in this paper are those of the authors and should not be interpreted as representing any funding agencies.

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

# A. Detailed Description of the KL Local Framework

For the reader's convenience, we present in this section a comprehensive description of the KL local framework (Altschuler et al., 2025). Our goal is to provide details of Theorem 3.7 with greater rigor, including the specific choices of parameters that were omitted in the main text. The framework is built upon a shifted chain rule for the KL divergence, which we first state below.

**Theorem A.1** (Theorem 2.4 in Altschuler et al. 2025). *Let $X, X', Y$ be three jointly defined random variables on a standard probability space $\Omega$. Let $\mathbb{P}, \mathbb{Q}$ be two probability measures over $\Omega$, with superscripts denoting the laws of random variables under these measures. Then,*

$$\mathrm{KL}\big(\mathbb{P}^{\boldsymbol{Y}}\|\mathbb{Q}^{\boldsymbol{Y}}\big) \leq \mathrm{KL}\big(\mathbb{P}^{\boldsymbol{X}'}\|\mathbb{Q}^{\boldsymbol{X}}\big) + \inf_{\gamma \in \mathscr{C}(\mathbb{P}^{\boldsymbol{X}}, \mathbb{P}^{\boldsymbol{X}'})} \int \mathrm{KL}\big(\mathbb{P}^{\boldsymbol{Y}|\boldsymbol{X}=\mathbf{x}}\|\mathbb{Q}^{\boldsymbol{Y}|\boldsymbol{X}=\mathbf{x}'}\big)\gamma(d\mathbf{x}, d\mathbf{x}'),$$

*where $\mathscr{C}(\mathbb{P}^{\boldsymbol{X}}, \mathbb{P}^{\boldsymbol{X}'})$ is the set of couplings of $\mathbb{P}^{\boldsymbol{X}}, \mathbb{P}^{\boldsymbol{X}'}$.*

This result can be interpreted as a shifted version of the standard chain rule of KL divergence, by introducing a third auxiliary random variable $\boldsymbol{X}'$. In the special case where $\boldsymbol{X}' = \boldsymbol{X}$, the theorem reduces to the standard chain rule.

Recall the shifted process $(\boldsymbol{\psi}_n^{\mathrm{sh}})_{n=0}^N$ and the auxiliary process $(\boldsymbol{\psi}_n^{\mathrm{aux}})_{n=0}^N$ defined in Definition 3.6, the discrete-time ULD process $(\boldsymbol{\psi}_n)_{n=0}^N$ defined in (3.5), and the numerical discretization $(\boldsymbol{\psi}_n^{\mathrm{alg}})_{n=0}^N$ defined in (3.6). For any $n \leq N$, let $\nu_n^{\mathrm{aux}}$ be the distribution of the auxiliary process $\boldsymbol{\psi}_n^{\mathrm{aux}}$, $\nu_n^{\mathrm{sh}}$ be the distribution of the shifted process $\boldsymbol{\psi}_n^{\mathrm{sh}}$, $\nu_n$ be the distribution of $\boldsymbol{\psi}_n$, and $\mu_n^{\mathrm{alg}}$ be the distribution of $\boldsymbol{\psi}_n^{\mathrm{alg}}$.

We now use Theorem A.1 with the following specifications: Under $\mathbb{P}$, we let $\boldsymbol{X} \sim \mu_{N-1}^{\mathrm{alg}}$, $\boldsymbol{X}' \sim \nu_{N-1}^{\mathrm{aux}}$, $\boldsymbol{Y} \sim \mu_N^{\mathrm{alg}}$. Under $\mathbb{Q}$, we let $\boldsymbol{X} \sim \nu_{N-1}$ and $\boldsymbol{Y}|\boldsymbol{X} \sim \delta_{\boldsymbol{X}}\mathcal{P}_h$, and thus $\boldsymbol{Y} \sim \nu_N$. Then, Theorem A.1 indicates

$$\mathrm{KL}\big(\mu_N^{\mathrm{alg}}\|\nu_N\big) \leq \mathrm{KL}\big(\nu_{N-1}^{\mathrm{aux}}\|\nu_{N-1}\big) + \mathbb{E}\big[\mathrm{KL}\big(\delta_{\boldsymbol{\psi}_{N-1}^{\mathrm{alg}}}\mathcal{P}_h^{\mathrm{alg}}\big\|\delta_{\boldsymbol{\psi}_{N-1}^{\mathrm{aux}}}\mathcal{P}_h\big)\big]$$

$$\leq \underbrace{\mathrm{KL}\big(\nu_{N-1}^{\mathrm{aux}}\|\nu_{N-1}\big)}_{I_1} + O\Bigg(\underbrace{\frac{\mathbb{E}\big[\|\boldsymbol{X}_{N-1}^{\mathrm{alg}} - \boldsymbol{X}_{N-1}^{\mathrm{aux}}\|^2\big]}{\gamma h^3} + \frac{\mathbb{E}\big[\|\boldsymbol{P}_{N-1}^{\mathrm{alg}} - \boldsymbol{P}_{N-1}^{\mathrm{aux}}\|^2\big]}{\gamma h}}_{I_2} + \underbrace{\mathbb{E}\big[b^2\big(\boldsymbol{X}_{N-1}^{\mathrm{alg}}, \boldsymbol{P}_{N-1}^{\mathrm{alg}}\big)\big]}_{I_3}\Bigg),$$

where the last inequality holds due to the cross-regularity assumption (Definition 3.5) of the last step.

Among these terms, $I_1$ is the dominant term. Thus, we only discuss $I_1$ in this section. For more details regarding $I_2$ and $I_3$, we refer the readers to Altschuler et al. (2025).For $I_1$, it can be bounded iteratively using Theorem A.1 as follows: For any $n \leq N - 1$, we consider the following choices: under $\mathbb{P}$, we let $\boldsymbol{X} \sim \nu_n^{\mathrm{sh}}$, $\boldsymbol{X}' \sim \nu_n^{\mathrm{aux}}$, $\boldsymbol{Y} \sim \nu_{n+1}^{\mathrm{aux}}$. Under $\mathbb{Q}$, we let $\boldsymbol{X} \sim \nu_n$ and $\boldsymbol{Y}|\boldsymbol{X} \sim \delta_{\boldsymbol{X}}\mathcal{P}_h$, and thus $\boldsymbol{Y} \sim \nu_{n+1}$. Then, Theorem A.1 indicates

$$\mathrm{KL}\big(\nu_{n+1}^{\mathrm{aux}}\|\nu_{n+1}\big) \leq \mathrm{KL}\big(\nu_n^{\mathrm{aux}}\|\nu_n\big) + \mathbb{E}\big[\mathrm{KL}\big(\delta_{\boldsymbol{\psi}_n^{\mathrm{sh}}}\mathcal{P}_h\big\|\delta_{\boldsymbol{\psi}_n^{\mathrm{aux}}}\mathcal{P}_h\big)\big].$$

As a result, we have

$$I_1 \leq \sum_{n=0}^{N-2} \mathbb{E}\big[\mathrm{KL}\big(\delta_{\boldsymbol{\psi}_n^{\mathrm{sh}}}\mathcal{P}_h\big\|\delta_{\boldsymbol{\psi}_n^{\mathrm{aux}}}\mathcal{P}_h\big)\big]. \tag{A.1}$$

**Bounding $I_1$ using Harnack's inequality.**

To further bound the term $I_1$, we need the following inequality proved in Altschuler et al. (2025).

**Theorem A.2** (Theorem 3.2 in Altschuler et al. 2025). *Under Assumption 3.1, let $\mathcal{P}_h$ be the time-$h$ transition operator of ULD. Then, there exist parameter-dependent constants $C = C(\alpha, \beta, \gamma, h)$ and $\gamma_0 = \gamma_0(\alpha, \beta, \gamma, h)$, such that for all $q \geq 1$, and all $\mathbf{x}, \bar{\mathbf{x}}, \mathbf{p}, \bar{\mathbf{p}} \in \mathbb{R}^d$, the Rényi divergence between two processes starting from $(\mathbf{x}, \mathbf{p})$ and $(\bar{\mathbf{x}}, \bar{\mathbf{p}})$ can be upper bounded by*

$$R_q\big(\delta_{\mathbf{x},\mathbf{p}}\mathcal{P}_h\|\delta_{\bar{\mathbf{x}},\bar{\mathbf{p}}}\mathcal{P}_h\big) \leq qC\bigg\{\big\|\mathbf{x} - \bar{\mathbf{x}}\big\|^2 + \frac{1}{\gamma_0^2}\big\|\mathbf{p} - \bar{\mathbf{p}}\big\|^2\bigg\}.$$

Specifically, when $q = 1$, the Rényi divergence reduces to the KL divergence, and the upper bound becomes

$$\mathrm{KL}\big(\delta_{\mathbf{x},\mathbf{p}}\mathcal{P}_h \| \delta_{\bar{\mathbf{x}},\bar{\mathbf{p}}}\mathcal{P}_h\big) \leq C\left\{\|\mathbf{x} - \bar{\mathbf{x}}\|^2 + \frac{1}{\gamma_0^2}\|\mathbf{p} - \bar{\mathbf{p}}\|^2\right\}.$$

**Remark A.3.** We now clarify the choice of the constants appearing in this theorem. We first introduce an auxiliary constant $\omega$ whose value is defined differently in two regimes: the strongly convex setting, and the general convex setting:

$$\omega := \begin{cases} \alpha/(3\gamma) & \text{if } \alpha > 0, \\ 0 & \text{if } \alpha = 0. \end{cases}$$

Then, in the strongly convex setting, the constants $C$ and $\gamma_0$ satisfy:

$$C(\alpha, \beta, \gamma, h) \lesssim \frac{1}{\gamma}\left(\frac{\omega}{\exp(c\omega h) - 1}\right)^3 + \gamma \frac{\omega}{\exp(c\omega h) - 1}, \quad \gamma_0(\alpha, \beta, \gamma, h) \gtrsim \gamma + \frac{\mathbb{1}(h \leq 1/|\omega|)}{h}, \qquad \text{(A.2)}$$

where $c = 1/48$ is a universal constant. In the general convex setting, the inequality holds when taking the limit $\alpha \to 0$, which shows

$$C(\alpha, \beta, \gamma, h) \lesssim \frac{1}{\gamma h^3} + \frac{\gamma}{h}, \quad \gamma_0(\alpha, \beta, \gamma, h) = \gamma + \frac{1}{h}.$$

Under the assumption within Theorem 3.7, we have $h \lesssim \gamma^{-1} \wedge \gamma/\beta$. Thus, we have $h \leq 1/|\omega|$, and $\gamma_0 \gtrsim 1/h$. Moreover, we have

$$\frac{1}{\gamma}\left(\frac{\omega}{\exp(c\omega h) - 1}\right)^3 + \gamma \frac{\omega}{\exp(c\omega h) - 1} \simeq \frac{1}{\gamma h^3} + \frac{\gamma}{h} \lesssim \frac{1}{\gamma h^3},$$

where the last inequality holds due to $h \lesssim \gamma^{-1}$. Using Theorem A.2, we can further bound (A.1) as follows:

$$\begin{aligned} I_1 &\leq \sum_{n=0}^{N-2} \mathbb{E}\big[\mathrm{KL}\big(\delta_{\boldsymbol{\psi}_n^{\mathrm{sh}}}\mathcal{P}_h \big\| \delta_{\boldsymbol{\psi}_n^{\mathrm{aux}}}\mathcal{P}_h\big)\big] \\ &\lesssim \sum_{n=0}^{N-2} \frac{1}{\gamma h^3}\mathbb{E}\left[\big\|\boldsymbol{X}_n^{\mathrm{sh}} - \boldsymbol{X}_n^{\mathrm{aux}}\big\|^2 + \frac{1}{\gamma_0^2}\big\|\boldsymbol{P}_n^{\mathrm{sh}} - \boldsymbol{P}_n^{\mathrm{aux}}\big\|^2\right] \\ &= \sum_{n=0}^{N-2} \frac{1}{\gamma h}\mathbb{E}\big\|\eta_n^{\mathbf{x}}\big(\boldsymbol{X}_n^{\mathrm{alg}} - \boldsymbol{X}_n^{\mathrm{aux}}\big) + \eta_n^{\mathbf{P}}\big(\boldsymbol{P}_n^{\mathrm{alg}} - \boldsymbol{P}_n^{\mathrm{aux}}\big)\big\|^2, \end{aligned}$$

where the last equation holds due to the definition of the shifted process. Finally, we get

$$I_1 \lesssim \sum_{n=0}^{N-2} \frac{1}{\gamma h}\left((\eta_n^{\mathbf{x}})^2\mathbb{E}\big\|\boldsymbol{X}_n^{\mathrm{alg}} - \boldsymbol{X}_n^{\mathrm{aux}}\big\|^2 + (\eta_n^{\mathbf{P}})^2\mathbb{E}\big\|\boldsymbol{P}_n^{\mathrm{alg}} - \boldsymbol{P}_n^{\mathrm{aux}}\big\|^2\right).$$

Instead of directly bounding the distance between $\boldsymbol{\psi}_n^{\mathrm{alg}}$ and $\boldsymbol{\psi}_n^{\mathrm{aux}}$, we consider the distance between $\boldsymbol{\psi}_n^{\mathrm{alg}}$ and $\boldsymbol{\psi}_n^{\mathrm{sh}}$. To be more specific, define

$$\eta_t^{\mathbf{P}} := \frac{c_0\omega}{\exp(\omega(Nh - t + Ah)) - 1}, \quad \eta_t^{\mathbf{x}} := \frac{(\gamma + \eta_t^{\mathbf{P}})\eta_t^{\mathbf{P}}}{2},$$

where $c_0, A$ are absolute constants such that both $c_0$ and $A/c_0$ are sufficiently large. Then we have $\eta_t^{\mathbf{P}} \lesssim c_0/(Ah)$. The shifted parameters at step $n$ are then defined as

$$\eta_n^{\mathbf{x}} := \int_{nh}^{(n+1)h} \eta_t^{\mathbf{x}}\,\mathrm{d}t, \quad \eta_n^{\mathbf{P}} := \int_{nh}^{(n+1)h} \eta_t^{\mathbf{P}}\,\mathrm{d}t,$$

In this case, calculation indicates the distance between $\psi_n^{\mathrm{alg}}$ and $\psi_n^{\mathrm{aux}}$, and the distance between $\psi_n^{\mathrm{alg}}$ and $\psi_n^{\mathrm{sh}}$ are in the same order, i.e.,

$$(\eta_n^{\mathbf{x}})^2 \mathbb{E}\big\|\boldsymbol{X}_n^{\mathrm{alg}} - \boldsymbol{X}_n^{\mathrm{aux}}\big\|^2 + (\eta_n^{\mathbf{P}})^2 \mathbb{E}\big\|\boldsymbol{P}_n^{\mathrm{alg}} - \boldsymbol{P}_n^{\mathrm{aux}}\big\|^2 \simeq (\eta_n^{\mathbf{x}})^2 \bigg[\mathbb{E}\big\|\boldsymbol{X}_n^{\mathrm{alg}} - \boldsymbol{X}_n^{\mathrm{aux}}\big\|^2 + \frac{1}{(\gamma + \eta_{nh}^{\mathbf{P}})^2} \mathbb{E}\big\|\boldsymbol{P}_n^{\mathrm{alg}} - \boldsymbol{P}_n^{\mathrm{aux}}\big\|^2\bigg]$$
$$\simeq (\eta_n^{\mathbf{x}})^2 \bigg[\mathbb{E}\big\|\boldsymbol{X}_n^{\mathrm{alg}} - \boldsymbol{X}_n^{\mathrm{sh}}\big\|^2 + \frac{1}{(\gamma + \eta_{nh}^{\mathbf{P}})^2} \mathbb{E}\big\|\boldsymbol{P}_n^{\mathrm{alg}} - \boldsymbol{P}_n^{\mathrm{sh}}\big\|^2\bigg],$$

where the last step holds using Lemma 4.9 in Altschuler et al. (2025). Let

$$(d_n^{\mathrm{sh}})^2 = \mathbb{E}\big\|\boldsymbol{X}_n^{\mathrm{alg}} - \boldsymbol{X}_n^{\mathrm{sh}}\big\|^2 + \frac{1}{(\gamma + \eta_{nh}^{\mathbf{P}})^2} \mathbb{E}\big\|\boldsymbol{P}_n^{\mathrm{alg}} - \boldsymbol{P}_n^{\mathrm{sh}}\big\|^2.$$

Then, it can be proved that the distance satisfy a recursion inequality, with contraction factor $L_n$, defined as follows:

$$L_n = \exp\bigg(-c \int_{nh}^{(n+1)h} (\omega_+ + \eta_t^{\mathbf{P}})\, \mathrm{d}t\bigg).$$

The following lemma holds:

**Lemma A.4** (Lemma 4.3 in Altschuler et al. 2025)**.** For all $n < N - 1$, the following inequality holds:

$$(d_{n+1}^{\mathrm{sh}})^2 \leq L_n (d_n^{\mathrm{sh}})^2 + O\bigg(\frac{(\bar{\mathcal{E}}^w)^2}{(\omega_+ + \eta_{nh}^{\mathbf{P}})h} + \bigg[1 + \frac{\beta^2 h}{\gamma_{nh}^2 (\omega_+ + \eta_{nh}^{\mathbf{P}})}\bigg](\bar{\mathcal{E}}^s)^2\bigg).$$

### Avoiding $\bar{f}$ with Lipschitz assumption.

The following assumption characterizes the conditions when the expectation of $\mu[\mathcal{P}^{\mathrm{alg}}]^n$ can be replaced with that of the auxiliary process.

**Assumption A.5** (Lipschitz errors)**.** Let $\mathcal{E}^w, \mathcal{E}^s, b$ be defined in Definition 3.4. Suppose they satisfy the following Lipschitz conditions: For any $(\mathbf{x}, \mathbf{p}), (\bar{\mathbf{x}}, \bar{\mathbf{p}})$, the following inequality holds:

$$\big|\mathcal{E}^w(\mathbf{x}, \mathbf{p}) - \mathcal{E}^w(\bar{\mathbf{x}}, \bar{\mathbf{p}})\big| \leq L_{w,\mathbf{x}} \|\mathbf{x} - \bar{\mathbf{x}}\| + L_{w,\mathbf{p}} \|\mathbf{p} - \bar{\mathbf{p}}\|,$$
$$\big|\mathcal{E}^s(\mathbf{x}, \mathbf{p}) - \mathcal{E}^s(\bar{\mathbf{x}}, \bar{\mathbf{p}})\big| \leq L_{s,\mathbf{x}} \|\mathbf{x} - \bar{\mathbf{x}}\| + L_{s,\mathbf{p}} \|\mathbf{p} - \bar{\mathbf{p}}\|,$$
$$\big|b(\mathbf{x}, \mathbf{p}) - b(\bar{\mathbf{x}}, \bar{\mathbf{p}})\big| \leq L_{b,\mathbf{x}} \|\mathbf{x} - \bar{\mathbf{x}}\| + L_{b,\mathbf{p}} \|\mathbf{p} - \bar{\mathbf{p}}\|.$$

Furthermore, suppose that

$$\max\bigg\{\frac{(L_{w,\mathbf{x}} + (\gamma + \eta_{nh}^{\mathbf{P}})L_{w,\mathbf{p}})^2}{(\omega_+ + \eta_{nh}^{\mathbf{P}})(\gamma + \eta_{nh}^{\mathbf{P}})^2 h}, \bigg(1 + \frac{\beta^2 h}{(\gamma + \eta_{nh}^{\mathbf{P}})^2 (\omega_+ + \eta_{nh}^{\mathbf{P}})}\bigg)\frac{(L_{s,\mathbf{x}} + (\gamma + \eta_{nh}^{\mathbf{P}})L_{s,\mathbf{p}})^2}{(\gamma + \eta_{nh}^{\mathbf{P}})^2}\bigg\} \lesssim 1 - L_n,$$

and

$$(L_{b,\mathbf{x}} + h^{-1}L_{b,\mathbf{p}})^2 \gamma h^3 \lesssim 1$$

for sufficiently small constants.

We then have the following theorem:

**Theorem A.6** (Lemma C.2 in Altschuler et al. 2025)**.** If Assumption A.5 holds, we can further define

$$\widetilde{f} = \max_{1 \leq n \leq N-1} \|f\|_{L^2(\nu_n^{\mathrm{aux}})}$$

for $f \in \{b, \mathcal{E}^w, \mathcal{E}^s\}$. Then, Theorem 3.7 and Corollary 3.8 still hold if we replace $\bar{f}$ with $\widetilde{f}$ for $f \in \{b, \mathcal{E}^w, \mathcal{E}^s\}$. To be more specific, using the same setting and notation as Theorem 3.7, the KL-divergence between $\nu_n^{\mathrm{aux}}$ and $\nu_n$ satisfies

$$\mathrm{KL}\big(\nu_n^{\mathrm{aux}}\|\nu_n\big) \lesssim C \mathcal{W}_2^2(\mu, \nu) + A_w\bigg(\max_{1 \leq i \leq n-1} \mathbb{E}_{\nu_i^{\mathrm{aux}}}\big[(\mathcal{E}^w)^2\big]\bigg) + A_s\bigg(\max_{1 \leq i \leq n-1} \mathbb{E}_{\nu_i^{\mathrm{aux}}}\big[(\mathcal{E}^s)^2\big]\bigg).$$

Moreover, the KL-divergence between the distribution resulting from the composition of $N - 1$ steps of $\mathcal{P}^{\mathrm{alg}}$, and one step of $\widetilde{\mathcal{P}}$, and the distribution from $N$ steps of $\mathcal{P}$, can be upper bounded as follows:

$$\mathrm{KL}\big(\mu(\mathcal{P}^{\mathrm{alg}})^{N-1}\widetilde{\mathcal{P}}\|\nu\mathcal{P}^N\big) \lesssim C\mathcal{W}_2^2(\mu,\nu) + A_w\Big(\max_{1\leq i\leq N-1}\mathbb{E}_{\nu_i^{\mathrm{aux}}}\big[(\mathcal{E}^w)^2\big]\Big)$$

$$+ A_s\Big(\max_{1\leq i\leq N-1}\mathbb{E}_{\nu_i^{\mathrm{aux}}}\big[(\mathcal{E}^s)^2\big]\Big) + \max_{1\leq i\leq n-1}\mathbb{E}_{\nu_i^{\mathrm{aux}}}\big[b^2\big].$$

**Specifying the constants in Theorem 3.7.**

In this section, we specify the value of the constants $C$, $A_w$ and $A_s$ in Theorem 3.7. First, $C = C(\alpha, \beta, \gamma, Nh)$ defined in (A.2). Second, in different case, the values of $A_w$ and $A_s$ are

- Strongly convex and high friction: $(\alpha > 0, \gamma = \sqrt{32\beta})$

$$A_w = \frac{1}{\alpha h^2}, A_s = \frac{1}{\beta^{1/2}h}\log\frac{3\gamma}{\alpha h}.$$

- Generally convex and high friction: $(\alpha = 0, \gamma = \sqrt{32\beta})$

$$A_w = \frac{N}{\beta^{1/2}h}, A_s = \frac{1}{\beta^{1/2}h}\log N + \beta^{1/2}Nh.$$

# B. Analysis of Weak and Strong Errors

In this section, we analyze the weak and strong errors of the standard ULMC (3.2) and the randomized midpoint discretization (3.4). Throughout this section, we assume that the initial condition is $(\mathbf{x}, \mathbf{p})$, and assume the synchronous coupling of all Markov chains $\{(\boldsymbol{X}_h, \boldsymbol{P}_h)\}_{n\in\mathbb{N}}$, $\{(\boldsymbol{X}_{nh}^{\mathrm{ULMC}}, \boldsymbol{P}_{nh}^{\mathrm{ULMC}})\}_{n\in\mathbb{N}}$, and $\{(\boldsymbol{X}_{nh}^{\mathrm{RM}}, \boldsymbol{P}_{nh}^{\mathrm{RM}})\}$, i.e., they are driven by the same Brownian motion.

## B.1. Standard ULMC

We first investigate the one-step error vectors, i.e., $\boldsymbol{X}_h^{\mathrm{ULMC}} - \boldsymbol{X}_h$ and $\boldsymbol{P}_h^{\mathrm{ULMC}} - \boldsymbol{P}_h$:

$$\boldsymbol{X}_h^{\mathrm{ULMC}} - \boldsymbol{X}_h = \int_0^h \frac{1-e^{-\gamma(h-t)}}{\gamma}[\nabla V(\mathbf{x}) - \nabla V(\boldsymbol{X}_t)]\,\mathrm{d}t, \tag{B.1}$$

$$\boldsymbol{P}_h^{\mathrm{ULMC}} - \boldsymbol{P}_h = \int_0^h e^{-\gamma(h-t)}[\nabla V(\mathbf{x}) - \nabla V(\boldsymbol{X}_t)]\,\mathrm{d}t. \tag{B.2}$$

Therefore, the term $\nabla V(\boldsymbol{X}_h) - \nabla V(\mathbf{x})$ is of central interest in order to bound the weak and strong errors. We thus present the following lemma:

**Lemma B.1.** Suppose that Assumption 3.1 holds, and the step size satisfies $h \lesssim 1/\sqrt{\beta}$. Then for any $t \in [0, h]$, the following inequality holds:

$$\|\nabla V(\boldsymbol{X}_t) - \nabla V(\mathbf{x})\|_{L_2} \lesssim \beta^{1/2}h\|\mathbf{p}\|_{\mathbf{H}} + \gamma^{1/2}\beta^{1/2}h^{3/2}\sqrt{\mathrm{tr}(\mathbf{H})} + \beta h^2\|\nabla V(\mathbf{x})\|.$$

The proof of Lemma B.1 is given in Appendix D.1. We now provide the proof of Lemma 4.1:

*Proof of Lemma 4.1.* Starting from (B.1) and (B.2), we bound the weak and strong errors as follows:

**Weak error.** The position error in the weak error satisfies

$$\|\mathbb{E}\boldsymbol{X}_h^{\mathrm{ULMC}} - \mathbb{E}\boldsymbol{X}_h\| = \left\|\int_0^h \frac{1-e^{-\gamma(h-t)}}{\gamma}\mathbb{E}[\nabla V(\boldsymbol{X}_t) - \nabla V(\mathbf{x})]\,\mathrm{d}t\right\|$$

$$\leq \int_0^h \frac{1-e^{-\gamma(h-t)}}{\gamma}\big\|\mathbb{E}[\nabla V(\boldsymbol{X}_t) - \nabla V(\mathbf{x})]\big\|\,\mathrm{d}t$$

$$\leq \int_0^h \frac{1 - e^{-\gamma(h-t)}}{\gamma} \, \mathrm{d}t \cdot \sup_{t \in [0,h]} \left\| \mathbb{E}[\nabla V(\boldsymbol{X}_t) - \nabla V(\mathbf{x})] \right\|, \tag{B.3}$$

where the first inequality holds due to Jensen's inequality ($\| \cdot \|$ is a convex function), and the second inequality holds due to Hölder's inequality. We first note that

$$\int_0^h \frac{1 - e^{-\gamma(h-t)}}{\gamma} \, \mathrm{d}t \leq \int_0^h (h-t) \, \mathrm{d}t \lesssim h^2, \tag{B.4}$$

where the first inequality holds because $1 - e^{-z} \leq z$. We also note that for any $t \in [0, h]$,

$$\left\| \mathbb{E}[\nabla V(\boldsymbol{X}_t) - \nabla V(\mathbf{x})] \right\| = \sqrt{\left\| \mathbb{E}[\nabla V(\boldsymbol{X}_t) - \nabla V(\mathbf{x})] \right\|^2} \leq \sqrt{\mathbb{E}\big[\| \nabla V(\boldsymbol{X}_t) - \nabla V(\mathbf{x}) \|^2 \big]} = \| \nabla V(\boldsymbol{X}_t) - \nabla V(\mathbf{x}) \|_{L_2}$$
$$\lesssim \beta^{1/2} h \|\mathbf{p}\|_{\mathbf{H}} + \gamma^{1/2} \beta^{1/2} h^{3/2} \sqrt{\mathrm{tr}(\mathbf{H})} + \beta h^2 \|\nabla V(\mathbf{x})\|, \tag{B.5}$$

where the first inequality holds due to the Jensen's inequality ($\| \cdot \|^2$ is a convex function), and the second inequality holds due to Lemma B.1. Plugging (B.4) and (B.5) into (B.3), we obtain

$$\| \mathbb{E}\boldsymbol{X}_h^{\mathrm{ULMC}} - \mathbb{E}\boldsymbol{X}_h \| \leq \beta^{1/2} h^3 \|\mathbf{p}\|_{\mathbf{H}} + \gamma^{1/2} \beta^{1/2} h^{7/2} \sqrt{\mathrm{tr}(\mathbf{H})} + \beta h^4 \|\nabla V(\mathbf{x})\|.$$

The momentum error in the weak error satisfies

$$\begin{aligned}
\| \mathbb{E}\boldsymbol{P}_h^{\mathrm{ULMC}} - \mathbb{E}\boldsymbol{P}_h \| &= \left\| \int_0^h e^{-\gamma(h-t)} \mathbb{E}[\nabla V(\boldsymbol{X}_t) - \nabla V(\mathbf{x})] \, \mathrm{d}t \right\| \\
&\leq \int_0^h e^{-\gamma(h-t)} \left\| \mathbb{E}[\nabla V(\boldsymbol{X}_t) - \nabla V(\mathbf{x})] \right\| \mathrm{d}t \\
&\leq h \sup_{t \in [0,h]} \left\| \mathbb{E}[\nabla V(\boldsymbol{X}_t) - \nabla V(\mathbf{x})] \right\| \\
&\lesssim \beta^{1/2} h^2 \|\mathbf{p}\|_{\mathbf{H}} + \gamma^{1/2} \beta^{1/2} h^{5/2} \sqrt{\mathrm{tr}(\mathbf{H})} + \beta h^3 \|\nabla V(\mathbf{x})\|,
\end{aligned}$$

where the first inequality holds due to Jensen's inequality ($\| \cdot \|$ is a convex function), the second inequality holds because $e^{-z} \leq 1$, and the last inequality holds due to (B.5). Therefore, the weak error of standard ULMC satisfies

$$\mathcal{E}^w(\mathbf{x}, \mathbf{p}) \lesssim \beta^{1/2} h^2 \|\mathbf{p}\|_{\mathbf{H}} + \gamma^{1/2} \beta^{1/2} h^{5/2} \sqrt{\mathrm{tr}(\mathbf{H})} + \beta h^3 \|\nabla V(\mathbf{x})\|.$$

**Strong error.** The position error in the strong error satisfies

$$\begin{aligned}
\| \boldsymbol{X}_h^{\mathrm{ULMC}} - \boldsymbol{X}_h \|_{L_2} &= \left\| \int_0^h \frac{1 - e^{-\gamma(h-t)}}{\gamma} [\nabla V(\boldsymbol{X}_t) - \nabla V(\mathbf{x})] \, \mathrm{d}t \right\|_{L_2} \\
&\leq \int_0^h \frac{1 - e^{-\gamma(h-t)}}{\gamma} \| \nabla V(\boldsymbol{X}_t) - \nabla V(\mathbf{x}) \|_{L_2} \, \mathrm{d}t \\
&\leq \int_0^h \frac{1 - e^{-\gamma(h-t)}}{\gamma} \, \mathrm{d}t \cdot \sup_{t \in [0,h]} \| \nabla V(\boldsymbol{X}_t) - \nabla V(\mathbf{x}) \|_{L_2} \\
&\lesssim \beta^{1/2} h^3 \|\mathbf{p}\|_{\mathbf{H}} + \gamma^{1/2} \beta^{1/2} h^{7/2} \sqrt{\mathrm{tr}(\mathbf{H})} + \beta h^4 \|\nabla V(\mathbf{x})\|, 
\end{aligned} \tag{B.6}$$

where the first inequality holds due to the Jensen's inequality ($\| \cdot \|_{L_2}$ is a convex function), the second inequality holds due to the Jensen's inequality, and the last inequality holds due to (B.4) and Lemma B.1. The momentum error in the strong error satisfies

$$\begin{aligned}
\| \boldsymbol{P}_h^{\mathrm{ULMC}} - \boldsymbol{P}_h \|_{L_2} &= \left\| \int_0^h e^{-\gamma(h-t)} [\nabla V(\boldsymbol{X}_t) - \nabla V(\mathbf{x})] \, \mathrm{d}t \right\|_{L_2} \\
&\leq \int_0^h e^{-\gamma(h-t)} \| \nabla V(\boldsymbol{X}_t) - \nabla V(\mathbf{x}) \|_{L_2} \, \mathrm{d}t
\end{aligned}$$

$$\leq h \sup_{t \in [0,h]} \|\nabla V(\boldsymbol{X}_t) - \nabla V(\mathbf{x})\|_{L_2}$$
$$\lesssim \beta^{1/2} h^2 \|\mathbf{p}\|_{\mathbf{H}} + \gamma^{1/2} \beta^{1/2} h^{5/2} \sqrt{\mathrm{tr}(\mathbf{H})} + \beta h^3 \|\nabla V(\mathbf{x})\|,$$

where the first inequality holds due to the Jensen's inequality ($\|\cdot\|_{L_2}$ is a convex function), the second inequality holds because $e^{-z} \leq 1$, and the last inequality holds due to Lemma B.1. Therefore, the strong error of standard ULMC satisfies

$$\mathcal{E}^s(\mathbf{x}, \mathbf{p}) \lesssim \beta^{1/2} h^2 \|\mathbf{p}\|_{\mathbf{H}} + \gamma^{1/2} \beta^{1/2} h^{5/2} \sqrt{\mathrm{tr}(\mathbf{H})} + \beta h^3 \|\nabla V(\mathbf{x})\|.$$

$\square$

## B.2. Randomized Midpoint Discretization

Similar to the analysis of the standard ULMC, for the randomized midpoint discretization, we consider the error vectors $\boldsymbol{X}_h^{\mathrm{RM}} - \boldsymbol{X}_h$ and $\boldsymbol{P}_h^{\mathrm{RM}} - \boldsymbol{P}_h$:

$$
\begin{aligned}
\boldsymbol{X}_h^{\mathrm{RM}} - \boldsymbol{X}_h &= \int_0^h \frac{1 - e^{-\gamma(h-t)}}{\gamma} [\nabla V(\widehat{\boldsymbol{X}}_{uh}^+) - \nabla V(\boldsymbol{X}_t)] \, \mathrm{d}t \\
&= \int_0^h \frac{1 - e^{-\gamma(h-t)}}{\gamma} \, \mathrm{d}t \cdot {\color{blue}[\nabla V(\widehat{\boldsymbol{X}}_{uh}^+) - \nabla V(\boldsymbol{X}_{uh})]} + \int_0^h \frac{1 - e^{-\gamma(h-t)}}{\gamma} {\color{red}[\nabla V(\boldsymbol{X}_{uh}) - \nabla V(\boldsymbol{X}_t)]} \, \mathrm{d}t, \quad \text{(B.7)}
\end{aligned}
$$

$$
\begin{aligned}
\boldsymbol{P}_h^{\mathrm{RM}} - \boldsymbol{P}_h &= \int_0^h e^{-\gamma(h-t)} [\nabla V(\widehat{\boldsymbol{X}}_{vh}^{++}) - \nabla V(\boldsymbol{X}_t)] \, \mathrm{d}t \\
&= \int_0^h e^{-\gamma(h-t)} \, \mathrm{d}t \cdot {\color{blue}[\nabla V(\widehat{\boldsymbol{X}}_{vh}^{++}) - \nabla V(\boldsymbol{X}_{vh})]} + \int_0^h e^{-\gamma(h-t)} {\color{red}[\nabla V(\boldsymbol{X}_{vh}) - \nabla V(\boldsymbol{X}_t)]} \, \mathrm{d}t. \quad \text{(B.8)}
\end{aligned}
$$

We also recall the following properties of the randomized midpoint:

$$\frac{1}{\gamma}\left(h - \frac{1 - e^{-\gamma h}}{\gamma}\right) \mathbb{E}_u[\nabla V(\widehat{\boldsymbol{X}}_{uh}^+)] = \int_0^h \frac{1 - e^{-\gamma(h-t)}}{\gamma} \nabla V(\widehat{\boldsymbol{X}}_t^+) \, \mathrm{d}t; \quad \text{(B.9)}$$

$$\frac{1 - e^{-\gamma h}}{\gamma} \mathbb{E}_v[\nabla V(\widehat{\boldsymbol{X}}_{vh}^{++})] = \int_0^h e^{-\gamma(h-t)} \nabla V(\widehat{\boldsymbol{X}}_t^{++}) \, \mathrm{d}t \quad \text{(B.10)}$$

Thus, the expectation of the error vectors satisfy

$$\mathbb{E}[\boldsymbol{X}_h^{\mathrm{RM}} - \boldsymbol{X}_h] = \int_0^h \frac{1 - e^{-\gamma(h-t)}}{\gamma} \mathbb{E}[\nabla V(\widehat{\boldsymbol{X}}_t^+) - \nabla V(\boldsymbol{X}_t)] \, \mathrm{d}t, \quad \text{(B.11)}$$

$$\mathbb{E}[\boldsymbol{P}_h^{\mathrm{RM}} - \boldsymbol{P}_h] = \int_0^h e^{-\gamma(h-t)} \mathbb{E}[\nabla V(\widehat{\boldsymbol{X}}_t^{++}) - \nabla V(\boldsymbol{X}_t)] \, \mathrm{d}t. \quad \text{(B.12)}$$

The terms in red are the differences of $\nabla V$ at times of the **ground truth** ULD dynamics, so they can be bounded using Lemma B.1. The terms in blue are caused by the error of standard ULMC sequence, which are characterized with the following lemma:

**Lemma B.2.** Suppose that Assumption 3.1 holds, and the step size satisfies $h \lesssim 1/\sqrt{\beta}$. Then for any $t \in [0, h]$, the following inequality holds:

$$\|\nabla V(\boldsymbol{X}_t^{\mathrm{ULMC}}) - \nabla V(\boldsymbol{X}_t)\|_{L_2} \leq \beta^{3/2} h^3 \|\mathbf{p}\|_{\mathbf{H}} + \gamma^{1/2} \beta^{3/2} h^{7/2} \sqrt{\mathrm{tr}(\mathbf{H})} + \beta^2 h^4 \|\nabla V(\mathbf{x})\|,$$

The proof of Lemma B.2 is given in Appendix D.2. We now provide the proof of Lemma 5.1.

*Proof of Lemma 5.1.* We characterize the weak error with (B.11) and (B.12), and the strong error with (B.7) and (B.8):

**Weak error.** Based on (B.11), the weak error of the position vector satisfies

$$\|\mathbb{E}[\boldsymbol{X}_h^{\mathrm{RM}} - \boldsymbol{X}_h]\| = \left\| \int_0^h \frac{1 - e^{-\gamma(h-t)}}{\gamma} \mathbb{E}[\nabla V(\widehat{\boldsymbol{X}}_t^+) - \nabla V(\boldsymbol{X}_t)] \, \mathrm{d}t \right\|$$

$$\leq \int_0^h \frac{1 - e^{-\gamma(h-t)}}{\gamma} \left\| \mathbb{E}[\nabla V(\widehat{\boldsymbol{X}}_t^+) - \nabla V(\boldsymbol{X}_t)] \right\| \, \mathrm{d}t$$

$$\leq \int_0^h \frac{1 - e^{-\gamma(h-t)}}{\gamma} \, \mathrm{d}t \cdot \sup_{t \in [0,h]} \left\| \mathbb{E}[\nabla V(\widehat{\boldsymbol{X}}_t^+) - \nabla V(\boldsymbol{X}_t)] \right\|$$

$$\leq \int_0^h \frac{1 - e^{-\gamma(h-t)}}{\gamma} \, \mathrm{d}t \cdot \sup_{t \in [0,h]} \left\| \nabla V(\widehat{\boldsymbol{X}}_t^+) - \nabla V(\boldsymbol{X}_t) \right\|_{L_2}$$

$$\leq \beta^{3/2} h^5 \|\mathbf{p}\|_{\mathbf{H}} + \gamma^{1/2} \beta^{3/2} h^{11/2} \sqrt{\mathrm{tr}(\mathbf{H})} + \beta^2 h^6 \|\nabla V(\mathbf{x})\|,$$

where the first inequality holds due to the Jensen's inequality ($\|\cdot\|$ is a convex function), the second inequality holds due to the Hölder inequality, the third inequality holds due to Jensen's inequality ($\|\cdot\|^2$ is a convex function, similar to (B.5)), and the last inequality holds due to (B.4) and Lemma B.2. Based on (B.12), the weak error of the momentum vector satisfies

$$\left\| \mathbb{E}[\boldsymbol{P}_h^{\mathrm{RM}} - \boldsymbol{P}_h] \right\| = \left\| \int_0^h e^{-\gamma(h-t)} \mathbb{E}[\nabla V(\widehat{\boldsymbol{X}}_t^{++}) - \nabla V(\boldsymbol{X}_t)] \, \mathrm{d}t \right\|$$

$$\leq \int_0^h e^{-\gamma(h-t)} \left\| \mathbb{E}[\nabla V(\widehat{\boldsymbol{X}}_t^{++}) - \nabla V(\boldsymbol{X}_t)] \right\| \, \mathrm{d}t$$

$$\leq \int_0^h e^{-\gamma(h-t)} \, \mathrm{d}t \cdot \sup_{t \in [0,h]} \left\| \mathbb{E}[\nabla V(\widehat{\boldsymbol{X}}_t^{++}) - \nabla V(\boldsymbol{X}_t)] \right\|$$

$$\leq \int_0^h e^{-\gamma(h-t)} \, \mathrm{d}t \cdot \sup_{t \in [0,h]} \left\| \nabla V(\widehat{\boldsymbol{X}}_t^{++}) - \nabla V(\boldsymbol{X}_t) \right\|$$

$$\lesssim \beta^{3/2} h^4 \|\mathbf{p}\|_{\mathbf{H}} + \gamma^{1/2} \beta^{3/2} h^{9/2} \sqrt{\mathrm{tr}(\mathbf{H})} + \beta^2 h^5 \|\nabla V(\mathbf{x})\|,$$

where the first inequality holds due to the Jensen's inequality ($\|\cdot\|$ is a convex function), the second inequality holds due to the Hölder inequality, the third inequality holds due to Jensen's inequality ($\|\cdot\|^2$ is a convex function, similar to (B.5)), and the last inequality holds because $e^{-z} \leq 1$ and Lemma B.2. Therefore, the weak error of the randomized midpoint discretization satisfies

$$\mathcal{E}^w(\mathbf{x}, \mathbf{p}) \lesssim \beta^{3/2} h^4 \|\mathbf{p}\|_{\mathbf{H}} + \gamma^{1/2} \beta^{3/2} h^{9/2} \sqrt{\mathrm{tr}(\mathbf{H})} + \beta^2 h^5 \|\nabla V(\mathbf{x})\|.$$

**Strong error.** Based on (B.7), the strong error of the position vector satisfies

$$\|\boldsymbol{X}_h^{\mathrm{RM}} - \boldsymbol{X}_h\|_{L_2}$$
$$\leq \underbrace{\int_0^h \frac{1 - e^{-\gamma(h-t)}}{\gamma} \, \mathrm{d}t \cdot \left\| \nabla V(\widehat{\boldsymbol{X}}_{uh}^+) - \nabla V(\boldsymbol{X}_{uh}) \right\|_{L_2}}_{I_1} + \underbrace{\left\| \int_0^h \frac{1 - e^{-\gamma(h-t)}}{\gamma} [\nabla V(\boldsymbol{X}_{uh}) - \nabla V(\boldsymbol{X}_t)] \, \mathrm{d}t \right\|_{L_2}}_{I_2}, \quad \text{(B.13)}$$

where the inequality holds due to the triangle inequality. The term $I_1$ satisfies

$$I_1 \leq h^2 \cdot \sup_{t \in [0,h]} \left\| \nabla V(\widehat{\boldsymbol{X}}_t^+) - \nabla V(\boldsymbol{X}_{uh}) \right\|_{L_2} \lesssim \beta^{3/2} h^5 \|\mathbf{p}\|_{\mathbf{H}} + \gamma^{1/2} \beta^{3/2} h^{11/2} \sqrt{\mathrm{tr}(\mathbf{H})} + \beta^2 h^6 \|\nabla V(\mathbf{x})\|, \quad \text{(B.14)}$$

where the first inequality holds due to (B.4), and the second inequality holds due to Lemma B.2. The term $I_2$ satisfies

$$I_2 \leq \int_0^h \frac{1 - e^{-\gamma(h-t)}}{\gamma} \left\| \nabla V(\boldsymbol{X}_{uh}) - \nabla V(\boldsymbol{X}_t) \right\|_{L_2} \, \mathrm{d}t$$

$$\leq \int_0^h \frac{1 - e^{-\gamma(h-t)}}{\gamma} \left[ \left\| \nabla V(\boldsymbol{X}_{uh}) - \nabla V(\mathbf{x}) \right\|_{L_2} + \left\| \nabla V(\mathbf{x}) - \nabla V(\boldsymbol{X}_t) \right\|_{L_2} \right] \, \mathrm{d}t$$

$$\lesssim \int_0^h \frac{1 - e^{-\gamma(h-t)}}{\gamma} \, \mathrm{d}t \cdot \sup_{t \in [0,h]} \left\| \nabla V(\boldsymbol{X}_t) - \nabla V(\mathbf{x}) \right\|_{L_2}$$

$$\lesssim \beta^{1/2} h^3 \|\mathbf{p}\|_{\mathbf{H}} + \gamma^{1/2} \beta^{1/2} h^{7/2} \sqrt{\mathrm{tr}(\mathbf{H})} + \beta h^4 \|\nabla V(\mathbf{x})\|, \quad \text{(B.15)}$$

where the first inequality holds due to the Jensen's inequality ($\| \cdot \|_{L_2}$ is a convex function), the second inequality holds due to the triangle inequality, the third inequality holds due to the Hölder inequality, and the last inequality holds due to (B.4) and Lemma B.1. Since $h \lesssim 1/\sqrt{\beta}$, we have $I_1 \lesssim I_2$, so plugging (B.14) and (B.15) into (B.13), we have

$$\|\boldsymbol{X}_h^{\mathrm{RM}} - \boldsymbol{X}_h\|_{L_2} \lesssim \beta^{1/2} h^3 \|\mathbf{p}\|_{\mathbf{H}} + \gamma^{1/2} \beta^{1/2} h^{7/2} \sqrt{\mathrm{tr}(\mathbf{H})} + \beta h^4 \|\nabla V(\mathbf{x})\|.$$

Based on (B.8), the strong error of the momentum vector satisfies

$$\|\boldsymbol{P}_h^{\mathrm{RM}} - \boldsymbol{P}_h\|_{L_2} \leq \underbrace{\int_0^h e^{-\gamma(h-t)}\,\mathrm{d}t \cdot \left\|\nabla V(\widehat{\boldsymbol{X}}_{vh}^{++}) - \nabla V(\boldsymbol{X}_{vh})\right\|_{L_2}}_{J_1} + \underbrace{\left\|\int_0^h e^{-\gamma(h-t)}[\nabla V(\boldsymbol{X}_{vh}) - \nabla V(\boldsymbol{X}_t)]\,\mathrm{d}t\right\|_{L_2}}_{J_2},$$

(B.16)

where the inequality holds due to the triangle inequality. The term $J_1$ satisfies

$$J_1 \leq h \cdot \sup_{t \in [0,h]} \left\|\nabla V(\widehat{\boldsymbol{X}}_t^{++}) - \nabla V(\boldsymbol{X}_t)\right\|_{L_2} \lesssim \beta^{3/2} h^4 \|\mathbf{p}\|_{\mathbf{H}} + \gamma^{1/2} \beta^{3/2} h^{9/2} \sqrt{\mathrm{tr}(\mathbf{H})} + \beta^2 h^5 \|\nabla V(\mathbf{x})\|, \quad \text{(B.17)}$$

where the first inequality holds because $e^{-z} \leq 1$, and the second inequality holds due to Lemma B.2. The term $J_2$ satisfies

$$\begin{aligned}
J_2 &\leq \int_0^h e^{-\gamma(h-t)} \left\|\nabla V(\boldsymbol{X}_{vh}) - \nabla V(\boldsymbol{X}_t)\right\|_{L_2}\,\mathrm{d}t \\
&\leq \int_0^h e^{-\gamma(h-t)} \left[\left\|\nabla V(\boldsymbol{X}_{vh}) - \nabla V(\mathbf{x})\right\|_{L_2} + \left\|\nabla V(\mathbf{x}) - \nabla V(\boldsymbol{X}_t)\right\|_{L_2}\right]\,\mathrm{d}t \\
&\lesssim \int_0^h e^{-\gamma(h-t)}\,\mathrm{d}t \cdot \sup_{t \in [0,h]} \left\|\nabla V(\boldsymbol{X}_t) - \nabla V(\mathbf{x})\right\|_{L_2} \\
&\lesssim \beta^{1/2} h^2 \|\mathbf{p}\|_{\mathbf{H}} + \gamma^{1/2} \beta^{1/2} h^{5/2} \sqrt{\mathrm{tr}(\mathbf{H})} + \beta h^3 \|\nabla V(\mathbf{x})\|,
\end{aligned}$$

(B.18)

where the first inequality holds due to the Jensen's inequality ($\| \cdot \|$ is a convex function), the second inequality holds due to the triangle inequality, the third inequality holds due to the Hölder inequality, and the last inequality holds due to $e^{-z} \leq 1$ and Lemma B.1. Since $h \lesssim 1/\sqrt{\beta}$, we have $J_1 \lesssim J_2$, so plugging (B.17) and (B.18) into (B.16), we have

$$\|\boldsymbol{P}_h^{\mathrm{RM}} - \boldsymbol{P}_h\|_{L_2} \lesssim \beta^{1/2} h^2 \|\mathbf{p}\|_{\mathbf{H}} + \gamma^{1/2} \beta^{1/2} h^{5/2} \sqrt{\mathrm{tr}(\mathbf{H})} + \beta h^3 \|\nabla V(\mathbf{x})\|.$$

Therefore, the strong error of the randomized midpoint discretization satisfies

$$\mathcal{E}^s(\mathbf{x}, \mathbf{p}) \lesssim \beta^{1/2} h^2 \|\mathbf{p}\|_{\mathbf{H}} + \gamma^{1/2} \beta^{1/2} h^{5/2} \sqrt{\mathrm{tr}(\mathbf{H})} + \beta h^3 \|\nabla V(\mathbf{x})\|.$$

$\square$

# C. Analysis of Cross Regularity Condition

In this section, we are going to prove the dimension-free version of the cross-regularity condition for the standard ULMC, similar to Lemma 4.2 in Altschuler et al. (2025). We first define the Rényi divergence as follows:

**Definition C.1.** Consider two measures $\mu$ and $\nu$, $q > 1$, $R_q$ is the Rényi divergence, defined by:

$$R_q(\mu\|\nu) = \frac{1}{q-1} \log \int \left(\frac{\mathrm{d}\mu}{\mathrm{d}\nu}\right)^q \mathrm{d}\nu.$$

In the limit of $q \to 1^+$, the Rényi divergence will reduce to KL divergence:

$$R_1(\mu\|\nu) = \int \left(\frac{\mathrm{d}\mu}{\mathrm{d}\nu} \log \frac{\mathrm{d}\mu}{\mathrm{d}\nu}\right)\mathrm{d}\nu = \mathbb{E}_\mu\left[\log \frac{\mathrm{d}\mu}{\mathrm{d}\nu}\right].$$

Here are the basic useful propositions of Rényi divergence.

**Lemma C.2** (Data processing inequality)**.** Let $\mu, \nu$ be probability measures on a measurable space $(\Omega, \mathcal{F})$, and let $T : \Omega \to E$ be any measurable map. Then for any Rényi order $q \in [1, \infty]$,

$$R_q(T_\# \mu \,\|\, T_\# \nu) \;\leq\; R_q(\mu \,\|\, \nu).$$

**Lemma C.3** (Weak triangle inequality for Rényi divergence)**.** For any probability measures $\mu, \nu, \pi$, any Rényi order $q \in [1, \infty)$, and any relaxation parameter $\lambda \in (0, 1)$, it holds that

$$R_q(\mu \,\|\, \pi) \;\leq\; \frac{q - \lambda}{q - 1} R_{q/\lambda}(\mu \,\|\, \nu) \;+\; R_{(q-\lambda)/(1-\lambda)}(\nu \,\|\, \pi). \tag{C.1}$$

In particular, by setting $\lambda = 1 - \varepsilon$ and $q = 1 + \varepsilon$ and letting $\varepsilon \downarrow 0$, Proposition C.3 yields the classical bound for the Kullback–Leibler divergence,

$$\mathrm{KL}(\mu\|\pi) \leq 2\mathrm{KL}(\mu\|\nu) + \log\left(1 + \chi^2(\nu\|\pi)\right). \tag{C.2}$$

We then present the Girsanov Theorem:

**Theorem C.4** (Girsanov)**.** Let $(\boldsymbol{B}_t)_{t \in [0,T]}$ be a standard Brownian motion under the Wiener measure $\mathbb{W}$ and let $(\boldsymbol{Y}_t)_{t \in [0,T]}$ be a progressive process with $\mathbb{E}^{\mathbb{W}} \int_0^T \|\boldsymbol{Y}_s\|^2 ds < \infty$. Let $M_t := \int_0^t \langle \boldsymbol{Y}_s, \mathrm{d}\boldsymbol{B}_s \rangle$ for $t \in [0, T]$ and let $[\boldsymbol{M}, \boldsymbol{M}] = \int_0^\cdot \|\boldsymbol{Y}_s\|^2 \, \mathrm{d}s$ denote the quadratic variation. Define the exponential martingale

$$\mathcal{E}(\boldsymbol{M}) := \exp\left(\boldsymbol{M} - \frac{1}{2}[\boldsymbol{M}, \boldsymbol{M}]\right).$$

Assume that $\mathcal{E}(\boldsymbol{M})$ is a $\mathbb{W}$-martingale and define the measure $\mathbb{Q}$ on path space via

$$\frac{\mathrm{d}\mathbb{Q}}{\mathrm{d}\mathbb{W}} = \mathcal{E}(\boldsymbol{M})_T.$$

Then, under $\mathbb{Q}$,

$$t \mapsto \widetilde{\boldsymbol{B}}_t := \boldsymbol{B}_t - [\boldsymbol{B}, \boldsymbol{M}]_t = \boldsymbol{B}_t - \int_0^t \boldsymbol{Y}_s ds$$

is a standard Brownian motion.

**Lemma C.5.** Suppose $(\boldsymbol{X}_t, \boldsymbol{P}_t)$ satisfies the following SDE for a function $\mathbf{b} : \mathbb{R}^d \times \mathbb{R}^d \to \mathbb{R}^d$:

$$\begin{cases} \mathrm{d}\boldsymbol{X}_t = \boldsymbol{P}_t \, \mathrm{d}t, \\ \mathrm{d}\boldsymbol{P}_t = \mathbf{b}(\boldsymbol{X}_t, \boldsymbol{P}_t) \, \mathrm{d}t + \sqrt{2\gamma} \, \mathrm{d}\boldsymbol{B}_t, \end{cases}$$

Another process $(\boldsymbol{X}'_t, \boldsymbol{P}'_t)$ satisfies the following SDE for a function $\mathbf{b}' : \mathbb{R}^d \times \mathbb{R}^d \to \mathbb{R}^d$:

$$\begin{cases} \mathrm{d}\boldsymbol{X}'_t = \boldsymbol{P}'_t \, \mathrm{d}t, \\ \mathrm{d}\boldsymbol{P}'_t = \mathbf{b}'(\boldsymbol{X}'_t, \boldsymbol{P}'_t) \, \mathrm{d}t + \sqrt{2\gamma} \, \mathrm{d}\boldsymbol{B}_t, \end{cases}$$

Moreover, define

$$\mathbb{P} := \mathrm{Law}\big((\boldsymbol{X}_h, \boldsymbol{P}_h)\big), \qquad \mathbb{P}' := \mathrm{Law}\big((\boldsymbol{X}'_h, \boldsymbol{P}'_h)\big).$$

The KL divergence between $\mathbb{P}$ and $\mathbb{P}'$ can be bounded as follows:

$$\mathrm{KL}\big(\mathbb{P}\|\mathbb{P}'\big) \leq \frac{1}{4\gamma}\mathbb{E}\left[\int_0^h \|\Delta(\boldsymbol{X}'_s, \boldsymbol{P}'_s)\|^2 \, \mathrm{d}s\right]$$

where $\boldsymbol{\Delta}(\mathbf{x}, \mathbf{p}) = \mathbf{b}(\mathbf{x}, \mathbf{p}) - \mathbf{b}'(\mathbf{x}, \mathbf{p})$.

*Proof of Lemma C.5.* Using Theorem C.4 with $Y_t = -\Delta(X_t, P_t)/\sqrt{2\gamma}$, then $M_t = -\int_0^t \langle \Delta(X_s, P_s), \mathrm{d}B_s \rangle / \sqrt{2\gamma}$ and $[M, M]_t = \int_0^t \|\Delta(X_s, P_s)\|^2 \, \mathrm{d}s/(2\gamma)$. Thus, suppose $\mathbb{Q}_h$ is a distribution on the path space up to time $h$, which satisfies:

$$\frac{\mathrm{d}\mathbb{Q}_h}{\mathrm{d}\mathbb{W}_h} = \mathcal{E}(M)_h := \exp\Big(M_h - \frac{1}{2}[M, M]_h\Big).$$

Then we have under $\mathbb{Q}_h$, $\mathrm{d}\widetilde{B}_t := \mathrm{d}B_t + \Delta(X_t, P_t)\,\mathrm{d}t/\sqrt{2\gamma}$ is a standard Brownian motion. Note that the equation of $(X_t, P_t)$ satisfies:

$$\begin{cases} \mathrm{d}X_t = P_t \, \mathrm{d}t, \\ \mathrm{d}P_t = \mathbf{b}'(X_t, P_t) \, \mathrm{d}t + \sqrt{2\gamma}\big[ [\Delta(X_t, P_t)/\sqrt{2\gamma}] \, \mathrm{d}t + \mathrm{d}B_t \big]. \end{cases}$$

Thus, under $\widetilde{B}_t$, $(X_t, P_t)$ satisfies the same evolution equation as $(X_t', P_t')$. Using the data-processing inequality (Lemma C.2),

$$\begin{aligned} \mathrm{KL}(\mathbb{P}\|\mathbb{P}') &\leq \mathrm{KL}(\mathbb{Q}_h\|\mathbb{W}_h) \\ &= \mathbb{E}^{\mathbb{Q}}\Big[ \log \exp\Big(M_h - \frac{1}{2}[M, M]_h\Big)\Big] \\ &= \mathbb{E}^{\mathbb{Q}}\Big[M_h - \frac{1}{2}[M, M]_h\Big] \\ &= \mathbb{E}^{\mathbb{Q}}\Big[ -\int_0^h \frac{\langle \Delta(X_s, P_s), \mathrm{d}B_s \rangle}{\sqrt{2\gamma}} - \frac{1}{4\gamma}\int_0^h \|\Delta(X_s, P_s)\|^2 \, \mathrm{d}s\Big] \\ &= \mathbb{E}^{\mathbb{Q}}\Big[ -\int_0^h \frac{\langle \Delta(X_s, P_s), \mathrm{d}\widetilde{B}_s \rangle}{\sqrt{2\gamma}} + \frac{1}{2\gamma}\int_0^h \|\Delta(X_s, P_s)\|^2 \, \mathrm{d}s - \frac{1}{4\gamma}\int_0^h \|\Delta(X_s, P_s)\|^2 \, \mathrm{d}s\Big]. \end{aligned}$$

Since $\mathrm{d}\widetilde{B}_t$ is a standard Brownian motion under $\mathbb{Q}$, the first term is equal to 0. Therefore, we have

$$\mathrm{KL}(\mathbb{P}\|\mathbb{P}') \leq \mathbb{E}^{\mathbb{Q}}\Big[ \frac{1}{4\gamma}\int_0^h \|\Delta(X_s, P_s)\|^2 \, \mathrm{d}s\Big].$$

Moreover, under $\mathbb{Q}$, $(X_t, P_t)$ has the same evolution equation as $(X_t', P_t')$, and thus holds the same distribution. We finally derive the inequality:

$$\mathrm{KL}(\mathbb{P}\|\mathbb{P}') \leq \frac{1}{4\gamma}\mathbb{E}\Big[ \int_0^h \|\Delta(X_s', P_s')\|^2 \, \mathrm{d}s\Big].$$

$\square$

**Cross-regularity term.** With the local discretization error controlled (cf. Lemma C.5 and the bounds in Section B), we now bound the cross-regularity term.

*Proof of Lemma 4.2.* We apply (C.2) to decompose the cross-regularity KL divergence into a local discretization term and a kernel-stability term:

$$\mathrm{KL}(\delta_{\mathbf{x},\mathbf{p}}\mathcal{P}_h'\|\delta_{\bar{\mathbf{x}},\bar{\mathbf{p}}}\mathcal{P}_h) \leq 2\mathrm{KL}(\delta_{\mathbf{x},\mathbf{p}}\mathcal{P}_h'\|\delta_{\mathbf{x},\mathbf{p}}\mathcal{P}_h) + R_2(\delta_{\mathbf{x},\mathbf{p}}\mathcal{P}_h\|\delta_{\bar{\mathbf{x}},\bar{\mathbf{p}}}\mathcal{P}_h) \tag{C.3}$$

where $\mathcal{P}_h$ denotes the time-$h$ transition kernel of ULD and $\mathcal{P}_h'$ denotes the one-step ULMC kernel. We bound the two terms on the right-hand side of (C.3) separately. We first use Theorem A.2 to bound the term $R_2(\delta_{\mathbf{x},\mathbf{p}}\mathcal{P}_h\|\delta_{\bar{\mathbf{x}},\bar{\mathbf{p}}}\mathcal{P}_h)$:

$$R_2(\delta_{\mathbf{x},\mathbf{p}}\mathcal{P}_h\|\delta_{\bar{\mathbf{x}},\bar{\mathbf{p}}}\mathcal{P}_h) \lesssim \frac{1}{\gamma}\Big( \frac{\|\mathbf{x} - \bar{\mathbf{x}}\|^2}{h^3} + \frac{\|\mathbf{p} - \bar{\mathbf{p}}\|^2}{h}\Big).$$

Secondly, we bound the term $\mathrm{KL}(\delta_{\mathbf{x},\mathbf{p}}\mathcal{P}_h'\|\delta_{\mathbf{x},\mathbf{p}}\mathcal{P}_h)$. In Lemma C.5, we take

$$\mathbf{b}'(X_t, P_t) = -\gamma P_t - \nabla V(X_t) \qquad \mathbf{b}(X_t^{\mathrm{ULMC}}, P_t^{\mathrm{ULMC}}) = -\gamma P_t^{\mathrm{ULMC}} - \nabla V(\mathbf{x})$$

Then we obtain

$$
\begin{aligned}
\mathrm{KL}(\delta_{\mathbf{x},\mathbf{p}}\mathcal{P}_h' \| \delta_{\mathbf{x},\mathbf{p}}\mathcal{P}_h) &\leq \frac{1}{4\gamma}\mathbb{E}\bigg[\int_0^h \|\nabla V(\boldsymbol{X}_t) - \nabla V(\mathbf{x})\|^2\,\mathrm{d}t\bigg] \\
&\leq \frac{h}{4\gamma}\sup_{t\in[0,h]}\|\nabla V(\boldsymbol{X}_t) - \nabla V(\mathbf{x})\|_{L^2}^2 \\
&\lesssim \frac{1}{\gamma}\Big(\beta h^3\|\mathbf{p}\|_{\mathbf{H}}^2 + \gamma h^4\,\beta\,\mathrm{tr}(\mathbf{H}) + \beta^2 h^5\|\nabla V(\mathbf{x})\|^2\Big),
\end{aligned}
$$

where the second inequality holds due to the Hölder inequality, and the last inequality holds due to Lemma B.1. $\qquad\square$

## D. Proof of Lemmas in Appendix B

### D.1. Proof of Lemma B.1

*Proof of Lemma B.1.* Fix $\lambda > 0$. Using $\mathbf{H}$ is positive semi-definite, we have $(\lambda\mathbf{I} + \mathbf{H})$ is positive definite, thus invertible. Note that

$$
\begin{aligned}
\nabla V(\boldsymbol{X}_t) - \nabla V(\mathbf{x}) &= \bigg[\int_0^1 \nabla^2 V(\mathbf{x} + u(\boldsymbol{X}_t - \mathbf{x}))\,\mathrm{d}u\bigg](\boldsymbol{X}_t - \mathbf{x}) \\
&= (\lambda\mathbf{I} + \mathbf{H})^{1/2}\cdot(\lambda\mathbf{I} + \mathbf{H})^{-1/2}\bigg[\int_0^1 \nabla^2 V(\mathbf{x} + u(\boldsymbol{X}_t - \mathbf{x}))\,\mathrm{d}u\bigg](\lambda\mathbf{I} + \mathbf{H})^{-1/2}\cdot(\lambda\mathbf{I} + \mathbf{H})^{1/2}(\boldsymbol{X}_t - \mathbf{x}).
\end{aligned}
$$

So using the definition of the operator norm, we have

$$
\begin{aligned}
&\|\nabla V(\boldsymbol{X}_t) - \nabla V(\mathbf{x})\| \\
&\leq \|(\lambda\mathbf{I} + \mathbf{H})^{1/2}\|\cdot\bigg\|(\lambda\mathbf{I} + \mathbf{H})^{-1/2}\bigg[\int_0^1 \nabla^2 V(\mathbf{x} + u(\boldsymbol{X}_t - \mathbf{x}))\,\mathrm{d}u\bigg](\lambda\mathbf{I} + \mathbf{H})^{-1/2}\bigg\|\cdot\|(\lambda\mathbf{I} + \mathbf{H})^{1/2}(\boldsymbol{X}_t - \mathbf{x})\| \\
&\leq \sqrt{\lambda + \beta}\cdot\big\|(\lambda\mathbf{I} + \mathbf{H})^{-1/2}\cdot[(\lambda\mathbf{I} + \mathbf{H}) - \lambda\mathbf{I}]\cdot(\lambda\mathbf{I} + \mathbf{H})^{-1/2}\big\|\cdot\big\|(\lambda\mathbf{I} + \mathbf{H})^{1/2}(\boldsymbol{X}_t - \mathbf{x})\big\| \\
&= \sqrt{\lambda + \beta}\big\|(\lambda\mathbf{I} + \mathbf{H})^{1/2}(\boldsymbol{X}_t - \mathbf{x})\big\|\cdot\big\|\mathbf{I} - \lambda(\lambda\mathbf{I} + \mathbf{H})^{-1}\big\|,
\end{aligned}
$$

where the second inequality holds because $\mathbf{H} \preceq \beta\mathbf{I}$ and $\nabla^2 V(\mathbf{z}) \preceq \mathbf{H}$ for any $\mathbf{z}$. Moreover, $\lambda(\lambda\mathbf{I} + \mathbf{H})^{-1} = (\mathbf{I} + \mathbf{H}/\lambda)^{-1}$. Its eigenvalues are in the range of $[0, 1]$. Therefore, the eigenvalues of $(\mathbf{I} - \lambda(\lambda\mathbf{I} + \mathbf{H})^{-1})$ are bounded by 1. This indicates $\|\mathbf{I} - \lambda(\lambda\mathbf{I} + \mathbf{H})^{-1}\| \leq 1$. And we further have

$$
\|\nabla V(\boldsymbol{X}_t) - \nabla V(\mathbf{x})\|^2 \leq (\lambda + \beta)\big\|(\lambda\mathbf{I} + \mathbf{H})^{1/2}(\boldsymbol{X}_t - \mathbf{x})\big\|^2.
$$

Taking the limit $\lambda \to 0$ and the expectation over $\boldsymbol{X}_t$, we have

$$
\mathbb{E}\big[\|\nabla V(\boldsymbol{X}_t) - \nabla V(\mathbf{x})\|^2\big] \leq \beta\cdot\mathbb{E}\big[\|\boldsymbol{X}_t - \mathbf{x}\|_{\mathbf{H}}^2\big]. \tag{D.1}
$$

We then analyze the term $\|\boldsymbol{X}_t - \mathbf{x}\|_{\mathbf{H}}$. According to the integral form of ULD in (3.1),

$$
\begin{aligned}
\boldsymbol{X}_t - \mathbf{x} &= \frac{1 - e^{-\gamma t}}{\gamma}\mathbf{p} + \boldsymbol{\xi}_{0,t}^{(1)} - \int_0^t \frac{1 - e^{-\gamma(t-s)}}{\gamma}\nabla V(\boldsymbol{X}_s)\,\mathrm{d}s \\
&= \frac{1 - e^{-\gamma t}}{\gamma}\mathbf{p} + \boldsymbol{\xi}_{0,t}^{(1)} - \int_0^t \frac{1 - e^{-\gamma(t-s)}}{\gamma}\,\mathrm{d}s\cdot\nabla V(\mathbf{x}) + \int_0^t \frac{1 - e^{-\gamma(t-s)}}{\gamma}[\nabla V(\mathbf{x}) - \nabla V(\boldsymbol{X}_s)]\,\mathrm{d}s.
\end{aligned}
$$

Therefore, using the Cauchy-Schwarz inequality, we have

$$
\begin{aligned}
\mathbb{E}\big[\|\boldsymbol{X}_t - \mathbf{x}\|_{\mathbf{H}}^2\big] &\leq 4\Big(\frac{1 - e^{-\gamma t}}{\gamma}\Big)^2\|\mathbf{p}\|_{\mathbf{H}}^2 + 4\mathbb{E}\big[\|\boldsymbol{\xi}_{0,t}^{(1)}\|_{\mathbf{H}}^2\big] + 4\bigg(\int_0^t \frac{1 - e^{-\gamma(t-s)}}{\gamma}\,\mathrm{d}s\bigg)^2\|\nabla V(\mathbf{x})\|_{\mathbf{H}}^2 \\
&\quad + 4\mathbb{E}\bigg[\bigg\|\int_0^t \frac{1 - e^{-\gamma(t-s)}}{\gamma}[\nabla V(\mathbf{x}) - \nabla V(\boldsymbol{X}_s)]\,\mathrm{d}s\bigg\|_{\mathbf{H}}^2\bigg].
\end{aligned}
$$

Firstly, since $1 - e^{-\gamma t} \leq \gamma t \leq \gamma h$, we have

$$\left(\frac{1 - e^{-\gamma t}}{\gamma}\right)^2 \|\mathbf{p}\|_{\mathbf{H}}^2 \leq h^2 \|\mathbf{p}\|_{\mathbf{H}}^2.$$

Secondly, the term $\|\boldsymbol{\xi}_{0,t}^{(1)}\|_{\mathbf{H}, L_2}$ satisfies

$$
\begin{aligned}
\mathbb{E}\big[\|\boldsymbol{\xi}_{0,t}^{(1)}\|_{\mathbf{H}}^2\big] &= 2\gamma \cdot \mathbb{E}\left\|\int_0^t \frac{1 - e^{-\gamma(t-s)}}{\gamma} \, \mathrm{d}\boldsymbol{B}_s\right\|_{\mathbf{H}}^2 \\
&= 2\gamma \operatorname{tr}(\mathbf{H}) \cdot \int_0^t \left(\frac{1 - e^{-\gamma(t-s)}}{\gamma}\right)^2 \mathrm{d}s \\
&\leq 2\gamma h^3 \operatorname{tr}(\mathbf{H}),
\end{aligned}
\tag{D.2}
$$

where the second equality holds due to the Itô isometry, and the first inequality holds because $1 - e^{-\gamma(t-s)} \leq \gamma(t-s) \leq \gamma h$. Thirdly, we have

$$\int_0^t \frac{1 - e^{-\gamma(t-s)}}{\gamma} \, \mathrm{d}s \leq \int_0^t (t-s) \, \mathrm{d}s = \frac{t^2}{2} \leq \frac{h^2}{2}, \tag{D.3}$$

where the first inequality holds because $1 - e^{-z} \leq z$; we also have $\|\nabla V(\mathbf{x})\|_{\mathbf{H}}^2 \leq \beta \|\nabla V(\mathbf{x})\|^2$ because $\mathbf{H} \preceq \beta \mathbf{I}$. Finally, since

$$
\begin{aligned}
&\left\|\int_0^t \frac{1 - e^{-\gamma(t-s)}}{\gamma} [\nabla V(\mathbf{x}) - \nabla V(\boldsymbol{X}_s)] \, \mathrm{d}s\right\|_{\mathbf{H}}^2 \\
&\leq t \int_0^t \left(\frac{1 - e^{-\gamma(t-s)}}{\gamma}\right)^2 \|\nabla V(\boldsymbol{X}_s) - \nabla V(\mathbf{x})\|_{\mathbf{H}}^2 \, \mathrm{d}s \\
&\leq \beta t \int_0^t (t-s)^2 \|\nabla V(\boldsymbol{X}_s) - \nabla V(\mathbf{x})\|^2 \, \mathrm{d}s,
\end{aligned}
\tag{D.4}
$$

where the first inequality holds due to the Cauchy-Schwarz inequality, and the second inequality holds because $1 - e^{-z} \leq z$ and $\mathbf{H} \preceq \beta \mathbf{I}$. Plugging (D.2), (D.3), and (D.4) into (D.1), we have

$$
\begin{aligned}
\mathbb{E}\big[\|\nabla V(\boldsymbol{X}_t) - \nabla V(\mathbf{x})\|^2\big] \leq{}& 4\beta h^2 \|\mathbf{p}\|_{\mathbf{H}}^2 + 8\beta\gamma h^3 \operatorname{tr}(\mathbf{H}) + \beta^2 h^4 \|\nabla V(\mathbf{x})\|^2 \\
&+ 4\beta^2 t \int_0^t (t-s)^2 \mathbb{E}\big[\|\nabla V(\boldsymbol{X}_s) - \nabla V(\mathbf{x})\|^2\big] \, \mathrm{d}s.
\end{aligned}
$$

Using the Grönwall's Inequality, we have

$$\mathbb{E}\big[\|\nabla V(\boldsymbol{X}_t) - \nabla V(\mathbf{x})\|^2\big] \lesssim e^{\beta^2 t^4} \big(\beta h^2 \|\mathbf{p}\|_{\mathbf{H}}^2 + \beta\gamma t^3 \operatorname{tr}(\mathbf{H}) + \beta^2 t^4 \|\nabla V(\mathbf{x})\|^2\big)$$

Therefore, as long as $t \in [0, h]$ where $h = \mathcal{O}(1/\sqrt{\beta})$, the inequality above becomes

$$\mathbb{E}\big[\|\nabla V(\boldsymbol{X}_t) - \nabla V(\mathbf{x})\|^2\big] \lesssim \beta h^2 \|\mathbf{p}\|_{\mathbf{H}}^2 + \beta\gamma t^3 \operatorname{tr}(\mathbf{H}) + \beta^2 t^4 \|\nabla V(\mathbf{x})\|^2.$$

Taking the square root on both sides, we have

$$\|\nabla V(\boldsymbol{X}_t) - \nabla V(\mathbf{x})\|_{L_2} \lesssim \beta^{1/2} h \|\mathbf{p}\|_{\mathbf{H}} + \beta^{1/2} \gamma^{1/2} t^{3/2} \sqrt{\operatorname{tr}(\mathbf{H})} + \beta t^2 \|\nabla V(\mathbf{x})\|.$$

$\square$

## D.2. Proof of Lemma B.2

*Proof of Lemma B.2.* Due to the $\beta$-Lipschitzness of $\nabla V$, we have

$$\|\nabla V(\boldsymbol{X}_t^{\mathrm{ULMC}}) - \nabla V(\boldsymbol{X}_t)\| \leq \beta \|\boldsymbol{X}_t^{\mathrm{ULMC}} - \boldsymbol{X}_t\|.$$

Taking the expectation $\sqrt{\mathbb{E}[\cdot]^2}$, we have

$$\|\nabla V(\boldsymbol{X}_t^{\mathrm{ULMC}}) - \nabla V(\boldsymbol{X}_t)\|_{L_2} \leq \beta \|\boldsymbol{X}_t^{\mathrm{ULMC}} - \boldsymbol{X}_t\|_{L_2} \lesssim \beta^{3/2} h^3 \|\mathbf{p}\|_{\mathbf{H}} + \gamma^{1/2} \beta^{3/2} h^{7/2} \sqrt{\operatorname{tr}(\mathbf{H})} + \beta^2 h^4 \|\nabla V(\mathbf{x})\|,$$

where the second inequality holds due to (B.6). $\square$

# E. Proof of change-of-measure lemma

In this section, we prove Lemma 6.1.

*Proof of Lemma 6.1.* In the Donsker–Varadhan variational lemma (Lemma I.3), by setting $U(\mathbf{x}) = \|\nabla V(\mathbf{x})\|^2/(4\beta)$, we have,

$$\frac{\mathbb{E}_\mu[\|\nabla V(\mathbf{x})\|^2]}{4\beta} \leq \mathrm{KL}(\mu\|\pi) + \log \mathbb{E}_\pi\Big[\exp\big(\|\nabla V(\mathbf{x})\|^2/(4\beta)\big)\Big] \leq \mathrm{KL}(\mu\|\pi) + \frac{\mathrm{tr}(\mathbf{H})}{2\beta},$$

where the second inequality holds due to Lemma I.4. Therefore, rearranging terms,

$$\mathbb{E}_\mu[\|\nabla V(\mathbf{x})\|^2] \leq 2\,\mathrm{tr}(\mathbf{H}) + 4\beta \cdot \mathrm{KL}(\mu\|\pi)$$

Similarly, for the bound of $\mathbb{E}_\mu[\|\mathbf{p}\|_{\mathbf{H}}^2]$, using Lemma I.3 with $U(\mathbf{p}) = \|\mathbf{p}\|_{\mathbf{H}}^2/(4\beta)$, we have

$$\frac{\mathbb{E}_\mu[\|\mathbf{p}\|_{\mathbf{H}}^2]}{4\beta} \leq \mathrm{KL}(\mu\|\pi) + \log \mathbb{E}_\pi\Big[\exp\big(\|\mathbf{p}\|_{\mathbf{H}}^2/(4\beta)\big)\Big] \leq \mathrm{KL}(\mu\|\pi) + \frac{\mathrm{tr}(\mathbf{H})}{2\beta},$$

where the second inequality holds due to Lemma I.5. Therefore, rearranging terms,

$$\mathbb{E}_\mu[\|\mathbf{p}\|_{\mathbf{H}}^2] \leq 2\,\mathrm{tr}(\mathbf{H}) + 4\beta \cdot \mathrm{KL}(\mu\|\pi).$$

$\square$

# F. Proof of Theorems in Section 4

In this section, we prove the sample complexity for ULMC. To start with, we first verify Assumption A.5 for ULMC. We have the following lemma:

**Lemma F.1.** For ULMC, we can compute the Lipschitz constants of the strong and weak errors:

$$L_{w,\mathbf{x}} = L_{s,\mathbf{x}} = \beta^2 h^3,$$
$$L_{w,\mathbf{p}} = L_{s,\mathbf{p}} = \beta h^2.$$

Moreover, if $h \lesssim 1/(\beta^{1/2}\kappa)$ in the strongly convex setting, or $h \lesssim N^{-1/2}\beta^{-1/2}$ in the general convex setting, Assumption A.5 holds for ULMC.

## F.1. Proof of Theorem 4.3

In this section, we make a formal proof of Theorem 4.3.

*Proof of Theorem 4.3.* Using Lemma F.1, we know that if Assumption A.5 holds, we need:

$$h \leq \frac{1}{\beta^{1/2}\kappa}. \tag{F.1}$$

We can apply Theorem A.6 if $h \lesssim 1/(\beta^{1/2}\kappa)$. For any $n \leq N - 1$, we have

$$\mathrm{KL}\big(\nu_n^{\mathrm{aux}}\|\nu_n\big) \lesssim C\mathcal{W}_2^2(\mu, \nu) + A_w\Big(\max_{1 \leq i \leq n-1} \mathbb{E}_{\nu_i^{\mathrm{aux}}}\big[(\mathcal{E}^w)^2\big]\Big) + A_s\Big(\max_{1 \leq i \leq n-1} \mathbb{E}_{\nu_i^{\mathrm{aux}}}\big[(\mathcal{E}^s)^2\big]\Big).$$

Let $\nu = \pi$ be the invariant distribution of the underdamped Langevin. Then we can see $\nu_n = \pi$ for any $n$. We have

$$\mathrm{KL}\big(\nu_n^{\mathrm{aux}}\|\pi\big) \lesssim C\mathcal{W}_2^2(\mu, \pi) + A_w\Big(\max_{1 \leq i \leq n-1} \mathbb{E}_{\nu_i^{\mathrm{aux}}}\big[(\mathcal{E}^w)^2\big]\Big) + A_s\Big(\max_{1 \leq i \leq n-1} \mathbb{E}_{\nu_i^{\mathrm{aux}}}\big[(\mathcal{E}^s)^2\big]\Big). \tag{F.2}$$

In ULMC, there is no separate weak error term, and we may take

$$\mathcal{E}^{\mathrm{w}}(\mathbf{x}, \mathbf{p}) = \mathcal{E}^{\mathrm{s}}(\mathbf{x}, \mathbf{p}).$$

Moreover, Lemma 4.1 gives

$$\mathcal{E}^{\mathrm{s}}(\mathbf{x}, \mathbf{p}) \lesssim \beta^{1/2} h^2 \|\mathbf{p}\|_{\mathbf{H}} + \beta h^3 \|\nabla V(\mathbf{x})\| + \beta^{1/2} \gamma^{1/2} h^{5/2} \sqrt{\operatorname{tr}(\mathbf{H})}.$$

Thus, taking the expectation,

$$\max_{1 \le i \le n-1} \mathbb{E}_{\nu_i^{\mathrm{aux}}} \big[ (\mathcal{E}^w)^2 \big] = \max_{1 \le i \le n-1} \mathbb{E}_{\nu_i^{\mathrm{aux}}} \big[ (\mathcal{E}^s)^2 \big] \lesssim \beta h^4 \Big[ \max_{1 \le i \le n-1} \mathbb{E}_{\nu_i^{\mathrm{aux}}} \big[ \|\mathbf{p}\|_{\mathbf{H}}^2 \big] \Big] + \beta^2 h^6 \Big[ \max_{1 \le i \le n-1} \mathbb{E}_{\nu_i^{\mathrm{aux}}} \big[ \|\nabla V(\mathbf{x})\|^2 \big] \Big]$$
$$+ \beta \gamma h^5 \operatorname{tr}(\mathbf{H}). \tag{F.3}$$

Then, substituting (F.3) into (F.2), we have

$$\operatorname{KL}\big(\nu_n^{\mathrm{aux}} \| \pi\big) \lesssim C \mathcal{W}_2^2(\mu, \pi) + \big( A_w + A_s \big) \beta \gamma h^5 \operatorname{tr}(\mathbf{H})$$
$$+ \big( A_w + A_s \big) \beta h^4 \Big[ \max_{1 \le i \le n-1} \mathbb{E}_{\nu_i^{\mathrm{aux}}} \big[ \|\mathbf{p}\|_{\mathbf{H}}^2 \big] \Big] + \big( A_w + A_s \big) \beta^2 h^6 \Big[ \max_{1 \le i \le n-1} \mathbb{E}_{\nu_i^{\mathrm{aux}}} \big[ \|\nabla V(\mathbf{x})\|^2 \big] \Big].$$

In the strongly convex case,

$$A_w = \frac{1}{\alpha h^2}, \qquad A_s = \frac{1}{\beta^{1/2} h} \log\Big( \frac{3\gamma}{\alpha h} \Big).$$

When $h \le \beta^{-1/2} \kappa \log^{-1}\big[ (3\gamma)/(\alpha h) \big]$, there is $A_s \le A_w$, so we can drop all of the $A_s$ terms, and reduce the inequality to

$$\operatorname{KL}\big(\nu_n^{\mathrm{aux}} \| \pi\big) \lesssim C \mathcal{W}_2^2(\mu, \pi) + \kappa \gamma h^3 \operatorname{tr}(\mathbf{H}) + \kappa h^2 \Big[ \max_{1 \le i \le n-1} \mathbb{E}_{\nu_i^{\mathrm{aux}}} \big[ \|\mathbf{p}\|_{\mathbf{H}}^2 \big] \Big]$$
$$+ \kappa \beta h^4 \Big[ \max_{1 \le i \le n-1} \mathbb{E}_{\nu_i^{\mathrm{aux}}} \big[ \|\nabla V(\mathbf{x})\|^2 \big] \Big]. \tag{F.4}$$

Using Lemma 6.1, we have

$$\operatorname{KL}\big(\nu_n^{\mathrm{aux}} \| \pi\big) \lesssim C \mathcal{W}_2^2(\mu, \pi) + \kappa \gamma h^3 \operatorname{tr}(\mathbf{H})$$
$$+ \kappa h^2 \Big[ \operatorname{tr}(\mathbf{H}) + \beta \max_{1 \le i \le n-1} \operatorname{KL}(\nu_i^{\mathrm{aux}} \| \pi) \Big] + \kappa \beta h^4 \Big[ \operatorname{tr}(\mathbf{H}) + \beta \max_{1 \le i \le n-1} \operatorname{KL}(\nu_i^{\mathrm{aux}} \| \pi) \Big]$$
$$\lesssim C \mathcal{W}_2^2(\mu, \pi) + \kappa h^2 \operatorname{tr}(\mathbf{H}) + \kappa \beta h^2 \Big[ \max_{1 \le i \le n-1} \operatorname{KL}(\nu_i^{\mathrm{aux}} \| \pi) \Big],$$

where we used $\gamma = O(\beta^{1/2})$ and $h \le \beta^{-1/2}$ to absorb the $\kappa \gamma h^3 \operatorname{tr}(\mathbf{H})$ and $\kappa \beta h^4 \operatorname{tr}(\mathbf{H})$ terms into $\kappa h^2 \log(\cdot) \operatorname{tr}(\mathbf{H})$. Since this inequality holds for any $n \le N - 1$, we have

$$\max_{1 \le i \le n} \operatorname{KL}\big(\nu_i^{\mathrm{aux}} \| \pi\big) \lesssim C \mathcal{W}_2^2(\mu, \pi) + \kappa h^2 \operatorname{tr}(\mathbf{H}) + \kappa \beta h^2 \max_{1 \le i \le n-1} \operatorname{KL}(\nu_i^{\mathrm{aux}} \| \pi).$$

Using the condition $h \le \beta^{-1/2} \kappa^{-1/2}$, we know

$$\kappa \beta h^2 \lesssim 1.$$

for a small constant. Then, we can see that for any $n \le N - 1$,

$$\max_{1 \le i \le n} \operatorname{KL}\big(\nu_i^{\mathrm{aux}} \| \pi\big) \lesssim C \mathcal{W}_2^2(\mu, \pi) + \kappa h^2 \operatorname{tr}(\mathbf{H}). \tag{F.5}$$

Finally, using the second inequality in Theorem A.6 with $\mu = \pi$, we have

$$\operatorname{KL}\big(\mu(\mathcal{P}')^N \| \pi\big) \lesssim C \mathcal{W}_2^2(\mu, \pi) + A_w \Big( \max_{1 \le i \le N-1} \mathbb{E}_{\nu_i^{\mathrm{aux}}} \big[ (\mathcal{E}^w)^2 \big] \Big) + A_s \Big( \max_{1 \le i \le N-1} \mathbb{E}_{\nu_i^{\mathrm{aux}}} \big[ (\mathcal{E}^s)^2 \big] \Big) + \max_{1 \le i \le N-1} \mathbb{E}_{\nu_i^{\mathrm{aux}}} \big[ b^2 \big]$$
$$\lesssim C \mathcal{W}_2^2(\mu, \pi) + A_w \Big( \max_{1 \le i \le N-1} \mathbb{E}_{\nu_i^{\mathrm{aux}}} \big[ (\mathcal{E}^s)^2 \big] \Big) + \max_{1 \le i \le N-1} \mathbb{E}_{\nu_i^{\mathrm{aux}}} \big[ b^2 \big]$$
$$\lesssim C \mathcal{W}_2^2(\mu, \pi) + \kappa \gamma h^3 \operatorname{tr}(\mathbf{H}) + \kappa h^2 \Big[ \max_{1 \le i \le N-1} \mathbb{E}_{\nu_i^{\mathrm{aux}}} \big[ \|\mathbf{p}\|_{\mathbf{H}}^2 \big] \Big] + \kappa \beta h^4 \Big[ \max_{1 \le i \le N-1} \mathbb{E}_{\nu_i^{\mathrm{aux}}} \big[ \|\nabla V(\mathbf{x})\|^2 \big] \Big]$$

$$+ \max_{1 \le i \le N-1} \mathbb{E}_{\nu_i^{\mathrm{aux}}}\big[b^2\big],$$

where the second inequality holds since in ULMC there is no separate weak error term and we may take $(\mathcal{E}^w)^2 = (\mathcal{E}^s)^2$, and the third inequality holds due to the same calculation as (F.4). Moreover, by Lemma 4.2, we have

$$\max_{1 \le i \le N-1} \mathbb{E}_{\nu_i^{\mathrm{aux}}}\big[b^2\big] \lesssim \frac{\beta h^3}{\gamma}\Big[\max_{1 \le i \le N-1} \mathbb{E}_{\nu_i^{\mathrm{aux}}}\big[\|\mathbf{p}\|_{\mathbf{H}}^2\big]\Big] + \beta h^4 \operatorname{tr}(\mathbf{H}) + \frac{\beta^2 h^5}{\gamma}\Big[\max_{1 \le i \le N-1} \mathbb{E}_{\nu_i^{\mathrm{aux}}}\big[\|\nabla V(\mathbf{x})\|^2\big]\Big].$$

We substitute the bound above into the inequality and derive

$$\begin{aligned}
\mathrm{KL}\big(\mu(\mathcal{P}')^N\|\pi\big) &\lesssim C\mathcal{W}_2^2(\mu,\pi) + \kappa\gamma h^3 \operatorname{tr}(\mathbf{H}) + \kappa h^2 \Big[\max_{1 \le i \le N-1} \mathbb{E}_{\nu_i^{\mathrm{aux}}}\big[\|\mathbf{p}\|_{\mathbf{H}}^2\big]\Big] \\
&\quad + \kappa\beta h^4 \Big[\max_{1 \le i \le N-1} \mathbb{E}_{\nu_i^{\mathrm{aux}}}\big[\|\nabla V(\mathbf{x})\|^2\big]\Big] + \frac{\beta h^3}{\gamma}\Big[\max_{1 \le i \le N-1} \mathbb{E}_{\nu_i^{\mathrm{aux}}}\big[\|\mathbf{p}\|_{\mathbf{H}}^2\big]\Big] \\
&\quad + \beta h^4 \operatorname{tr}(\mathbf{H}) + \frac{\beta^2 h^5}{\gamma}\Big[\max_{1 \le i \le N-1} \mathbb{E}_{\nu_i^{\mathrm{aux}}}\big[\|\nabla V(\mathbf{x})\|^2\big]\Big].
\end{aligned}$$

Using Lemma 6.1, we have

$$\begin{aligned}
\mathrm{KL}\big(\mu(\mathcal{P}')^N\|\pi\big) &\lesssim C\mathcal{W}_2^2(\mu,\pi) + \kappa\gamma h^3 \operatorname{tr}(\mathbf{H}) + \kappa h^2 \Big[\operatorname{tr}(\mathbf{H}) + \beta \max_{1 \le i \le N-1} \mathrm{KL}(\nu_i^{\mathrm{aux}}\|\pi)\Big] \\
&\quad + \kappa\beta h^4 \Big[\operatorname{tr}(\mathbf{H}) + \beta \max_{1 \le i \le N-1} \mathrm{KL}(\nu_i^{\mathrm{aux}}\|\pi)\Big] + \frac{\beta h^3}{\gamma}\Big[\operatorname{tr}(\mathbf{H}) + \beta \max_{1 \le i \le N-1} \mathrm{KL}(\nu_i^{\mathrm{aux}}\|\pi)\Big] \\
&\quad + \beta h^4 \operatorname{tr}(\mathbf{H}) + \frac{\beta^2 h^5}{\gamma}\Big[\operatorname{tr}(\mathbf{H}) + \beta \max_{1 \le i \le N-1} \mathrm{KL}(\nu_i^{\mathrm{aux}}\|\pi)\Big].
\end{aligned}$$

Here, since $\beta h^2 \le 1$ and $\gamma \simeq \beta^{1/2}$, there is $\beta h^3/\gamma \simeq \beta^{1/2}h^3 \lesssim \kappa\beta^{1/2}h^3$, $\beta^2 h^5/\gamma \le \beta^{1/2}h^3 \le \kappa h^2$. We can drop the lower-order terms induced by the cross-regularity and obtain

$$\mathrm{KL}\big(\mu(\mathcal{P}')^N\|\pi\big) \lesssim C\mathcal{W}_2^2(\mu,\pi) + \kappa h^2 \operatorname{tr}(\mathbf{H}) + \kappa\beta h^2 \max_{1 \le i \le N-1} \mathrm{KL}(\nu_i^{\mathrm{aux}}\|\pi),$$

Substituting (F.5) into the inequality above and dropping the low-order terms, we get the final bound of the KL divergence as:

$$\mathrm{KL}\big(\mu(\mathcal{P}^{\mathrm{alg}})^{N-1}\widetilde{\mathcal{P}}\|\pi\big) \lesssim C\mathcal{W}_2^2(\mu,\pi) + \kappa h^2 \operatorname{tr}(\mathbf{H}).$$

Let $Nh \simeq \alpha^{-1}\beta^{1/2}\log(\alpha W^2/\epsilon^2)$ such that $C\mathcal{W}_2^2(\mu,\pi) \lesssim \epsilon^2$. Moreover, let the stepsize be

$$h \simeq \frac{\epsilon}{\kappa^{1/2}\big(\operatorname{tr}(\mathbf{H})\big)^{1/2}},$$

Then, to fulfill the condition discussed in (F.1), we require $\epsilon \le \big(\operatorname{tr}(\mathbf{H})\big)^{1/2}/(\beta^{1/2}\kappa^{1/2})$.
So, when the required sample complexity is

$$N \simeq \frac{\alpha^{-1}\beta^{1/2}\log(\alpha W^2/\epsilon^2)}{\big(\epsilon^2/(\kappa \operatorname{tr}(\mathbf{H}))\big)^{1/2}} = \widetilde{\Theta}\Big(\frac{\beta\big(\operatorname{tr}(\mathbf{H})\big)^{1/2}}{\alpha^{3/2}\epsilon}\Big)\log\Big(\frac{\alpha W^2}{\epsilon^2}\Big),$$

the KL divergence can be upper bounded by $\epsilon^2$, i.e.,

$$\mathrm{KL}\big(\mu(\mathcal{P}^{\mathrm{alg}})^N\|\pi\big) \lesssim \epsilon^2.$$

$\square$

## F.2. Proof of Theorem 4.4

In this section, we now turn into the general convex (i.e. $\alpha = 0$) case. Using Lemma F.1, if the Assumption A.5 holds, we require:

$$h \lesssim N^{-1/2}\beta^{-1/2}. \tag{F.6}$$

Same as (F.2), for any $n \leq N - 1$, we also have:

$$\mathrm{KL}\big(\nu_n^{\mathrm{aux}}\|\pi\big) \lesssim C\mathcal{W}_2^2(\mu, \pi) + A_w\Big(\max_{1 \leq i \leq n-1} \mathbb{E}_{\nu_i^{\mathrm{aux}}}\big[(\mathcal{E}^w)^2\big]\Big) + A_s\Big(\max_{1 \leq i \leq n-1} \mathbb{E}_{\nu_i^{\mathrm{aux}}}\big[(\mathcal{E}^s)^2\big]\Big). \tag{F.7}$$

And we still have the bound of strong and weak errors. In ULMC, there is no separate weak error, and we may take $\mathcal{E}^w(\mathbf{x}, \mathbf{p}) = \mathcal{E}^s(\mathbf{x}, \mathbf{p})$, where $\mathcal{E}^s$ is the same as the strong error bound in the RMD case.

$$\mathcal{E}^w(\mathbf{x}, \mathbf{p}) = \mathcal{E}^s(\mathbf{x}, \mathbf{p}) \lesssim \beta^{1/2}h^2 \|\mathbf{p}\|_{\mathbf{H}} + \beta h^3 \|\nabla V(\mathbf{x})\| + \beta^{1/2}\gamma^{1/2}h^{5/2}\sqrt{\mathrm{tr}(\mathbf{H})}.$$

Thus, taking the expectation,

$$\max_{1 \leq i \leq n-1} \mathbb{E}_{\nu_i^{\mathrm{aux}}}\big[(\mathcal{E}^w)^2\big] \lesssim \beta h^4 \Big[\max_{1 \leq i \leq n-1} \mathbb{E}_{\nu_i^{\mathrm{aux}}}\big[\|\mathbf{p}\|_{\mathbf{H}}^2\big]\Big] + \beta^2 h^6 \Big[\max_{1 \leq i \leq n-1} \mathbb{E}_{\nu_i^{\mathrm{aux}}}\big[\|\nabla V(\mathbf{x})\|^2\big]\Big] + \beta\gamma h^5 \mathrm{tr}(\mathbf{H}),$$

$$\max_{1 \leq i \leq n-1} \mathbb{E}_{\nu_i^{\mathrm{aux}}}\big[(\mathcal{E}^s)^2\big] \lesssim \beta h^4 \Big[\max_{1 \leq i \leq n-1} \mathbb{E}_{\nu_i^{\mathrm{aux}}}\big[\|\mathbf{p}\|_{\mathbf{H}}^2\big]\Big] + \beta^2 h^6 \Big[\max_{1 \leq i \leq n-1} \mathbb{E}_{\nu_i^{\mathrm{aux}}}\big[\|\nabla V(\mathbf{x})\|^2\big]\Big] + \beta\gamma h^5 \mathrm{tr}(\mathbf{H}). \tag{F.8}$$

Then, substituting (F.8) into (F.7), we have

$$\mathrm{KL}\big(\nu_n^{\mathrm{aux}}\|\pi\big) \lesssim C\mathcal{W}_2^2(\mu, \pi) + (A_w + A_s)\beta\gamma h^5 \mathrm{tr}(\mathbf{H}) + (A_w + A_s)\beta h^4 \Big[\max_{1 \leq i \leq n-1} \mathbb{E}_{\nu_i^{\mathrm{aux}}}\big[\|\mathbf{p}\|_{\mathbf{H}}^2\big]\Big]$$
$$+ (A_w + A_s)\beta^2 h^6 \Big[\max_{1 \leq i \leq n-1} \mathbb{E}_{\nu_i^{\mathrm{aux}}}\big[\|\nabla V(\mathbf{x})\|^2\big]\Big]. \tag{F.9}$$

In the general convex case,

$$A_w = \frac{N}{\beta^{1/2}h}, \qquad A_s = \frac{1}{\beta^{1/2}h}\log N + \beta^{1/2}Nh. \tag{F.10}$$

Moreover, assume in addition that

$$\log N \leq \beta Nh^2,$$

so that

$$A_s = \frac{1}{\beta^{1/2}h}\log N + \beta^{1/2}Nh \lesssim \beta^{1/2}Nh.$$

And since $h \lesssim \beta^{-1/2}$, we have $\beta^{1/2}Nh \leq A_w$, so $A_s \lesssim A_w$. Substituting the results into (F.9) and dropping the lower-order terms, we obtain

$$\mathrm{KL}\big(\nu_n^{\mathrm{aux}}\|\pi\big) \lesssim C\mathcal{W}_2^2(\mu, \pi) + A_w\beta\gamma h^5 \mathrm{tr}(\mathbf{H}) + A_w\beta h^4 \Big[\max_{1 \leq i \leq n-1} \mathbb{E}_{\nu_i^{\mathrm{aux}}}\big[\|\mathbf{p}\|_{\mathbf{H}}^2\big]\Big] + A_w\beta^2 h^6 \Big[\max_{1 \leq i \leq n-1} \mathbb{E}_{\nu_i^{\mathrm{aux}}}\big[\|\nabla V(\mathbf{x})\|^2\big]\Big]$$
$$\lesssim C\mathcal{W}_2^2(\mu, \pi) + \beta^{1/2}\gamma Nh^4 \mathrm{tr}(\mathbf{H}) + \beta^{1/2}Nh^3 \Big[\max_{1 \leq i \leq n-1} \mathbb{E}_{\nu_i^{\mathrm{aux}}}\big[\|\mathbf{p}\|_{\mathbf{H}}^2\big]\Big] + \beta^{3/2}Nh^5 \Big[\max_{1 \leq i \leq n-1} \mathbb{E}_{\nu_i^{\mathrm{aux}}}\big[\|\nabla V(\mathbf{x})\|^2\big]\Big].$$

Using Lemma 6.1, we have

$$\mathrm{KL}\big(\nu_n^{\mathrm{aux}}\|\pi\big) \lesssim C\mathcal{W}_2^2(\mu, \pi) + \beta^{1/2}\gamma Nh^4 \mathrm{tr}(\mathbf{H}) + \beta^{1/2}Nh^3 \Big[\mathrm{tr}(\mathbf{H}) + \beta \max_{1 \leq i \leq n-1} \mathrm{KL}(\nu_i^{\mathrm{aux}}\|\pi)\Big]$$
$$+ \beta^{3/2}Nh^5 \Big[\mathrm{tr}(\mathbf{H}) + \beta \max_{1 \leq i \leq n-1} \mathrm{KL}(\nu_i^{\mathrm{aux}}\|\pi)\Big].$$

Since this inequality holds for any $n \leq N-1$, and we have $\gamma h \simeq \beta^{1/2} h \leq 1$, so we have $\beta^{1/2}\gamma N h^4 \leq \beta^{1/2} N h^3$. Besides, we also have $\beta^{3/2} N h^5 \leq \beta^{1/2} N h^3$ because of $\beta h^2 \leq 1$, so to drop all of the lower-order term, we have:

$$\max_{1 \leq i \leq n} \mathrm{KL}\big(\nu_i^{\mathrm{aux}} \| \pi\big) \lesssim C \mathcal{W}_2^2(\mu, \pi) + \beta^{1/2} N h^3 \, \mathrm{tr}(\mathbf{H}) + \beta^{3/2} N h^3 \max_{1 \leq i \leq n-1} \mathrm{KL}(\nu_i^{\mathrm{aux}} \| \pi).$$

Using the condition

$$\beta^{3/2} N h^3 \lesssim c < 1, \tag{F.11}$$

we can see that for any $n \leq N-1$,

$$\max_{1 \leq i \leq n} \mathrm{KL}\big(\nu_i^{\mathrm{aux}} \| \pi\big) \lesssim C \mathcal{W}_2^2(\mu, \pi) + \beta^{1/2} N h^3 \, \mathrm{tr}(\mathbf{H}). \tag{F.12}$$

Finally, using the second inequality in Corollary 3.8 with $\nu = \pi$, we have

$$
\begin{aligned}
\mathrm{KL}\big(\mu(\mathcal{P}')^N \| \pi\big) &\lesssim C \mathcal{W}_2^2(\mu, \pi) + A_w \big(\bar{\mathcal{E}}^w\big)^2 + A_s \big(\bar{\mathcal{E}}^s\big)^2 + \bar{b}^2 \\
&\lesssim C \mathcal{W}_2^2(\mu, \pi) + A_w \Big( \max_{1 \leq i \leq N-1} \mathbb{E}_{\nu_i^{\mathrm{aux}}} \big[(\mathcal{E}^w)^2\big] \Big) + A_s \Big( \max_{1 \leq i \leq N-1} \mathbb{E}_{\nu_i^{\mathrm{aux}}} \big[(\mathcal{E}^s)^2\big] \Big) + \max_{1 \leq i \leq N-1} \mathbb{E}_{\nu_i^{\mathrm{aux}}} \big[b^2\big].
\end{aligned}
$$

In ULMC, there is no separate weak error, and we may take $(\mathcal{E}^w)^2 = (\mathcal{E}^s)^2$. Moreover, by (F.8), we have

$$
\begin{aligned}
\max_{1 \leq i \leq N-1} \mathbb{E}_{\nu_i^{\mathrm{aux}}} \big[(\mathcal{E}^w)^2\big] = \max_{1 \leq i \leq N-1} \mathbb{E}_{\nu_i^{\mathrm{aux}}} \big[(\mathcal{E}^s)^2\big] &\lesssim \beta h^4 \Big[ \max_{1 \leq i \leq N-1} \mathbb{E}_{\nu_i^{\mathrm{aux}}} \big[\|\mathbf{p}\|_{\mathbf{H}}^2\big] \Big] + \beta^2 h^6 \Big[ \max_{1 \leq i \leq N-1} \mathbb{E}_{\nu_i^{\mathrm{aux}}} \big[\|\nabla V(\mathbf{x})\|^2\big] \Big] \\
&\quad + \beta\gamma h^5 \mathrm{tr}(\mathbf{H}).
\end{aligned}
$$

Furthermore, by Lemma 4.2, we have

$$\max_{1 \leq i \leq N-1} \mathbb{E}_{\nu_i^{\mathrm{aux}}} \big[b^2\big] \lesssim \frac{\beta h^3}{\gamma} \Big[ \max_{1 \leq i \leq N-1} \mathbb{E}_{\nu_i^{\mathrm{aux}}} \big[\|\mathbf{p}\|_{\mathbf{H}}^2\big] \Big] + \frac{\beta^2 h^5}{\gamma} \Big[ \max_{1 \leq i \leq N-1} \mathbb{E}_{\nu_i^{\mathrm{aux}}} \big[\|\nabla V(\mathbf{x})\|^2\big] \Big] + \beta h^4 \, \mathrm{tr}(\mathbf{H}).$$

Similarly, we also have $A_s \lesssim A_w$, so to drop the lower-order terms, we obtain:

$$
\begin{aligned}
\mathrm{KL}\big(\mu(\mathcal{P}')^N \| \pi\big) &\lesssim C \mathcal{W}_2^2(\mu, \pi) + A_w \beta\gamma h^5 \, \mathrm{tr}(\mathbf{H}) \\
&\quad + A_w \beta h^4 \Big[ \max_{1 \leq i \leq N-1} \mathbb{E}_{\nu_i^{\mathrm{aux}}} \big[\|\mathbf{p}\|_{\mathbf{H}}^2\big] \Big] + A_w \beta^2 h^6 \Big[ \max_{1 \leq i \leq N-1} \mathbb{E}_{\nu_i^{\mathrm{aux}}} \big[\|\nabla V(\mathbf{x})\|^2\big] \Big] \\
&\quad + \frac{\beta h^3}{\gamma} \Big[ \max_{1 \leq i \leq N-1} \mathbb{E}_{\nu_i^{\mathrm{aux}}} \big[\|\mathbf{p}\|_{\mathbf{H}}^2\big] \Big] + \frac{\beta^2 h^5}{\gamma} \Big[ \max_{1 \leq i \leq N-1} \mathbb{E}_{\nu_i^{\mathrm{aux}}} \big[\|\nabla V(\mathbf{x})\|^2\big] \Big] + \beta h^4 \, \mathrm{tr}(\mathbf{H}).
\end{aligned}
$$

Using Lemma 6.1, and $A_w = N/(\beta^{1/2} h)$, we have

$$
\begin{aligned}
\mathrm{KL}\big(\mu(\mathcal{P}')^N \| \pi\big) &\lesssim C \mathcal{W}_2^2(\mu, \pi) + \beta^{1/2}\gamma h^4 N \, \mathrm{tr}(\mathbf{H}) \\
&\quad + \beta^{1/2} h^3 N \Big[ \mathrm{tr}(\mathbf{H}) + \beta \max_{1 \leq i \leq N-1} \mathrm{KL}(\nu_i^{\mathrm{aux}} \| \pi) \Big] + \beta^{3/2} h^5 N \Big[ \mathrm{tr}(\mathbf{H}) + \beta \max_{1 \leq i \leq N-1} \mathrm{KL}(\nu_i^{\mathrm{aux}} \| \pi) \Big] \\
&\quad + \frac{\beta h^3}{\gamma} \Big[ \mathrm{tr}(\mathbf{H}) + \beta \max_{1 \leq i \leq N-1} \mathrm{KL}(\nu_i^{\mathrm{aux}} \| \pi) \Big] + \frac{\beta^2 h^5}{\gamma} \Big[ \mathrm{tr}(\mathbf{H}) + \beta \max_{1 \leq i \leq N-1} \mathrm{KL}(\nu_i^{\mathrm{aux}} \| \pi) \Big] + \beta h^4 \, \mathrm{tr}(\mathbf{H}).
\end{aligned}
$$

Since $\gamma \simeq \sqrt{\beta}$, and $\sqrt{\beta} h \lesssim 1, 1 \leq N$, so we could drop all of the lower-order term, and finally obtain

$$\mathrm{KL}\big(\mu(\mathcal{P}')^N \| \pi\big) \lesssim C \mathcal{W}_2^2(\mu, \pi) + \beta^{1/2} N h^3 \, \mathrm{tr}(\mathbf{H}) + \beta^{3/2} N h^3 \max_{1 \leq i \leq N-1} \mathrm{KL}(\nu_i^{\mathrm{aux}} \| \pi),$$

where the last inequality holds due to the same calculation as above. Substituting (F.12) into the inequality above, we obtain

$$\mathrm{KL}\big(\mu(\mathcal{P}')^N \| \pi\big) \lesssim C \mathcal{W}_2^2(\mu, \pi) + \beta^{1/2} N h^3 \, \mathrm{tr}(\mathbf{H}) + \beta^{3/2} N h^3 \Big[ C \mathcal{W}_2^2(\mu, \pi) + \beta^{1/2} N h^3 \, \mathrm{tr}(\mathbf{H}) \Big].$$

Dropping the low-order terms, and applying the condition (F.11), we finally obtain

$$\mathrm{KL}\big(\mu(\mathcal{P}')^N \| \pi\big) \lesssim C \mathcal{W}_2^2(\mu, \pi) + \beta^{1/2} N h^3 \, \mathrm{tr}(\mathbf{H}). \tag{F.13}$$

In general convex case, $C \simeq \gamma/(Nh)$, so if $CW^2 \simeq \epsilon^2$, we have:

$$Nh \simeq \frac{\beta^{1/2}W^2}{\epsilon^2}.$$

While we also need $\beta^{1/2}Nh^3 tr(\mathbf{H}) \simeq \epsilon^2$ required in (F.13), and $\beta^{3/2}Nh^3 \lesssim c < 1$. Substitute the $Nh$, Then, we have:

$$\beta h^2 \, \mathrm{tr}(\mathbf{H})W^2 \simeq \epsilon^4, \frac{\beta^2 h^2 W^2}{\epsilon^2} \lesssim 1.$$

Besides, we require the Assumption A.5 holds, which is discussed in (F.6) $\epsilon \lesssim \frac{1}{N^{1/2}\beta^{1/2}}$, substitute the $Nh$, that is: $h \lesssim \epsilon^2/(\beta^{3/2}W^2)$. We could finally get:

$$h = \widetilde{\Theta}\left( \min\left\{ \frac{\epsilon^2}{\beta^{1/2}\,\mathrm{tr}(\mathbf{H})^{1/2}W}, \frac{\epsilon^2}{\beta^{3/2}W^2}, \frac{\epsilon}{\beta W} \right\}\right).$$

Only the first two are dominant, when

$$\epsilon \leq \beta^{1/2}W.$$

Therefore, given the condition that

$$N = \Theta\left( \max\left\{ \frac{\beta[\mathrm{tr}(\mathbf{H})]^{1/2}W^3}{\epsilon^4}, \frac{\beta^2 W^4}{\epsilon^4} \right\}\right),$$

the KL divergence can be upper bounded by $\epsilon^2$, i.e.,

$$\mathrm{KL}\left(\mu(\mathcal{P}^{\mathrm{alg}})^{N-1}\widetilde{\mathcal{P}}\|\pi\right) \lesssim \epsilon^2.$$

# G. Proof of Theorems in Section 5

In this section, we prove the sample complexity for RMD. To start with, we first verify Assumption A.5 for RMD. We have the following lemma:

**Lemma G.1.** For RMD, we can compute the Lipschitz constants of the strong and weak errors:

$$L_{w,\mathbf{x}} = \beta^3 h^5, L_{w,\mathbf{p}} = \beta^2 h^4,$$
$$L_{s,\mathbf{x}} = \beta^2 h^3, L_{s,\mathbf{p}} = \beta h^2.$$

Moreover, if $h \lesssim 1/(\beta^{7/6}\kappa^{1/2})$ in the strongly convex setting, or $h \lesssim N^{-1/4}\beta^{-1/2}$ in the general convex setting, Assumption A.5 holds for RMD.

## G.1. Proof of Theorem 5.2

In this section, we make a formal proof of Theorem 5.2. To start with, we first verify Assumption A.5, which is discussed in Lemma G.1:

$$h \lesssim 1/(\beta^{7/6}\kappa^{1/2}). \tag{G.1}$$

We apply Theorem A.6. Then, for any $n \leq N - 1$, we have

$$\mathrm{KL}\left(\nu_n^{\mathrm{aux}}\|\nu_n\right) \lesssim C\mathcal{W}_2^2(\mu,\nu) + A_w\left( \max_{1\leq i \leq n-1} \mathbb{E}_{\nu_i^{\mathrm{aux}}}\left[(\mathcal{E}^w)^2\right]\right) + A_s\left( \max_{1\leq i \leq n-1} \mathbb{E}_{\nu_i^{\mathrm{aux}}}\left[(\mathcal{E}^s)^2\right]\right).$$

Let $\nu = \pi$ be the invariant distribution of the underdamped Langevin. Then we can see $\nu_n = \pi$ for any $n$. We have

$$\mathrm{KL}\left(\nu_n^{\mathrm{aux}}\|\pi\right) \lesssim C\mathcal{W}_2^2(\mu,\pi) + A_w\left( \max_{1\leq i \leq n-1} \mathbb{E}_{\nu_i^{\mathrm{aux}}}\left[(\mathcal{E}^w)^2\right]\right) + A_s\left( \max_{1\leq i \leq n-1} \mathbb{E}_{\nu_i^{\mathrm{aux}}}\left[(\mathcal{E}^s)^2\right]\right). \tag{G.2}$$

Using Lemma 5.1, we can bound the strong and weak errors accordingly.

$$
\mathcal{E}^{\mathrm{w}}(\mathbf{x}, \mathbf{p}) \lesssim \beta^{3/2} h^4 \|\mathbf{p}\|_{\mathbf{H}} + \beta^2 h^5 \|\nabla V(\mathbf{x})\| + \beta^{3/2} \gamma^{1/2} h^{9/2} \sqrt{\mathrm{tr}(\mathbf{H})},
$$
$$
\mathcal{E}^{\mathrm{s}}(\mathbf{x}, \mathbf{p}) \lesssim \beta^{1/2} h^2 \|\mathbf{p}\|_{\mathbf{H}} + \beta h^3 \|\nabla V(\mathbf{x})\| + \beta^{1/2} \gamma^{1/2} h^{5/2} \sqrt{\mathrm{tr}(\mathbf{H})}.
$$

Thus, taking the expectation,

$$
\max_{1 \le i \le n-1} \mathbb{E}_{\nu_i^{\mathrm{aux}}}\big[(\mathcal{E}^w)^2\big] \lesssim \beta^3 h^8 \Big[\max_{1 \le i \le n-1} \mathbb{E}_{\nu_i^{\mathrm{aux}}}\big[\|\mathbf{p}\|_{\mathbf{H}}^2\big]\Big] + \beta^4 h^{10}\Big[\max_{1 \le i \le n-1} \mathbb{E}_{\nu_i^{\mathrm{aux}}}\big[\|\nabla V(\mathbf{x})\|^2\big]\Big] + \beta^3 \gamma h^9 \mathrm{tr}(\mathbf{H}).
$$
$$
\max_{1 \le i \le n-1} \mathbb{E}_{\nu_i^{\mathrm{aux}}}\big[(\mathcal{E}^s)^2\big] \lesssim \beta h^4 \Big[\max_{1 \le i \le n-1} \mathbb{E}_{\nu_i^{\mathrm{aux}}}\big[\|\mathbf{p}\|_{\mathbf{H}}^2\big]\Big] + \beta^2 h^6 \Big[\max_{1 \le i \le n-1} \mathbb{E}_{\nu_i^{\mathrm{aux}}}\big[\|\nabla V(\mathbf{x})\|^2\big]\Big] + \beta \gamma h^5 \mathrm{tr}(\mathbf{H}), \quad \text{(G.3)}
$$

Then, substituting (G.3) into (G.2), we have

$$
\begin{aligned}
\mathrm{KL}\big(\nu_n^{\mathrm{aux}}\|\pi\big) \lesssim{} & C\mathcal{W}_2^2(\mu, \pi) + A_w \beta^3 \gamma h^9 \, \mathrm{tr}(\mathbf{H}) + A_s \beta \gamma h^5 \, \mathrm{tr}(\mathbf{H}) \\
& + A_w \beta^3 h^8 \Big[\max_{1 \le i \le n-1} \mathbb{E}_{\nu_i^{\mathrm{aux}}}\big[\|\mathbf{p}\|_{\mathbf{H}}^2\big]\Big] + A_w \beta^4 h^{10}\Big[\max_{1 \le i \le n-1} \mathbb{E}_{\nu_i^{\mathrm{aux}}}\big[\|\nabla V(\mathbf{x})\|^2\big]\Big] \\
& + A_s \beta h^4 \Big[\max_{1 \le i \le n-1} \mathbb{E}_{\nu_i^{\mathrm{aux}}}\big[\|\mathbf{p}\|_{\mathbf{H}}^2\big]\Big] + A_s \beta^2 h^6 \Big[\max_{1 \le i \le n-1} \mathbb{E}_{\nu_i^{\mathrm{aux}}}\big[\|\nabla V(\mathbf{x})\|^2\big]\Big].
\end{aligned} \quad \text{(G.4)}
$$

In the strongly convex case,

$$
A_w = \frac{1}{\alpha h^2}, A_s = \frac{1}{\beta^{1/2} h} \log\Big(\frac{3\gamma}{\alpha h}\Big).
$$

When $h \le \beta^{-1/2} \kappa^{-1/3} \log^{-1/3}[(3\gamma)/(\alpha h)]$, we can drop the low-order terms and reduce the inequality to

$$
\begin{aligned}
\mathrm{KL}\big(\nu_n^{\mathrm{aux}}\|\pi\big) \lesssim{} & C\mathcal{W}_2^2(\mu, \pi) + \beta^{1/2} \gamma h^4 \log\Big(\frac{3\gamma}{\alpha h}\Big) \mathrm{tr}(\mathbf{H}) + \beta^{1/2} h^3 \log\Big(\frac{3\gamma}{\alpha h}\Big)\Big[\max_{1 \le i \le n-1} \mathbb{E}_{\nu_i^{\mathrm{aux}}}\big[\|\mathbf{p}\|_{\mathbf{H}}^2\big]\Big] \\
& + \beta^{3/2} h^5 \log\Big(\frac{3\gamma}{\alpha h}\Big)\Big[\max_{1 \le i \le n-1} \mathbb{E}_{\nu_i^{\mathrm{aux}}}\big[\|\nabla V(\mathbf{x})\|^2\big]\Big].
\end{aligned} \quad \text{(G.5)}
$$

Using Lemma 6.1, we have

$$
\begin{aligned}
\mathrm{KL}\big(\nu_n^{\mathrm{aux}}\|\pi\big) \lesssim{} & C\mathcal{W}_2^2(\mu, \pi) + \beta^{1/2} \gamma h^4 \log\Big(\frac{3\gamma}{\alpha h}\Big) \mathrm{tr}(\mathbf{H}) + \beta^{1/2} h^3 \log\Big(\frac{3\gamma}{\alpha h}\Big)\Big[\mathrm{tr}(\mathbf{H}) + \beta \max_{1 \le i \le n-1} \mathrm{KL}(\nu_i^{\mathrm{aux}}\|\pi)\Big] \\
& + \beta^{3/2} h^5 \log\Big(\frac{3\gamma}{\alpha h}\Big)\Big[\mathrm{tr}(\mathbf{H}) + \beta \max_{1 \le i \le n-1} \mathrm{KL}(\nu_i^{\mathrm{aux}}\|\pi)\Big] \\
\lesssim{} & C\mathcal{W}_2^2(\mu, \pi) + \beta h^4 \log\Big(\frac{3\gamma}{\alpha h}\Big) \mathrm{tr}(\mathbf{H}) + \beta^{3/2} h^3 \log\Big(\frac{3\gamma}{\alpha h}\Big)\Big[\max_{1 \le i \le n-1} \mathrm{KL}(\nu_i^{\mathrm{aux}}\|\pi)\Big].
\end{aligned}
$$

Since this inequality holds for any $n \le N - 1$, we have

$$
\max_{1 \le i \le n} \mathrm{KL}\big(\nu_i^{\mathrm{aux}}\|\pi\big) \lesssim C\mathcal{W}_2^2(\mu, \pi) + \beta h^4 \log\Big(\frac{3\gamma}{\alpha h}\Big) \mathrm{tr}(\mathbf{H}) + \beta^{3/2} h^3 \log\Big(\frac{3\gamma}{\alpha h}\Big) \max_{1 \le i \le n-1} \mathrm{KL}(\nu_i^{\mathrm{aux}}\|\pi).
$$

Using the condition $h \le \beta^{-1/2} \kappa^{-1/3} \log^{-1/3}[(3\gamma)/(\alpha h)]$, we know $\beta^{3/2} h^3 \log\big((3\gamma)/(\alpha h)\big) \lesssim 1$ for a small constant. Then, we can see that for any $n \le N - 1$,

$$
\max_{1 \le i \le n} \mathrm{KL}\big(\nu_i^{\mathrm{aux}}\|\pi\big) \lesssim C\mathcal{W}_2^2(\mu, \pi) + \beta h^4 \log\Big(\frac{3\gamma}{\alpha h}\Big) \mathrm{tr}(\mathbf{H}). \quad \text{(G.6)}
$$

Finally, using the second inequality in Theorem A.6 with $\mu = \pi$, we have

$$
\begin{aligned}
\mathrm{KL}\big(\mu(\mathcal{P}^{\mathrm{alg}})^{N-1}\widetilde{\mathcal{P}}\|\pi\big) \lesssim{} & C\mathcal{W}_2^2(\mu, \pi) + A_w\Big(\max_{1 \le i \le N-1} \mathbb{E}_{\nu_i^{\mathrm{aux}}}\big[(\mathcal{E}^w)^2\big]\Big) \\
& + A_s\Big(\max_{1 \le i \le N-1} \mathbb{E}_{\nu_i^{\mathrm{aux}}}\big[(\mathcal{E}^s)^2\big]\Big) + \max_{1 \le i \le N-1} \mathbb{E}_{\nu_i^{\mathrm{aux}}}\big[b^2\big]
\end{aligned}
$$

$$\lesssim C\mathcal{W}_2^2(\mu, \pi) + \beta h^4 \log\left(\frac{3\gamma}{\alpha h}\right) \mathrm{tr}(\mathbf{H}) + \beta^{1/2} h^3 \log\left(\frac{3\gamma}{\alpha h}\right) \left[\max_{1 \le i \le n-1} \mathbb{E}_{\nu_i^{\mathrm{aux}}}\left[\|\mathbf{p}\|_{\mathbf{H}}^2\right]\right]$$

$$+ \beta^{3/2} h^5 \log\left(\frac{3\gamma}{\alpha h}\right) \left[\max_{1 \le i \le n-1} \mathbb{E}_{\nu_i^{\mathrm{aux}}}\left[\|\nabla V(\mathbf{x})\|^2\right]\right]$$

$$+ \frac{\beta h^3}{\gamma}\left[\max_{1 \le i \le N-1} \mathbb{E}_{\nu_i^{\mathrm{aux}}}\left[\|\mathbf{p}\|_{\mathbf{H}}^2\right]\right] + \frac{\beta^2 h^5}{\gamma}\left[\max_{1 \le i \le N-1} \mathbb{E}_{\nu_i^{\mathrm{aux}}}\left[\|\nabla V(\mathbf{x})\|^2\right]\right],$$

$$\lesssim C\mathcal{W}_2^2(\mu, \pi) + \beta h^4 \log\left(\frac{3\gamma}{\alpha h}\right) \mathrm{tr}(\mathbf{H}) + \beta^{1/2} h^3 \log\left(\frac{3\gamma}{\alpha h}\right) \left[\max_{1 \le i \le N-1} \mathbb{E}_{\nu_i^{\mathrm{aux}}}\left[\|\mathbf{p}\|_{\mathbf{H}}^2\right]\right]$$

$$+ \beta^{3/2} h^5 \log\left(\frac{3\gamma}{\alpha h}\right) \left[\max_{1 \le i \le N-1} \mathbb{E}_{\nu_i^{\mathrm{aux}}}\left[\|\nabla V(\mathbf{x})\|^2\right]\right]$$

where the second inequality holds due to the same calculation as (G.5) and Lemma 4.2. The last inequality holds by dropping the lower-order term induced by the cross-regularity, since $\gamma \simeq \beta^{1/2}$. We substitute (G.3) into the inequality above and derive

$$\mathrm{KL}\left(\mu(\mathcal{P}^{\mathrm{alg}})^{N-1}\widetilde{\mathcal{P}}\|\pi\right) \lesssim C\mathcal{W}_2^2(\mu, \pi) + \beta h^4 \log\left(\frac{3\gamma}{\alpha h}\right) \mathrm{tr}(\mathbf{H})$$

$$+ \beta^{1/2} h^3 \log\left(\frac{3\gamma}{\alpha h}\right) \left[\mathrm{tr}(\mathbf{H}) + \beta \max_{1 \le i \le N-1} \mathrm{KL}(\nu_i^{\mathrm{aux}}\|\pi)\right]$$

$$+ \beta^{3/2} h^5 \log\left(\frac{3\gamma}{\alpha h}\right) \left[\mathrm{tr}(\mathbf{H}) + \beta \max_{1 \le i \le N-1} \mathrm{KL}(\nu_i^{\mathrm{aux}}\|\pi)\right]$$

$$\lesssim C\mathcal{W}_2^2(\mu, \pi) + \beta^{1/2} h^3 \log\left(\frac{3\gamma}{\alpha h}\right) \mathrm{tr}(\mathbf{H}) + \beta^{3/2} h^3 \log\left(\frac{3\gamma}{\alpha h}\right) \left[\max_{1 \le i \le N-1} \mathrm{KL}(\nu_i^{\mathrm{aux}}\|\pi)\right],$$

$$\lesssim C\mathcal{W}_2^2(\mu, \pi) + \beta^{1/2} h^3 \log\left(\frac{3\gamma}{\alpha h}\right) \mathrm{tr}(\mathbf{H})$$

$$+ \beta^{3/2} h^3 \log\left(\frac{3\gamma}{\alpha h}\right) \left[C\mathcal{W}_2^2(\mu, \pi) + \beta h^4 \log\left(\frac{3\gamma}{\alpha h}\right) \mathrm{tr}(\mathbf{H})\right],$$

where the last inequality holds due to (G.6), dropping the low-order terms. Again, dropping the low-order terms, we get the final bound of the KL divergence as:

$$\mathrm{KL}\left(\mu(\mathcal{P}^{\mathrm{alg}})^{N-1}\widetilde{\mathcal{P}}\|\pi\right) \lesssim C\mathcal{W}_2^2(\mu, \pi) + \beta^{1/2} h^3 \log\left(\frac{3\gamma}{\alpha h}\right) \mathrm{tr}(\mathbf{H}).$$

Let $Nh \simeq \alpha^{-1}\beta^{1/2} \log(\alpha W^2/\epsilon^2)$ such that $C\mathcal{W}_2^2(\mu, \pi) \lesssim \epsilon^2$. Moreover, let the stepsize be $h \simeq \beta^{-1/6} \mathrm{tr}^{-1/3}(\mathbf{H})\epsilon^{2/3}$, then the required sample complexity is

$$N \simeq \frac{\alpha^{-1}\beta^{1/2} \log(\alpha W^2/\epsilon^2)}{\beta^{-1/6} \mathrm{tr}^{-1/3}(\mathbf{H})\epsilon^{2/3}}$$

$$= \widetilde{\Theta}\left(\kappa\left[\beta^{-1} \mathrm{tr}(\mathbf{H})\right]^{1/3}\epsilon^{-2/3}\right).$$

Therefore, the KL divergence can be upper bounded by $\epsilon^2$, i.e.,

$$\mathrm{KL}\left(\mu(\mathcal{P}^{\mathrm{alg}})^{N-1}\widetilde{\mathcal{P}}\|\pi\right) \lesssim \epsilon^2.$$

The last thing that remains to be done is to check Assumption A.5. As shown in (G.1), we need $h \lesssim 1/(\beta^{7/6}\kappa^{1/2})$, which is not dominant when $\epsilon \le [\mathrm{tr}(\mathbf{H})]^{1/2}\beta^{-3/2}\kappa^{-3/4}$.

### G.2. Proof of Theorem 5.4

In this section, we now turn into the general convex (i.e. $\alpha = 0$) case. To fulfill Assumption A.5, as discussed in Lemma G.1, we need

$$h \lesssim \frac{1}{N^{1/4}\beta^{1/2}} \tag{G.7}$$

We apply Theorem 3.7. Same as (G.2), for any $n \leq N - 1$, we also have:

$$\mathrm{KL}\big(\nu_n^{\mathrm{aux}}\|\pi\big) \lesssim C\mathcal{W}_2^2(\mu, \pi) + A_w\Big(\max_{1 \leq i \leq n-1} \mathbb{E}_{\nu_i^{\mathrm{aux}}}\big[(\mathcal{E}^w)^2\big]\Big) + A_s\Big(\max_{1 \leq i \leq n-1} \mathbb{E}_{\nu_i^{\mathrm{aux}}}\big[(\mathcal{E}^s)^2\big]\Big). \tag{G.8}$$

And we still have the bound of strong and weak errors.

$$\mathcal{E}^{\mathrm{w}}(\mathbf{x}, \mathbf{p}) \lesssim \beta^{3/2}h^4\|\mathbf{p}\|_{\mathbf{H}} + \beta^2 h^5\|\nabla V(\mathbf{x})\| + \beta^{3/2}\gamma^{1/2}h^{9/2}\sqrt{\mathrm{tr}(\mathbf{H})},$$
$$\mathcal{E}^{\mathrm{s}}(\mathbf{x}, \mathbf{p}) \lesssim \beta^{1/2}h^2\|\mathbf{p}\|_{\mathbf{H}} + \beta h^3\|\nabla V(\mathbf{x})\| + \beta^{1/2}\gamma^{1/2}h^{5/2}\sqrt{\mathrm{tr}(\mathbf{H})}.$$

Thus, taking the expectation,

$$\max_{1 \leq i \leq n-1} \mathbb{E}_{\nu_i^{\mathrm{aux}}}\big[(\mathcal{E}^w)^2\big] \lesssim \beta^3 h^8\Big[\max_{1 \leq i \leq n-1} \mathbb{E}_{\nu_i^{\mathrm{aux}}}\big[\|\mathbf{p}\|_{\mathbf{H}}^2\big]\Big] + \beta^4 h^{10}\Big[\max_{1 \leq i \leq n-1} \mathbb{E}_{\nu_i^{\mathrm{aux}}}\big[\|\nabla V(\mathbf{x})\|^2\big]\Big] + \beta^3\gamma h^9\mathrm{tr}(\mathbf{H}),$$

$$\max_{1 \leq i \leq n-1} \mathbb{E}_{\nu_i^{\mathrm{aux}}}\big[(\mathcal{E}^s)^2\big] \lesssim \beta h^4\Big[\max_{1 \leq i \leq n-1} \mathbb{E}_{\nu_i^{\mathrm{aux}}}\big[\|\mathbf{p}\|_{\mathbf{H}}^2\big]\Big] + \beta^2 h^6\Big[\max_{1 \leq i \leq n-1} \mathbb{E}_{\nu_i^{\mathrm{aux}}}\big[\|\nabla V(\mathbf{x})\|^2\big]\Big] + \beta\gamma h^5\mathrm{tr}(\mathbf{H}). \tag{G.9}$$

Then, substituting (G.9) into (G.8), we have

$$\mathrm{KL}\big(\nu_n^{\mathrm{aux}}\|\pi\big) \lesssim C\mathcal{W}_2^2(\mu, \pi) + A_w\beta^3\gamma h^9\,\mathrm{tr}(\mathbf{H}) + A_s\beta\gamma h^5\,\mathrm{tr}(\mathbf{H})$$
$$+ A_w\beta^3 h^8\Big[\max_{1 \leq i \leq n-1} \mathbb{E}_{\nu_i^{\mathrm{aux}}}\big[\|\mathbf{p}\|_{\mathbf{H}}^2\big]\Big] + A_w\beta^4 h^{10}\Big[\max_{1 \leq i \leq n-1} \mathbb{E}_{\nu_i^{\mathrm{aux}}}\big[\|\nabla V(\mathbf{x})\|^2\big]\Big]$$
$$+ A_s\beta h^4\Big[\max_{1 \leq i \leq n-1} \mathbb{E}_{\nu_i^{\mathrm{aux}}}\big[\|\mathbf{p}\|_{\mathbf{H}}^2\big]\Big] + A_s\beta^2 h^6\Big[\max_{1 \leq i \leq n-1} \mathbb{E}_{\nu_i^{\mathrm{aux}}}\big[\|\nabla V(\mathbf{x})\|^2\big]\Big]. \tag{G.10}$$

In the general convex case,

$$A_w = \frac{N}{\beta^{1/2}h}, \qquad A_s = \frac{1}{\beta^{1/2}h}\log N + \beta^{1/2}Nh. \tag{G.11}$$

Moreover, assume in addition that

$$\log N \leq \beta N h^2,$$

so that

$$A_s = \frac{1}{\beta^{1/2}h}\log N + \beta^{1/2}Nh \lesssim \beta^{1/2}Nh.$$

Substituting $A_w = \frac{N}{\beta^{1/2}h}$ and $A_s \lesssim \beta^{1/2}Nh$ into (G.10), we obtain

$$\mathrm{KL}\big(\nu_n^{\mathrm{aux}}\|\pi\big) \lesssim C\mathcal{W}_2^2(\mu, \pi) + \frac{N}{\beta^{1/2}h}\beta^3\gamma h^9\,\mathrm{tr}(\mathbf{H}) + \beta^{1/2}Nh \cdot \beta\gamma h^5\,\mathrm{tr}(\mathbf{H})$$
$$+ \frac{N}{\beta^{1/2}h}\beta^3 h^8\Big[\max_{1 \leq i \leq n-1} \mathbb{E}_{\nu_i^{\mathrm{aux}}}\big[\|\mathbf{p}\|_{\mathbf{H}}^2\big]\Big] + \frac{N}{\beta^{1/2}h}\beta^4 h^{10}\Big[\max_{1 \leq i \leq n-1} \mathbb{E}_{\nu_i^{\mathrm{aux}}}\big[\|\nabla V(\mathbf{x})\|^2\big]\Big]$$
$$+ \beta^{1/2}Nh \cdot \beta h^4\Big[\max_{1 \leq i \leq n-1} \mathbb{E}_{\nu_i^{\mathrm{aux}}}\big[\|\mathbf{p}\|_{\mathbf{H}}^2\big]\Big] + \beta^{1/2}Nh \cdot \beta^2 h^6\Big[\max_{1 \leq i \leq n-1} \mathbb{E}_{\nu_i^{\mathrm{aux}}}\big[\|\nabla V(\mathbf{x})\|^2\big]\Big]$$
$$\lesssim C\mathcal{W}_2^2(\mu, \pi) + \beta^{5/2}\gamma N h^8\,\mathrm{tr}(\mathbf{H}) + \beta^{3/2}\gamma N h^6\,\mathrm{tr}(\mathbf{H})$$
$$+ \beta^{5/2}N h^7\Big[\max_{1 \leq i \leq n-1} \mathbb{E}_{\nu_i^{\mathrm{aux}}}\big[\|\mathbf{p}\|_{\mathbf{H}}^2\big]\Big] + \beta^{7/2}N h^9\Big[\max_{1 \leq i \leq n-1} \mathbb{E}_{\nu_i^{\mathrm{aux}}}\big[\|\nabla V(\mathbf{x})\|^2\big]\Big]$$
$$+ \beta^{3/2}N h^5\Big[\max_{1 \leq i \leq n-1} \mathbb{E}_{\nu_i^{\mathrm{aux}}}\big[\|\mathbf{p}\|_{\mathbf{H}}^2\big]\Big] + \beta^{5/2}N h^7\Big[\max_{1 \leq i \leq n-1} \mathbb{E}_{\nu_i^{\mathrm{aux}}}\big[\|\nabla V(\mathbf{x})\|^2\big]\Big].$$

Using $h \leq \beta^{-1/2}$ and $\gamma \simeq \Theta(\sqrt{\beta})$, we have $\beta h^2 \leq 1$, and hence

$$\beta^{5/2}\gamma N h^8\,\mathrm{tr}(\mathbf{H}) \lesssim \beta^{3/2}\gamma N h^6\,\mathrm{tr}(\mathbf{H}),$$

Similarly,

$$\beta^{5/2}Nh^7\Big[\max_{1\leq i\leq n-1}\mathbb{E}_{\nu_i^{\text{aux}}}\big[\|\mathbf{p}\|_{\mathbf{H}}^2\big]\Big] \leq (\beta h^2)\,\beta^{3/2}Nh^5\Big[\max_{1\leq i\leq n-1}\mathbb{E}_{\nu_i^{\text{aux}}}\big[\|\mathbf{p}\|_{\mathbf{H}}^2\big]\Big] \lesssim \beta^{3/2}Nh^5\Big[\max_{1\leq i\leq n-1}\mathbb{E}_{\nu_i^{\text{aux}}}\big[\|\mathbf{p}\|_{\mathbf{H}}^2\big]\Big],$$

and

$$\beta^{7/2}Nh^9\Big[\max_{1\leq i\leq n-1}\mathbb{E}_{\nu_i^{\text{aux}}}\big[\|\nabla V(\mathbf{x})\|^2\big]\Big] \lesssim \beta^{5/2}Nh^7\Big[\max_{1\leq i\leq n-1}\mathbb{E}_{\nu_i^{\text{aux}}}\big[\|\nabla V(\mathbf{x})\|^2\big]\Big].$$

Therefore, we can drop the low-order terms and reduce the inequality to

$$\text{KL}\big(\nu_n^{\text{aux}}\|\pi\big) \lesssim C\mathcal{W}_2^2(\mu,\pi) + \beta^{3/2}Nh^6\gamma\,\text{tr}(\mathbf{H}) + \beta^{3/2}Nh^5\Big[\max_{1\leq i\leq n-1}\mathbb{E}_{\nu_i^{\text{aux}}}\big[\|\mathbf{p}\|_{\mathbf{H}}^2\big]\Big]$$
$$+ \beta^{5/2}Nh^7\Big[\max_{1\leq i\leq n-1}\mathbb{E}_{\nu_i^{\text{aux}}}\big[\|\nabla V(\mathbf{x})\|^2\big]\Big]. \tag{G.12}$$

Using Lemma 6.1, we have

$$\text{KL}\big(\nu_n^{\text{aux}}\|\pi\big) \lesssim C\mathcal{W}_2^2(\mu,\pi) + \beta^{3/2}Nh^6\gamma\,\text{tr}(\mathbf{H}) + \beta^{3/2}Nh^5\Big[\text{tr}(\mathbf{H}) + \beta\max_{1\leq i\leq n-1}\text{KL}(\nu_i^{\text{aux}}\|\pi)\Big]$$
$$+ \beta^{5/2}Nh^7\Big[\text{tr}(\mathbf{H}) + \beta\max_{1\leq i\leq n-1}\text{KL}(\nu_i^{\text{aux}}\|\pi)\Big].$$

Since this inequality holds for any $n\leq N-1$, and we have $\gamma h\simeq\beta^{1/2}h\leq 1$, so we have $\beta^{3/2}\gamma h^6N\leq\beta^{3/2}h^5N$. Besides, we also have $\beta^{5/2}Nh^7\leq\beta^{3/2}Nh^5$ because of $\beta h^2\leq 1$, so to drop all of the lower-order term, we have:

$$\max_{1\leq i\leq n}\text{KL}\big(\nu_i^{\text{aux}}\|\pi\big) \lesssim C\mathcal{W}_2^2(\mu,\pi) + \beta^{3/2}Nh^5\,\text{tr}(\mathbf{H}) + \beta^{5/2}Nh^5\max_{1\leq i\leq n-1}\text{KL}(\nu_i^{\text{aux}}\|\pi).$$

Using the condition

$$\beta^{5/2}Nh^5 \lesssim c < 1, \tag{G.13}$$

we can see that for any $n\leq N-1$,

$$\max_{1\leq i\leq n}\text{KL}\big(\nu_i^{\text{aux}}\|\pi\big) \lesssim C\mathcal{W}_2^2(\mu,\pi) + \beta^{3/2}Nh^5\,\text{tr}(\mathbf{H}). \tag{G.14}$$

Finally, using the second inequality in Corollary 3.8 with $\nu=\pi$, we have

$$\text{KL}\big(\mu(\mathcal{P}^{\text{alg}})^{N-1}\widetilde{\mathcal{P}}\|\pi\big) \lesssim C\mathcal{W}_2^2(\mu,\pi) + A_w\big(\bar{\mathcal{E}}^w\big)^2 + A_s\big(\bar{\mathcal{E}}^s\big)^2 + \bar{b}^2$$
$$\lesssim C\mathcal{W}_2^2(\mu,\pi) + A_w\Big(\max_{1\leq i\leq N-1}\mathbb{E}_{\nu_i^{\text{aux}}}\big[(\mathcal{E}^w)^2\big]\Big) + A_s\Big(\max_{1\leq i\leq N-1}\mathbb{E}_{\nu_i^{\text{aux}}}\big[(\mathcal{E}^s)^2\big]\Big) + \max_{1\leq i\leq N-1}\mathbb{E}_{\nu_i^{\text{aux}}}\big[b^2\big]$$
$$\lesssim C\mathcal{W}_2^2(\mu,\pi) + A_w\beta^3\gamma h^9\,\text{tr}(\mathbf{H}) + A_s\beta\gamma h^5\,\text{tr}(\mathbf{H})$$
$$+ A_w\beta^3 h^8\Big[\max_{1\leq i\leq N-1}\mathbb{E}_{\nu_i^{\text{aux}}}\big[\|\mathbf{p}\|_{\mathbf{H}}^2\big]\Big] + A_w\beta^4 h^{10}\Big[\max_{1\leq i\leq N-1}\mathbb{E}_{\nu_i^{\text{aux}}}\big[\|\nabla V(\mathbf{x})\|^2\big]\Big]$$
$$+ A_s\beta h^4\Big[\max_{1\leq i\leq N-1}\mathbb{E}_{\nu_i^{\text{aux}}}\big[\|\mathbf{p}\|_{\mathbf{H}}^2\big]\Big] + A_s\beta^2 h^6\Big[\max_{1\leq i\leq N-1}\mathbb{E}_{\nu_i^{\text{aux}}}\big[\|\nabla V(\mathbf{x})\|^2\big]\Big] + \max_{1\leq i\leq N-1}\mathbb{E}_{\nu_i^{\text{aux}}}\big[b^2\big],$$

where the second inequality holds due to Corollary 3.8, and the third inequality holds due to the same calculation as above. Moreover, by Lemma 4.2, we have

$$\max_{1\leq i\leq N-1}\mathbb{E}_{\nu_i^{\text{aux}}}\big[b^2\big] \lesssim \frac{\beta h^3}{\gamma}\Big[\max_{1\leq i\leq N-1}\mathbb{E}_{\nu_i^{\text{aux}}}\big[\|\mathbf{p}\|_{\mathbf{H}}^2\big]\Big] + \frac{\beta^2 h^5}{\gamma}\Big[\max_{1\leq i\leq N-1}\mathbb{E}_{\nu_i^{\text{aux}}}\big[\|\nabla V(\mathbf{x})\|^2\big]\Big] + \beta h^4\,\text{tr}(\mathbf{H}).$$

We substitute the bound above into the inequality and obtain

$$\text{KL}\big(\mu(\mathcal{P}^{\text{alg}})^{N-1}\widetilde{\mathcal{P}}\|\pi\big) \lesssim C\mathcal{W}_2^2(\mu,\pi) + A_w\beta^3\gamma h^9\,\text{tr}(\mathbf{H}) + A_s\beta\gamma h^5\,\text{tr}(\mathbf{H})$$

$$+ A_w \beta^3 h^8 \Big[ \max_{1 \le i \le N-1} \mathbb{E}_{\nu_i^{\mathrm{aux}}} \big[ \|\mathbf{p}\|_{\mathbf{H}}^2 \big] \Big] + A_w \beta^4 h^{10} \Big[ \max_{1 \le i \le N-1} \mathbb{E}_{\nu_i^{\mathrm{aux}}} \big[ \|\nabla V(\mathbf{x})\|^2 \big] \Big]$$

$$+ A_s \beta h^4 \Big[ \max_{1 \le i \le N-1} \mathbb{E}_{\nu_i^{\mathrm{aux}}} \big[ \|\mathbf{p}\|_{\mathbf{H}}^2 \big] \Big] + A_s \beta^2 h^6 \Big[ \max_{1 \le i \le N-1} \mathbb{E}_{\nu_i^{\mathrm{aux}}} \big[ \|\nabla V(\mathbf{x})\|^2 \big] \Big]$$

$$+ \frac{\beta h^3}{\gamma} \Big[ \max_{1 \le i \le N-1} \mathbb{E}_{\nu_i^{\mathrm{aux}}} \big[ \|\mathbf{p}\|_{\mathbf{H}}^2 \big] \Big] + \frac{\beta^2 h^5}{\gamma} \Big[ \max_{1 \le i \le N-1} \mathbb{E}_{\nu_i^{\mathrm{aux}}} \big[ \|\nabla V(\mathbf{x})\|^2 \big] \Big] + \beta h^4 \operatorname{tr}(\mathbf{H}).$$

Using Lemma 6.1, we have

$$\mathrm{KL}\big( \mu (\mathcal{P}^{\mathrm{alg}})^{N-1} \widetilde{\mathcal{P}} \| \pi \big) \lesssim C \mathcal{W}_2^2(\mu, \pi) + A_w \beta^3 \gamma h^9 \operatorname{tr}(\mathbf{H}) + A_s \beta \gamma h^5 \operatorname{tr}(\mathbf{H})$$

$$+ A_w \beta^3 h^8 \Big[ \operatorname{tr}(\mathbf{H}) + \beta \max_{1 \le i \le N-1} \mathrm{KL}(\nu_i^{\mathrm{aux}} \| \pi) \Big] + A_w \beta^4 h^{10} \Big[ \operatorname{tr}(\mathbf{H}) + \beta \max_{1 \le i \le N-1} \mathrm{KL}(\nu_i^{\mathrm{aux}} \| \pi) \Big]$$

$$+ A_s \beta h^4 \Big[ \operatorname{tr}(\mathbf{H}) + \beta \max_{1 \le i \le N-1} \mathrm{KL}(\nu_i^{\mathrm{aux}} \| \pi) \Big] + A_s \beta^2 h^6 \Big[ \operatorname{tr}(\mathbf{H}) + \beta \max_{1 \le i \le N-1} \mathrm{KL}(\nu_i^{\mathrm{aux}} \| \pi) \Big]$$

$$+ \frac{\beta h^3}{\gamma} \Big[ \operatorname{tr}(\mathbf{H}) + \beta \max_{1 \le i \le N-1} \mathrm{KL}(\nu_i^{\mathrm{aux}} \| \pi) \Big] + \frac{\beta^2 h^5}{\gamma} \Big[ \operatorname{tr}(\mathbf{H}) + \beta \max_{1 \le i \le N-1} \mathrm{KL}(\nu_i^{\mathrm{aux}} \| \pi) \Big] + \beta h^4 \operatorname{tr}(\mathbf{H}).$$

Since $\gamma = O(\beta^{1/2})$, and $\beta h^2 \le 1$. So (G.11) shows that $A_w \beta h^2 \le A_s$. So $A_w \beta^4 h^{10} \le A_w \beta^3 h^8 \le A_s \beta^2 h^6 \le A_s \beta h^4$, and $\beta^2 h^5 / \gamma \simeq \beta^{3/2} h^5 \le \beta^{3/2} h^5 N$. And by the condition (G.13) we know that $\beta^{5/2} N h^5 \lesssim 1$. By $N \beta h^2 \ge \log N \ge 1$, we have $\beta h^3 / \gamma \le \beta^{3/2} h^5 N$.

Dropping the low-order terms, we finally obtain

$$\mathrm{KL}\big( \mu (\mathcal{P}^{\mathrm{alg}})^{N-1} \widetilde{\mathcal{P}} \| \pi \big) \lesssim C \mathcal{W}_2^2(\mu, \pi) + \beta^{3/2} N h^5 \operatorname{tr}(\mathbf{H}).$$

Here $C \simeq \gamma / (Nh) \simeq \sqrt{\beta}/Nh$. Let $Nh \simeq \beta^{1/2} W^2 / (\epsilon^2)$ such that $C \mathcal{W}_2^2(\mu, \pi) \lesssim \epsilon^2$.

While we also need $\beta^{5/2} N h^5 \lesssim c < 1$ required in (G.13), and $\beta^{3/2} N h^5 \operatorname{tr}(\mathbf{H}) \simeq \epsilon^2$. Then, we have:

$$\beta^{3/2} h^4 \operatorname{tr}(\mathbf{H}) \lesssim \frac{\epsilon^4}{\beta^{1/2} W^2}, \ \beta^{5/2} h^4 \lesssim \frac{\epsilon^2}{\beta^{1/2} W^2}$$

Combined with the condition discussed in (G.7), which is:

$$h \lesssim \frac{1}{N^{1/4} \beta^{1/2}}.$$

We could finally get:

$$h \simeq \min \left\{ \frac{\epsilon}{\beta^{1/2} \big( \operatorname{tr}(\mathbf{H}) \big)^{1/4} W^{1/2}}, \frac{\epsilon^{1/2}}{\beta^{3/4} W^{1/2}}, \frac{\epsilon^{2/3}}{\beta^{5/6} W^{2/3}} \right\}.$$

Only the last one is dominant, when

$$\epsilon \le \min \left\{ \sqrt{\beta} W, \frac{\big( \operatorname{tr}(\mathbf{H}) \big)^{3/4}}{\beta W^{1/2}} \right\}.$$

Therefore, given the condition that

$$N = \Theta \left( \frac{\beta \big( \operatorname{tr}(\mathbf{H}) \big)^{1/4} W^{5/2}}{\epsilon^3} \right),$$

the KL divergence can be upper bounded by $\epsilon^2$, i.e.,

$$\mathrm{KL}\big( \mu (\mathcal{P}^{\mathrm{alg}})^{N-1} \widetilde{\mathcal{P}} \| \pi \big) \lesssim \epsilon^2.$$

## H. Verification of Assumption A.5

In this section, we verify Assumption A.5 for ULMC and RMD, i.e., Lemma F.1 and Lemma G.1.

## H.1. Proof of Lemma F.1

Using Lemma 4.1, we can see the Lipschitz constants of the strong and weak errors:

$$L_{w,\mathbf{x}} = L_{s,\mathbf{x}} = \beta^2 h^3,$$
$$L_{w,\mathbf{p}} = L_{s,\mathbf{p}} = \beta h^2.$$

Moreover, using Lemma 4.2, the Lipschitz constants of $b$ can be derived as

$$L_{b,\mathbf{x}} = \beta^{3/4} h^{3/2}, L_{b,\mathbf{p}} = \beta^{5/4} h^{5/2}.$$

We need to check

$$\max\left\{ \frac{(L_{w,\mathbf{x}} + (\gamma + \eta_{nh}^{\mathbf{P}})L_{w,\mathbf{p}})^2}{(\omega_+ + \eta_{nh}^{\mathbf{P}})(\gamma + \eta_{nh}^{\mathbf{P}})^2 h}, \left(1 + \frac{\beta^2 h}{(\gamma + \eta_{nh}^{\mathbf{P}})^2(\omega_+ + \eta_{nh}^{\mathbf{P}})}\right)\frac{(L_{s,\mathbf{x}} + (\gamma + \eta_{nh}^{\mathbf{P}})L_{s,\mathbf{p}})^2}{(\gamma + \eta_{nh}^{\mathbf{P}})^2} \right\} \lesssim 1 - L_n, \qquad \text{(H.1)}$$

and

$$(L_{b,\mathbf{x}} + h^{-1}L_{b,\mathbf{p}})^2 \gamma h^3 \lesssim 1. \qquad \text{(H.2)}$$

Assume $\beta h^2 \lesssim 1$. Note that $\eta_{nh}^{\mathbf{P}} \lesssim 1/h$, and

$$1 - L_n = 1 - \exp\left(-c\int_{nh}^{(n+1)h}(\omega_+ + \eta_t^{\mathbf{P}})\,\mathrm{d}t\right)$$
$$\simeq \int_{nh}^{(n+1)h}(\omega_+ + \eta_t^{\mathbf{P}})\,\mathrm{d}t$$
$$\simeq (\omega_+ + \eta_{nh}^{\mathbf{P}})h.$$

Moreover, $\beta/(\gamma + \eta_{nh}^{\mathbf{P}})^2 \lesssim 1$, $\beta h^2 \lesssim 1$, $(\omega_+ + \eta_{nh}^{\mathbf{P}})h \lesssim 1$. Note that $L_{s,\mathbf{x}} = L_{w,\mathbf{x}}$, $L_{s,\mathbf{p}} = L_{w,\mathbf{p}}$ in this setting. Thus, the second term related with $L_{w,\mathbf{p}}$ and $L_{s,\mathbf{p}}$ is not dominant in (H.1). We only need to consider the first term.

$$\frac{(L_{w,\mathbf{x}} + (\gamma + \eta_{nh}^{\mathbf{P}})L_{w,\mathbf{p}})^2}{(\omega_+ + \eta_{nh}^{\mathbf{P}})(\gamma + \eta_{nh}^{\mathbf{P}})^2 h} \lesssim \frac{(L_{w,\mathbf{x}})^2}{(\omega_+ + \eta_{nh}^{\mathbf{P}})(\gamma + \eta_{nh}^{\mathbf{P}})^2 h} + \frac{(L_{w,\mathbf{p}})^2}{(\omega_+ + \eta_{nh}^{\mathbf{P}})h}$$
$$\lesssim \frac{\beta^4 h^6}{(\omega_+ + \eta_{nh}^{\mathbf{P}})(\gamma + \eta_{nh}^{\mathbf{P}})^2 h} + \frac{\beta^2 h^4}{(\omega_+ + \eta_{nh}^{\mathbf{P}})h}$$
$$\lesssim \frac{\beta^2 h^3}{(\omega_+ + \eta_{nh}^{\mathbf{P}})},$$

where the last inequality holds due to $\gamma \simeq \sqrt{\beta}$ and $\beta h^2 \lesssim 1$. To show (H.1), it suffices that

$$\frac{\beta^2 h^2}{\omega_+ + \eta_{nh}^{\mathbf{P}}} \lesssim (\omega_+ + \eta_{nh}^{\mathbf{P}})h,$$

which is equivalent to $\beta^2 h^2 \lesssim (\omega_+ + \eta_{nh}^{\mathbf{P}})^2$.

**Strongly Convex.** In this case, $\omega \simeq \alpha/\sqrt{\beta}$. We only need $\beta^2 h^2 \lesssim \omega_+^2$. This induces:

$$h \lesssim \alpha\beta^{-3/2} = 1/(\beta^{1/2}\kappa). \qquad \text{(H.3)}$$

**General Convex.** In this case, $\omega = 0$. We need $\beta^2 h^2 \lesssim (\eta_{nh}^{\mathbf{P}})^2$. Recall the definition of

$$\eta_{nh}^{\mathbf{P}} = \frac{c_0\omega}{\exp(\omega(Nh - nh + Ah)) - 1}.$$

It takes the minimum when $n = 0$. Thus, we have $\eta_{nh}^{\mathbf{P}} \gtrsim 1/(Nh)$. Thus, we only need $\beta^2 h^2 \lesssim 1/(Nh)^2$, which is equivalent to

$$h \lesssim N^{-1/2}\beta^{-1/2}. \qquad \text{(H.4)}$$

Finally, we consider (H.2). In both cases, we only need $\beta^4 h^6 \lesssim 1$. This induces $h \lesssim \beta^{-2/3}$. In both cases, this is not the dominant rate.

## H.2. Proof of Lemma G.1

Using Lemma 5.1, we can see the Lipschitz constants of the strong and weak errors:

$$L_{w,\mathbf{x}} = \beta^3 h^5, L_{w,\mathbf{p}} = \beta^2 h^4,$$
$$L_{s,\mathbf{x}} = \beta^2 h^3, L_{s,\mathbf{p}} = \beta h^2.$$

We need to check

$$\max\left\{ \frac{(L_{w,\mathbf{x}} + (\gamma + \eta_{nh}^{\mathbf{P}})L_{w,\mathbf{p}})^2}{(\omega_+ + \eta_{nh}^{\mathbf{P}})(\gamma + \eta_{nh}^{\mathbf{P}})^2 h}, \left(1 + \frac{\beta^2 h}{(\gamma + \eta_{nh}^{\mathbf{P}})^2(\omega_+ + \eta_{nh}^{\mathbf{P}})}\right) \frac{(L_{s,\mathbf{x}} + (\gamma + \eta_{nh}^{\mathbf{P}})L_{s,\mathbf{p}})^2}{(\gamma + \eta_{nh}^{\mathbf{P}})^2} \right\} \lesssim 1 - L_n, \qquad \text{(H.5)}$$

and

$$(L_{b,\mathbf{x}} + h^{-1}L_{b,\mathbf{p}})^2 \gamma h^3 \lesssim 1. \qquad \text{(H.6)}$$

Assume $\beta h^2 \lesssim 1$. Note that $\eta_{nh}^{\mathbf{P}} \lesssim 1/h$, and

$$1 - L_n = 1 - \exp\left(-c \int_{nh}^{(n+1)h} (\omega_+ + \eta_t^{\mathbf{P}}) \, \mathrm{d}t\right)$$
$$\simeq \int_{nh}^{(n+1)h} (\omega_+ + \eta_t^{\mathbf{P}}) \, \mathrm{d}t$$
$$\simeq (\omega_+ + \eta_{nh}^{\mathbf{P}})h.$$

For the first term in (H.5), we can compute as

$$\frac{(L_{w,\mathbf{x}} + (\gamma + \eta_{nh}^{\mathbf{P}})L_{w,\mathbf{p}})^2}{(\omega_+ + \eta_{nh}^{\mathbf{P}})(\gamma + \eta_{nh}^{\mathbf{P}})^2 h} \lesssim \frac{(L_{w,\mathbf{x}})^2}{(\omega_+ + \eta_{nh}^{\mathbf{P}})(\gamma + \eta_{nh}^{\mathbf{P}})^2 h} + \frac{(L_{w,\mathbf{p}})^2}{(\omega_+ + \eta_{nh}^{\mathbf{P}})h}$$
$$\lesssim \frac{\beta^6 h^{10}}{(\omega_+ + \eta_{nh}^{\mathbf{P}})(\gamma + \eta_{nh}^{\mathbf{P}})^2 h} + \frac{\beta^4 h^8}{(\omega_+ + \eta_{nh}^{\mathbf{P}})h}$$
$$\lesssim \frac{\beta^4 h^7}{(\omega_+ + \eta_{nh}^{\mathbf{P}})},$$

where the last inequality holds due to $\gamma \simeq \sqrt{\beta}$ and $\beta h^2 \lesssim 1$. For the second term in (H.5), we have

$$\left(1 + \frac{\beta^2 h}{(\gamma + \eta_{nh}^{\mathbf{P}})^2(\omega_+ + \eta_{nh}^{\mathbf{P}})}\right) \frac{(L_{s,\mathbf{x}} + (\gamma + \eta_{nh}^{\mathbf{P}})L_{s,\mathbf{p}})^2}{(\gamma + \eta_{nh}^{\mathbf{P}})^2}$$
$$\lesssim \frac{(L_{s,\mathbf{x}} + (\gamma + \eta_{nh}^{\mathbf{P}})L_{s,\mathbf{p}})^2}{(\gamma + \eta_{nh}^{\mathbf{P}})^2} + \frac{\beta^2 h(L_{s,\mathbf{x}} + (\gamma + \eta_{nh}^{\mathbf{P}})L_{s,\mathbf{p}})^2}{(\gamma + \eta_{nh}^{\mathbf{P}})^4(\omega_+ + \eta_{nh}^{\mathbf{P}})},$$
$$\lesssim \frac{(L_{s,\mathbf{x}})^2}{(\gamma + \eta_{nh}^{\mathbf{P}})^2} + (L_{s,\mathbf{p}})^2 + \frac{\beta^2 h(L_{s,\mathbf{x}})^2}{(\gamma + \eta_{nh}^{\mathbf{P}})^4(\omega_+ + \eta_{nh}^{\mathbf{P}})} + \frac{\beta^2 h(L_{s,\mathbf{p}})^2}{(\gamma + \eta_{nh}^{\mathbf{P}})^2(\omega_+ + \eta_{nh}^{\mathbf{P}})}$$
$$\lesssim \frac{\beta^4 h^6}{(\gamma + \eta_{nh}^{\mathbf{P}})^2} + \beta^2 h^4 + \frac{\beta^6 h^7}{(\gamma + \eta_{nh}^{\mathbf{P}})^4(\omega_+ + \eta_{nh}^{\mathbf{P}})} + \frac{\beta^4 h^5}{(\gamma + \eta_{nh}^{\mathbf{P}})^2(\omega_+ + \eta_{nh}^{\mathbf{P}})}$$
$$\lesssim \beta^2 h^4 + \frac{\beta^3 h^5}{(\omega_+ + \eta_{nh}^{\mathbf{P}})},$$

where we use $\beta h^2 \lesssim 1$ and $\gamma \simeq \sqrt{\beta}$. To show (H.5), it suffices that

$$\max\left\{\frac{\beta^4 h^7}{\omega_+ + \eta_{nh}^{\mathbf{P}}}, \beta^2 h^4, \frac{\beta^3 h^5}{\omega_+ + \eta_{nh}^{\mathbf{P}}}\right\} \lesssim (\omega_+ + \eta_{nh}^{\mathbf{P}})h.$$

**Strongly Convex.** In this case, $\omega \simeq \alpha/\sqrt{\beta}$. We only need

$$\beta^3 h^4 \lesssim \alpha^2/\beta, \beta^2 h^3 \lesssim \alpha/\sqrt{\beta}.$$

The solution to this is

$$h \lesssim \alpha^{1/2}\beta^{-1} = 1/(\beta^{1/2}\kappa^{1/2}), h \lesssim 1/(\beta^{7/6}\kappa^{1/3}).$$

The condition in this case suffices that

$$h \lesssim 1/(\beta^{7/6}\kappa^{1/2}). \tag{H.7}$$

**General Convex.** In this case, $\omega = 0$. We need

$$\beta^2 h^3 \lesssim \eta_{nh}^{\mathbf{P}}, \beta^3 h^4 \lesssim (\eta_{nh}^{\mathbf{P}})^2.$$

Recall the definition of

$$\eta_{nh}^{\mathbf{P}} = \frac{c_0\omega}{\exp(\omega(Nh - nh + Ah)) - 1}.$$

It takes the minimum when $n = 0$. Thus, we have $\eta_{nh}^{\mathbf{P}} \gtrsim 1/(Nh)$. Thus, we have

$$\beta^2 h^3 \lesssim \frac{1}{Nh}, \ \beta^3 h^4 \lesssim \frac{1}{N^2 h^2},$$

which is equivalent to

$$h \lesssim N^{-1/4}\beta^{-1/2} \ \wedge \ N^{-1/3}\beta^{-1/2}. \tag{H.8}$$

# I. Auxiliary Lemmas

**Lemma I.1** (Talagrand's $T_2$ inequality). Let $\pi(\mathbf{x}) \propto \exp(-V(\mathbf{x}))$. Suppose $V$ is $\alpha$-strongly convex for $\alpha > 0$. Then for any distribution $\mu$, the Wasserstein 2-distance can be bounded by the KL divergence, satisfying

$$W_2^2(\mu, \pi) \leq \frac{2}{\alpha}\mathrm{KL}(\mu\|\pi).$$

**Lemma I.2** (Stein's Identity). Let $V : \mathbb{R}^d \to \mathbb{R}$ be a differentiable function, and $\mathbf{F} : \mathbb{R}^d \to \mathbb{R}^d$ be a differentiable vector field. Let the distribution $\pi$ be defined as $\pi \propto \exp(-V)$. Then

$$\mathbb{E}_{\mathbf{x}\sim\pi}[\langle\nabla V, \mathbf{F}\rangle] = \mathbb{E}_\pi[\nabla \cdot \mathbf{F}].$$

**Lemma I.3** (Donsker-Varadhan's variational formula). Let $(\mathcal{X}, \mathcal{F}, P_0)$ be a probability space and $U(x)$ be a measurable function. Then for any distribution $P$ on $(\mathcal{X}, \mathcal{F})$, we have

$$\mathbb{E}_{x\sim P}[U(x)] + \mathrm{KL}(P\|P_0) \geq -\log \mathbb{E}_{x\sim P_0} \exp(-U(x)),$$

and the infimum is attained when $P(x) \propto P_0(x)\exp(-U(x))$.

**Lemma I.4.** Let $\lambda$ be a scalar such that $0 < \lambda \leq 1/(4\beta)$. Then the norm of $\nabla V$ satisfies the following inequality:

$$\log \mathbb{E}_\pi[\exp(\lambda\|\nabla V\|^2)] \leq 2\lambda \operatorname{tr}(\mathbf{H}).$$

*Proof of Lemma I.4.* Due to the Taylor expansion, $\mathbb{E}_\pi[\exp(\lambda\|\nabla V\|^2)]$ satisfies

$$\mathbb{E}_\pi[\exp(\lambda\|\nabla V\|^2)] = \sum_{k=0}^{\infty} \frac{\lambda^k \mathbb{E}_\pi[\|\nabla V\|^{2k}]}{k!}, \tag{I.1}$$

so it suffices to bound $M_k := \mathbb{E}_\pi[\|\nabla V\|^{2k}]$. For $k \geq 1$, we apply Stein's identity (Lemma I.2) with $\mathbf{F}(\mathbf{x}) = \|\nabla V(\mathbf{x})\|^{2k-2}\nabla V(\mathbf{x})$ and obtain

$$M_k = \mathbb{E}_\pi\big[\langle\|\nabla V(\mathbf{x})\|^{2k-2}\nabla V(\mathbf{x}), \nabla V(\mathbf{x})\rangle\big]$$

$$= \mathbb{E}_\pi\left[\nabla \cdot \left(\|\nabla V(\mathbf{x})\|^{2k-2}\nabla V(\mathbf{x})\right)\right]$$

$$= (2k-2)\cdot\underbrace{\mathbb{E}_\pi\left[\|\nabla V(\mathbf{x})\|^{2k-4}\langle\nabla^2 V(\mathbf{x})\cdot\nabla V(\mathbf{x}),\nabla V(\mathbf{x})\rangle\right]}_{I_1} + \underbrace{\mathbb{E}_\pi\left[\|\nabla V(\mathbf{x})\|^{2k-2}\cdot\Delta V(\mathbf{x})\right]}_{I_2}. \tag{I.2}$$

For the term $I_1$, when $k \geq 2$, note that $\nabla^2 V(\mathbf{x}) \preceq \beta\mathbf{I}$ due to the $\beta$-smoothness of $V$, so

$$\langle\nabla^2 V(\mathbf{x})\cdot\nabla V(\mathbf{x}),\nabla V(\mathbf{x})\rangle \leq \beta\|\nabla V(\mathbf{x})\|^2.$$

Therefore, the upper bound of $I_1$ is

$$I_1 \leq \beta \cdot \mathbb{E}_\pi[\|\nabla V(\mathbf{x})\|^{2k-2}] = \beta M_{k-1}. \tag{I.3}$$

For the term $I_2$, since $\nabla^2 V(\mathbf{x}) \preceq \mathbf{H}$, we have $\Delta V(\mathbf{x}) = \mathrm{tr}(\nabla^2 V(\mathbf{x})) \leq \mathrm{tr}(\mathbf{H})$, so the upper bound of $I_2$ is

$$I_2 \leq \mathbb{E}_\pi\left[\|\nabla V(\mathbf{x})\|^{2k-2}\cdot\mathrm{tr}(\mathbf{H})\right] = \mathrm{tr}(\mathbf{H})M_{k-1}. \tag{I.4}$$

Plugging (I.3) and (I.4) into (I.2), we have

$$M_k \leq [(2k-2)\beta + \mathrm{tr}(\mathbf{H})]M_{k-1}. \tag{I.5}$$

Since $M_0 = 1$, by recursively using (I.5), we have

$$M_k \leq \prod_{j=0}^{k-1}[2j\beta + \mathrm{tr}(\mathbf{H})] = (2\beta)^k\prod_{j=0}^{k-1}\left[j + \frac{\mathrm{tr}(\mathbf{H})}{2\beta}\right]. \tag{I.6}$$

Plugging (I.6) into (I.1), we have

$$\mathbb{E}_\pi[\exp(\lambda\|\nabla V(\mathbf{x})\|^2)] \leq \sum_{k=0}^{\infty}\frac{(2\beta\lambda)^k}{k!}\prod_{j=0}^{k-1}\left[j + \frac{\mathrm{tr}(\mathbf{H})}{2\beta}\right]. \tag{I.7}$$

We make the crucial observation that when $z \in (0,1)$ the Taylor expansion of $(1-z)^{-s}$ is

$$(1-z)^{-s} = \sum_{k=0}^{\infty}\frac{z^k}{k!}\prod_{j=0}^{k-1}(j+s).$$

Setting $z = 2\beta\lambda$ and $s = \mathrm{tr}(\mathbf{H})/(2\beta)$, under the condition $\beta\lambda < 1/4$, we have

$$\mathbb{E}_\pi[\exp(\lambda\|\nabla V(\mathbf{x})\|^2)] \leq (1-2\beta\lambda)^{-\mathrm{tr}(\mathbf{H})/2\beta}.$$

Taking the logarithm on both sides, we have

$$\log\mathbb{E}_\pi[\exp(\lambda\|\nabla V(\mathbf{x})\|^2)] \leq \frac{\mathrm{tr}(\mathbf{H})}{2\beta}\log\left(1/(1-2\beta\lambda)\right) \leq 2\lambda\,\mathrm{tr}(\mathbf{H}).$$

where the last inequality holds because $\log(1/(1-z)) \leq 2z$ for $0 < z \leq 1/2$.

$\square$

**Lemma I.5.** Suppose that $(\mathbf{x},\mathbf{p}) \sim \pi$, i.e., $\mathbf{p} \sim \mathcal{N}(\mathbf{0},\mathbf{I})$. Then for $\lambda < 1/(4\beta)$, the following inequality holds:

$$\log\mathbb{E}[\exp(\lambda\|\mathbf{p}\|_{\mathbf{H}}^2)] \leq 2\lambda\,\mathrm{tr}(\mathbf{H}).$$

*Proof of Lemma I.5.* Let the eigenvalue decomposition of $\mathbf{H}$ be

$$\mathbf{H} = \sum_{i=1}^{d}\mu_i\mathbf{u}_i\mathbf{u}_i^\top,$$

then $\|\mathbf{p}\|_{\mathbf{H}}^2$ can be written as

$$\|\mathbf{p}\|_{\mathbf{H}}^2 = \sum_{i=1}^d \mu_i q_i^2, \quad \text{where} \quad q_i = \mathbf{u}_i^\top \mathbf{p}.$$

Note that $\{q_i\}$ are i.i.d. standard normal random variables because $\{\mathbf{u}_i\}$ form an orthonormal basis of $\mathbb{R}^d$. Therefore, for $\lambda \le 1/(4\beta)$, we have

$$\log \mathbb{E}[\exp(\lambda \|\mathbf{p}\|_{\mathbf{H}}^2)] = \sum_{i=1}^d \log \mathbb{E}[\exp(\lambda \mu_i q_i^2)] = -\frac{1}{2} \sum_{i=1}^d \log(1 - 2\lambda \mu_i) \le \sum_{i=1}^d 2\lambda \mu_i = 2\lambda \operatorname{tr}(\mathbf{H}).$$

where the second equality holds because $\mathbb{E}[\exp(zq_i^2)] = (1 - 2z)^{-1/2}$ for $z < 1/2$, and the inequality holds because $\log(1/(1 - z)) \le 2z$ for $z \in (0, 1/2]$. $\qquad\square$

