# OpenReview forum: "Dimension-Independent Convergence of Underdamped Langevin Monte Carlo in KL Divergence"
_ICML.cc/2026/Conference — ICML 2026 regular_

### Official Review · Reviewer_t8Mi · 2026-02-20

**Soundness:** 4
**Presentation:** 4
**Significance:** 2
**Originality:** 2
**Overall Recommendation:** 4
**Confidence:** 4

**Summary:**

The paper discusses non-log-concave sampling using underdamped Langevin and randomized underdamped algorithm under the $L$-smooth condition.

The authors provide a full discretization analysis in KL divergence and obtain state of the art non-asymptotic guarantees for sampling from a potential of the form $\pi=e^{-f}$. Under the additional assumption that the Hessian of $f$ is dominated by a matrix $H$ they are able to obtain bounds that depend on $tr(H)$ instead of $d$, refining the KL local error framework of  \citep{altschuler2025shifted} to a dimension-free setting.

**Compliance With Llm Reviewing Policy:**

Affirmed.

**Final Justification:**

This is a nice contribution, theoretically sound with good presentation.

The authors have answered my questions and addressed a lot of my concerns regarding the interesting examples, where their novelty is applicable.
 I will raise my score to 4 as I believe the article makes indeed an interesting contribution but I am not exactly sure if it is up to the standards of ICML.

**Key Questions For Authors:**

1. As I have already stated, my main concern is relevant to the significance on the results, but I am very open to change my mind and raise my score.
Could you present any relevant examples where $\nabla^2 f\leq H$ where $tr(H)<<Ld$ for high dimensions?

2. Could the results of your work also be applied if one uses the refined framework presented in : Zhang, Matthew S. "Analysis of Langevin midpoint methods using an anticipative Girsanov theorem." (see below)?

3. Could you state the main points where your analysis changes compared to the KL local framework with details?
I believe that would be a nice addition to an otherwise well-written article.


References: Zhang, Matthew S. "Analysis of Langevin midpoint methods using an anticipative Girsanov theorem." arXiv preprint arXiv:2507.12791 (2025).

**Limitations:**

Yes

**Strengths And Weaknesses:**

Soundness: The claims are well supported and the proof roadmap seems robust.

Presentation: The paper is well-written. The ideas are presented in a clear manner and the authors are very honest when presenting their original contributions and other results they use. Perhaps, it would be nice if more emphasis/details where given at the points in the proof roadmap that their contribution is novel.

Originality: The paper follows very closely the KL framework \citep{altschuler2025shifted}, with few original tweaks, in order to express some of the quanities with respect to the matrix $H$ that dominates the Hessian of $f$. Although the original contributions are not to be diminished the bulk of the proof is the same as in \citep{altschuler2025shifted}.

Significance: Although, this is a good mathematical paper, I am unconvinced about the results, which is the main critique of the paper.

Considering the assumptions and the techniques used are based on another paper, I believe that the significance of the results is crucial to the assessment of the paper.

I feel that the significance of the result (turning the $Ld$- dependent bound to $tr(H)$-dependent bound) is somewhat trivial for an ICML paper.

If believe that the dimension independent claim is somewhat misleading and I need to be convinced about such claim.

My doubts about the significance of the results coupled with the limited originality of the article is reflected on my assessment of the article.

---

> ### Author Rebuttal · Authors · 2026-03-31
>
> Many thanks for your thoughtful review and suggestions!
>
> ---
>
> **Q1.** Although the original contributions are not to be diminished, the bulk of the proof is the same as in [1].
>
> **A1.** We will add a dedicated subsection highlighting these modifications in the revision, as the reviewer suggests. Dimension-independent analysis that depends on the spectrum of $H$ is popular in the community. We summarize the technical contributions as follows:
>
> 1. **$H$-norm in local error (Lemma 4.1/5.1).** Under $\pi$, momentum $p$ is standard Gaussian, so $\mathbb E_\pi[\\|p\\|^2] = d$. As noted in Remark 6.2, using the Euclidean norm makes the final bound dimension-dependent. Replacing $\\|p\\|$ with $\\|p\\|\_H$ (with $\mathbb E_\pi[p^\top Hp]=tr(H)$) requires re-deriving local error bounds from scratch.
> 2. **Novel MGF bound via Stein's identity (Lemma 6.1/I.3).** The standard change-of-measure incurs $d$-dependence when directly bounding the MGF $\log\mathbb E_\pi[\exp(\\|\nabla V\\|^2/(4\beta))]$. We develop new techniques applying Taylor expansions to the MGF, and bound the moments $M_k=\mathbb{E}[\\|\nabla V\\|^{2k}]$ with recursive arguments $M_k\le[(2k-2)\beta+tr(H)]M_{k-1}$ using the Steins identity.
> 3. **Matrix-level tracking in cross-regularity (Lemma 4.2).** Standard bounds use $\mathbb E[\\|G\\|^2] = d$ for Gaussian $G$, producing $\beta d$ terms. We preserve $H$'s matrix structure throughout, avoiding any intermediate scalar relaxation $H\preceq \beta I$ that would reintroduce $d$.
>
> ---
>
> **Q2.** Examples with $tr(H)\ll\beta d$.
>
> **A2.** We provide two concrete examples:
>
> 1. When $V$ is ridge-separable, i.e., $V(x)=\frac1n\sum_i\sigma(a_i^\top x)$, where $\sigma''(z)\le L$ and $\||a_i\||\le R$. In this case, $\nabla^2V(x)=\frac1n\sum_i\sigma''(a_i^\top x)a_ia_i^\top\preceq\frac Ln\sum_ia_ia_i^\top:=H$, and $tr(H)=\frac Ln\sum_i\\|a_i\\|^2\le LR^2$. On the other hand, there are cases where $\beta d\gg tr(H)$. For example, if $a_i=\epsilon_i\mu$ with $\\|\mu\\|=R$ and $\epsilon_i\in\{\pm1\}$, then $H=L\mu\mu^\top$, and $\beta d = LR^2d$.
> 2. When the eigenvalues follow power-law decay, i.e., $H=diag(\{\lambda_i\})$ where $\lambda_i\simeq i^{-a}$, $tr(H)=\sum_i\lambda_i=O(1)$ if $a>1$, while $\beta d=\Theta(d)$. The power-law decay is verified in a number of works in the kernel regime (e.g., [3, 4]).
>
> ---
>
> **Q3.** Could the results of your work also be applied if one uses the anticipative Girsanov framework of [2]?
>
> **A3.**
> Zhang (2025) proposed a new approach for analyzing midpoint discretizations of stochastic differential equations using tools from Malliavin calculus, and derived estimates for the Radon–Nikodym derivative of the associated processes. Using this technique, the work improved regularity and cross-regularity conditions for various discretization methods. This line of work is parallel to ours: while [2] developed new analysis techniques for midpoint discretizations with improved discretization properties, our goal is to obtain **dimension-free** bounds for existing discretization methods within the KL local error framework.
>
> We believe that such techniques could be combined. However, there are technical challenges, as summarized below.
>
> First, Lemma 20 in [2] exhibits explicit $d$-dependence in the local error estimates, arising from the use of Euclidean norms (e.g., $\mathbb{E}_\pi[\\|p\\|^2]=d$). To achieve dimension-free bounds, it is essential to replace $\\|p\\|$ with $\\|p\\|_H$ (as in our Lemma 4.1). Without this refinement, the resulting bounds remain dimension-dependent. However, the use of double midpoint discretization further complicates the analysis. In particular, incorporating the $H$-norm requires re-deriving the local error estimates while carefully tracking the matrix $H$ throughout the argument.
>
> Moreover, the ambient dimension $d$ in Lemma 13 in [2] arises from bounding the Carleman–Fredholm determinant term. Obtaining a dimension-free analogue in this framework is challenging. The determinant term is defined at the level of path-space measures, and the dependence on $d$ comes from summing over the eigenvalues of $(E_1^\top D \hat{\lambda}_1 + E_2^\top D \hat{\lambda}_2)$ (see the proof of Lemma 13 in [2]). Due to this operator-level structure, replacing such bounds with a spectrum-dependent quantity (e.g., $\mathrm{tr}(H)$) would require preserving anisotropic information at the path-space level, which is technically highly nontrivial.
>
> We leave this as an interesting direction for future work.
>
> ---
>
> [1] Altschuler et al. Shifted composition iv: Underdamped langevin and numerical discretizations with partial acceleration
>
> [2] Zhang, Matthew S. "Analysis of Langevin midpoint methods using an anticipative Girsanov theorem."
>
> [3] Bietti & Mairal. On the Inductive Bias of Neural Tangent Kernels. 2019
>
> [4] Bordelon et al. Spectrum Dependent Learning Curves in Kernel Regression and Wide Neural Networks. 2020.

---

> > ### Author Rebuttal · Reviewer_t8Mi · 2026-04-01
> >
> > The authors have answered my questions and addressed a lot of my concerns. I will raise my score to 4 as  I believe the article  makes indeed an interesting contribution but I am not exactly sure if it is up to the standards of ICML.

---

> > > ### Author Response · Authors · 2026-04-02
> > >
> > > Thank you for your valuable feedback. We are glad that we have addressed your concerns, and will keep improving our manuscript.

---

### Official Review · Reviewer_Gf8e · 2026-03-07

**Soundness:** 2
**Presentation:** 4
**Significance:** 2
**Originality:** 2
**Overall Recommendation:** 3
**Confidence:** 3

**Summary:**

The paper develops KL convergence bounds for underdamped Langevin dynamics (ULD). Instead of the dimension, these bounds depend on the trace of an upper bound of the Hessian of the potential function.

**Compliance With Llm Reviewing Policy:**

Affirmed.

**Final Justification:**

In the rebuttal, the authors make it clear how their results improve existing ones and provide examples to illustrate that the case studied (small trace) is meaningful. Clearly, the paper does make a contribution, but it may look marginal without enough 'scope', which seems to be a consensus among reviewers. I hesitate to join the "weak accept" rating as there are quite some issues in the technical part of the paper (as listed in my review), which makes me a bit worried about the overall quality (whether there are more issues in the long appendix, whether the main result is correct), independent of the scope issue.

**Key Questions For Authors:**

They are listed in "Strengths And Weaknesses".

**Limitations:**

They are also listed in "Strengths And Weaknesses".

**Strengths And Weaknesses:**

(Presentation.) I think the authors do a great job in describing the background, the gap, and their contribution. I enjoy reading through the paper and get the main idea without much difficulty.

(Significance and Originality.) There is the term "dimensional-independent" in the title(s) of this paper (and related papers Freund et al. (2022), Liu et al. (2023)), but I feel that this result may be better described as an "anisotropic" generalization of Altschuler et al. (2025). When $\nabla^2 V\preceq \beta I$ is assumed, $\beta d$ appears in the bounds. If $\beta I$ is replaced by $H$, then $\beta d$ "should" be replaced by $\mathrm{tr}(H)$. In the simplest example $V(x)=||x||^2/2$, the convergence is still dimension-dependent as $\mathrm{tr}(H)=d$. More broadly, in the strongly convex case $\alpha I\preceq H\Rightarrow\mathrm{tr}(H)\geq\alpha d$ with $\alpha>0$, shall we still called the corresponding bounds dimensional-independent?

To justify the significance of $\mathrm{tr}(H)$-based bounds, the authors may consider describe in detail at least one popular setting of great importance where $\mathrm{tr}(H)\ll d$. The authors also need to distinguish their bounds from other $\mathrm{tr}(H)$-based ones. Compared to Freund et al. (2022), what new insight do we get when moving from OLD to ULD? Does the bound show that ULD is strictly better than OLD? Compared to Liu et al. (2023), what new insight do we get when moving from W2 to KL? "Notably, in the strongly log-concave setting, KL convergence is strictly stronger", but $\mathrm{tr}(H)\geq\alpha d$ with $\alpha>0$ in this case, which goes back to the dimension-dependent world.

(Soundness.) The statement of Lemma I.4 is problematic: $1\leq0$ when $\lambda\rightarrow0$. It seems that a "log" is missing in the statement, according to the last line of the proof, which is also problematic as $\nabla V$ becomes $V$.

The constant "2" and "4" in the proof of Lemma 6.1 are missing in the lemma statement.

In the discussion after Theorem 5.2, the authors may want to check and clarify the computation from KL (line 351) to W2 (line 357).

Why do "N-1 steps of RMD" appear in Theorem 4.3/4.4 while Section 4 is about ULMC?

In the proof of Lemma B.1, there is $H^{-1/2}$. Lemma B.1 is under Assumption 3.1, where $H$ may not be invertible, unless $\alpha>0$, but as previously mentioned, $\alpha>0$ makes $\mathrm{tr}(H)$ linear in $d$, which is the less interesting case for $\mathrm{tr}(H)$-based bounds.

---

> ### Author Rebuttal · Authors · 2026-03-31
>
> Many thanks for your thoughtful review! We will correct the typos and keep improving our manuscript.
>
> ---
>
> **Q1.** There is the term "dimensional-independent" in the title(s) of this paper (and related papers [1, 2]), but I feel that this result may be better described as an "anisotropic" generalization of Altschuler et al. (2025). When $\nabla^2V(x)\preceq\beta I$ is assumed, $\beta d$ appears in the bounds. If $\beta I$ is replaced by $H$, then $\beta d$ "should" be replaced by $tr(H)$. In the simplest example $V(x)=\\|x\\|_2^2/2$, the convergence is still dimension-dependent as $tr(H)=d$.
>
> **A1.** We thank the reviewer for the suggestion. We will emphasize in the revision that dimension-independence means the bound does not explicitly depend on the dimension, but instead on spectral quantities of the Hessian upper bound $H$ such as $\text{tr}(H)$.
>
> We agree that for certain canonical examples (e.g., $V(x)=\frac12\\|x\\|^2$ where $\text{tr}(H) = d$), the bound does not improve over the dimension-dependent one. However, as demonstrated in response **A3**, the improvement is substantial in practically relevant settings where the Hessian has low effective rank.
>
> ---
>
> **Q2.** In the strongly convex case $\nabla V(x)\succeq\alpha I$ with $\alpha>0$, shall we still called the corresponding bounds dimensional-independent?
>
> **A2.** We appreciate this thoughtful comment. We agree that if $V(x)$ is strongly convex, then $\mathrm{tr}(H) \ge \alpha d$. However, reducing our results to this “dimension-dependent” form yields a substantial improvement over prior $\beta d$-dependent bounds, by up to a factor of $\kappa = \beta/\alpha$, which is significant in the optimization and sampling literature. Moreover, we provide concrete and practical examples where $\mathrm{tr}(H)$ is significantly smaller than $d$, leading to genuinely dimension-free improvements (See **A3** ridge-separable).
>
> ---
>
> **Q3.** Settings with $tr(H)\ll\beta d$.
>
>
> **A3.** We provide two concrete examples:
>
> 1. When $V$ is ridge-separable, i.e., $V(x)=\frac1n\sum_i\sigma(a_i^\top x)$, where $\sigma''(z)\le L$ and $\||a_i\||\le R$. In this case, $\nabla^2V(x)=\frac1n\sum_i\sigma''(a_i^\top x)a_ia_i^\top\preceq\frac Ln\sum_ia_ia_i^\top:=H$, and $tr(H)=\frac Ln\sum_i\\|a_i\\|^2\le LR^2$. On the other hand, there are cases where $\beta d\gg tr(H)$. For example, if $a_i=\epsilon_i\mu$ with $\\|\mu\\|=R$ and $\epsilon_i\in\{\pm1\}$, then $H=L\mu\mu^\top$, and $\beta d = LR^2d$.
> 2. When the eigenvalues follow power-law decay, i.e., $H=diag(\{\lambda_i\})$ where $\lambda_i\simeq i^{-a}$, $tr(H)=\sum_i\lambda_i=O(1)$ if $a>1$, while $\beta d=\Theta(d)$. The power-law decay is verified in a number of works in the kernel regime (e.g., [3, 4]).
>
> ---
>
> **Q4.** The authors also need to distinguish their bounds from other $tr(H)$-based ones. Compared to [1], what new insight do we get when moving from OLD to ULD? Does the bound show that ULD is strictly better than OLD?
>
> **A4.** It is well known that discretization of ULD can have some acceleration effect over OLD. And our insight can be  stated as follows: We do not lose the acceleration of ULD over OLD **even when characterizing the dimension-independent KL divergence bound**. Our results are consistent with prior works [1, 2].
>
> ---
>
> **Q5.** Compared to [2], what new insight do we get when moving from W2 to KL? "Notably, in the strongly log-concave setting, KL convergence is strictly stronger", but within this case, which goes back to the dimension-dependent world.
>
> **A5.** In the **strongly-convex** setting, our results not only indicate the result in [2], but further improves the $\kappa$ dependence in the sample complexity by a factor of $\kappa^{1/3}$ (See discussion after Theorem 5.2 and **A7**).
>
> Furthermore, in general cases, KL divergence can bound the TV distance using Pinsker's inequality, while W2 distance cannot.
>
> ---
>
> **Q6.** In the proof of Lemma B.1, there is $H^{-1/2}$, which may not be invertible in the general-convex setting.
>
> **A6.** Thank you for pointing out this point that we miss. In the general convex setting, we can add a small regularization term and replace $H^{-1/2}$ with $(\lambda I+H)^{-1/2}$, which is always invertible. We can follow the original proof and take the limit $\lambda\to0$.
>
> ---
>
> **Q7.** Typo in line 357.
>
> **A7.** We appreciate the reviewer for the thoughtful suggestion. The sample complexity in W2 distance induced from Theorem 5.2 should be $\tilde O(\kappa^{4/3}\beta^{-2/3}[tr(H)]^{1/3}\epsilon^{-2/3})$.
>
> ---
>
> [1] Freund et al. When is the convergence time of Langevin algorithms dimension-independent? A composite optimization viewpoint. 2022.
>
> [2] Liu et al. Double randomized underdamped Langevin with dimension-independent convergence guarantee. 2023.
>
> [3] Bietti & Mairal. On the Inductive Bias of Neural Tangent Kernels. 2019
>
> [4] Bordelon et al. Spectrum Dependent Learning Curves in Kernel Regression and Wide Neural Networks. 2020.

---

> > ### Author Rebuttal · Reviewer_Gf8e · 2026-04-02
> >
> > Thank you for the clarification and examples. They help me have a clearer picture of the paper.

---

> > > ### Author Response · Authors · 2026-04-02
> > >
> > > Thank you for your valuable feedback.
> > >
> > > We are glad that our response was helpful, and would be grateful if you could reconsider your evaluation in light of this. Please feel free to let us know if there are any further questions or concerns.
> > >
> > > Thank you again for your time and consideration.

---

### Official Review · Reviewer_gvZZ · 2026-03-11

**Soundness:** 3
**Presentation:** 3
**Significance:** 3
**Originality:** 3
**Overall Recommendation:** 4
**Confidence:** 4

**Summary:**

This paper studies discretizations of underdamped Langevin dynamics and establishes dimension-free convergence rates in KL divergence. Its main claim is that the complexity of both standard ULMC and randomized midpoint discretization (RMD) depends on
 tr(H), where H is an upper bound on the Hessian, rather than on the ambient dimension. The dimension-free convergence results are established in both the strongly convex and general convex settings.

**Compliance With Llm Reviewing Policy:**

Affirmed.

**Key Questions For Authors:**

1. The authors may consider adding a discussion on how the analysis could be extended to the non-convex setting.

2. The authors may consider adding a discussion on how the analysis could be extended to the stochastic gradient setting.

**Limitations:**

See my questions for authors.

**Strengths And Weaknesses:**

Strength: The theoretical results are reasonably strong.

Weakness:  1. The analysis is restricted to the strongly convex and general convex settings, which somewhat limits the scope of the paper. In practice, many problems of interest are non-convex, but that setting is not covered here.

2, The convergence of the algorithm with stochastic gradients is not covered by the theory presented in the paper.

---

> ### Author Rebuttal · Authors · 2026-03-31
>
> Thank you very much for the recommendation!
>
> **Q1.** The analysis is restricted to the strongly convex and general convex settings, which somewhat limits the scope of the paper. In practice, many problems of interest are non-convex, but that setting is not covered here.
>
> **A1.** We thank the reviewer for this comment. The non-convex setting is indeed an important and interesting direction. However, when $V$ is non-convex, finding an approximate global minimizer can be intractable. Thus, a reasonable alternative objective is to study quantities such as the Fisher information.
>
> Further research directions include establishing asymptotic convergence under functional inequalities, such as the log-Sobolev inequality (LSI) or the Poincaré inequality (PI). This becomes more challenging in the dimension-independent setting. For example, [1] conduct non-convex analysis (Theorem 5.7 therein, under LSI) that yields bounds explicitly dependent on the ambient dimension $d$. To our knowledge, no prior work establishes dimension-free KL guarantees for ULD in non-convex settings. We leave this as an important direction for future work.
>
> ---
>
> **Q2.** The convergence of the algorithm with stochastic gradients is not covered by the theory presented in the paper.
>
> **A2.** We thank the reviewer for this suggestion. Extending the analysis to stochastic gradients (e.g., SGLD / SG-ULMC) is a practically motivated direction.
>
> We believe that stochastic gradient analysis can be incorporated into our framework. Note that even with stochastic gradients, the transition kernel remains Markovian, and thus the KL local error framework (Theorem 3.5) still applies. However, when analyzing the one-step strong and weak errors, additional terms of the form $\mathbb{E}\\|g(x,\zeta)-\nabla V(x)\\|^2$ arise. Controlling these terms necessitates suitable assumptions on the tail behavior of the stochastic gradient $g(x, \zeta)$. Verifying such assumptions goes beyond our main contribution of establishing sample complexity bounds that do not explicitly depend on the ambient dimension. Thus, we leave this as future work.
>
> ---
>
> [1] Altschuler et al. Shifted composition IV: toward ballistic acceleration for log-concave sampling. 2025.

---

> > ### Author Rebuttal · Reviewer_gvZZ · 2026-04-05
> >
> > The authors acknowledged the weakness, but did not address it.

---

> > > ### Author Response · Authors · 2026-04-06
> > >
> > > Many thanks for the message!
> > >
> > > We will include more discussions about the potential extensions of our results in the revision as suggested, iincluding both the nonconvex setting and the stochastic gradient setting. We have already outlined the key challenges in the nonconvex case in the original rebuttal, and will further clarify these points. TIn addition, we will provide a more detailed treatment of the stochastic gradient setting. In particular, we will discuss the error of ULMC and the randomized midpoint discretization with stochastic gradient $g(x, \zeta)$ (ULMC-SG and RM-SG) in detail. We focus on the weak and strong error bounds.
> > >
> > > The error vectors of ULMC are
> > > $$
> > > X_h^{ULMC-RM}-X_h=\int_0^h\frac{1-e^{-\gamma(h-t)}}{\gamma}[g(x, \zeta)-\nabla V(X_t)]dt=X_h^{ULMC}-X_h+\int_0^h\frac{1-e^{-\gamma(h-t)}}{\gamma}d t\cdot[g(x, \zeta)-\nabla V(x)],
> > > $$
> > > and
> > > $$
> > > P_h^{ULMC-RM}-P_h=P_h^{ULMC}-P_h+\int_0^he^{-\gamma(h-t)}[g(x, \zeta)-\nabla V(x)]dt.
> > > $$
> > > Since the stochastic gradients are unbiased, the weak error is the same as the case with exact gradient. For the strong error, note that $\zeta$ is independent of the Brownian motion, so
> > > $$
> > > \\|X_h^{ULMC}-X_h\\|\_{L_2}\simeq\\|X_h^{ULMC}-X_h\\|\_{L_2}+\int_0^h\frac{1-e^{-\gamma(h-t)}}{\gamma}dt\cdot\\|g(x, \xi)-\nabla V(x)\\|\_{L_2},
> > > $$
> > > where $\int_0^h\frac{1-e^{-\gamma(h-t)}}{\gamma}dt\le h^2$. It then suffices to bound $\\|g(x, \xi)-\nabla V(x)\\|\_{L_2}$, which requires additional assumption on the stochastic gradient. Analysis for the randomized midpoint discretization is similar, where the stochastic gradient incurs the same additive error $\\|g(x, \xi)-\nabla V(x)\\|\_{L_2}$.

---

### Official Review · Reviewer_x1sQ · 2026-03-13

**Soundness:** 3
**Presentation:** 3
**Significance:** 2
**Originality:** 1
**Overall Recommendation:** 4
**Confidence:** 4

**Summary:**

This paper studies the complexity of sampling from high-dimensional distributions using discretizations of the Langevin SDE. The main contribution is to give a bound for strongly-log-concave and another weaker bound for log-concave measures, whose iteration complexity does not explicitly depend on dimension $d$ but rather on the trace of a global upper bound $H$ on the Hessian (hence giving an improvement when the measure exhibits e.g. low dimensional structure). The proof of this result proceeds through using the local error framework of Altschuler, Chewi, and Zhang, and the main technical contribution is showing that the local errors in question do not necessarily depend on $d$ explicitly but can be upper bounded in terms of $tr(H)$.

**Compliance With Llm Reviewing Policy:**

Affirmed.

**Final Justification:**

The rebuttal reinforced my prior assessment and I think the paper makes a contribution in sampling and is at the bar for acceptance.

**Key Questions For Authors:**

None

**Limitations:**

yes

**Strengths And Weaknesses:**

The paper provides an improvement in the field of log-concave sampling. The bound proven indeed is the best in the literature, see Table 1.   It is a nice to see further concrete applications of the local error framework. The paper also clearly presents the proofs and their ideas, including in the appendix.

The main weakness of the paper is that the contribution feels rather marginal. I read several of the proofs in the appendix, while the result is an improvement on the best known bounds, the proofs follow somewhat straightforwardly from the local error framework (which is from prior literature in the field). The main idea is to perform direct calculations showing these local errors that arise from the framework can be bounded in terms of $tr(H)$ and not $d$ explicitly (e.g. Lemmas B.1, B.2).

Thus the result makes improvements but feels like a low-hanging fruit that could be obtained rather directly from prior work.
It would be far more compelling if there were more conceptual advances that led to genuine improvements on the local error framework when $tr(H)$ is small, however this seems not to be the case. Note in Table 1 if one replaces $tr(H)$ by the worst case bound $\beta d$ one recovers exactly the bounds of Altschuler et al 2025 (also displayed in that table).

Similarly, beyond the techniques and more low-hanging nature of the result, it is also unclear to me how substantial this result actually is, are there natural examples in log-concave sampling (eg measures arising in Bayesian inference) for which $tr(H)$ is small? Of course one could find examples, but I don't know how motivated they are. For example the posterior in linear regression with a isotropic Gaussian prior, does not have small $tr(H)$. Indeed I feel the line of work on finding upper bounds that depend on $tr(H)$ rather than explicitly on $d$ is not part of the core line of work in log-concave sampling.

Another issue I have is the phrasing "dimension-free" which is used consistently and emphasized throughout in the writing. The bound doesn't explicitly depend on dimension, but the phrasing seems to suggest at times, that the bounds provided are dimension free for many canonical examples (which it isn't). I would instead adopt the phrasing "does not explicitly depend on dimension" for more of the paper.

Overall though I think the paper does make a contribution and is at the bar for acceptance.

---

> ### Author Rebuttal · Authors · 2026-03-31
>
> Thank you very much for the positive feedback!
>
> **Q1.** Contribution feels marginal. Proofs follow straightforwardly from [1]. If one replaces $tr(H)$ by the worst case bound $\beta d$, one recovers bound of [1].
>
> **A1.** We would argue that the strong connection of the results in our paper and [1] does not indicate the lack of technical novelty; instead, the reduction from our results to [1] by replacing $tr(H)$ is possible only through the sharp analysis we developed. We summarize the technical challenges as follows:
>
> 1. **$H$-norm in local error (Lemma 4.1/5.1).** Under $\pi$, momentum $p$ is standard Gaussian, so $\mathbb E_\pi[\\|p\\|^2] = d$. As noted in Remark 6.2, using the Euclidean norm makes the final bound dimension-dependent. Replacing $\|p\|$ with $\\|p\\|\_H$ (with $\mathbb E_\pi[p^\top Hp]=tr(H)$) requires re-deriving local error bounds from scratch.
> 2. **Novel MGF bound via Stein's identity (Lemma 6.1/I.3).** The standard change-of-measure incurs $d$-dependence when directly bounding the MGF $\log\mathbb E_\pi[\exp(\\|\nabla V\\|^2/(4\beta))]$. We develop new techniques applying Taylor expansions to the MGF, and bound the moments $M_k=\mathbb{E}[\\|\nabla V\\|^{2k}]$ with recursive arguments $M_k\le[(2k-2)\beta+tr(H)]M_{k-1}$ using the Steins identity.
> 3. **Matrix-level tracking in cross-regularity (Lemma 4.2).** Standard bounds use $\mathbb E[\\|G\\|^2] = d$ for Gaussian $G$, producing $\beta d$ terms. We preserve $H$'s matrix structure throughout, avoiding any intermediate scalar relaxation $H\preceq \beta I$ that would reintroduce $d$.
>
> ---
>
> **Q2.** Examples with small $tr(H)$.
>
> **A2.** We will add discussions of examples where dimension-free bounds lead to substantial improvement in the revision.
>
> Examples that are completely dimension-free:
>
> 1. When $V$ is ridge-separable, i.e., $V(x)=\frac1n\sum_i\sigma(a_i^\top x)$, where $\sigma''(z)\le L$ and $\||a_i\||\le R$. In this case, $\nabla^2V(x)=\frac1n\sum_i\sigma''(a_i^\top x)a_ia_i^\top\preceq\frac Ln\sum_ia_ia_i^\top:=H$, and $tr(H)=\frac Ln\sum_i\\|a_i\\|^2\le LR^2$. On the other hand, there are cases where $\beta d\gg tr(H)$. For example, if $a_i=\epsilon_i\mu$ with $\\|\mu\\|=R$ and $\epsilon_i\in\{\pm1\}$, then $H=L\mu\mu^\top$, and $\beta d = LR^2d$.
> 2. When the eigenvalues follow power-law decay, i.e., $H=diag(\{\lambda_i\})$ where $\lambda_i\simeq i^{-a}$, $tr(H)=\sum_i\lambda_i=O(1)$ if $a>1$, while $\beta d=\Theta(d)$. The power-law decay is verified in a number of works in the kernel regime (e.g., [2, 3]).
>
> **Linear regression with Gaussian prior: substantial improvement.** Let $X=(x_1,x_2,\ldots,x_n)\in \mathbb{R}^{d\times n}$, $y\in\mathbb{R}^n$ and $w\in\mathbb{R}^d$ with $y= X^\top w_\ast+\epsilon$, where $\epsilon \sim N(0,\sigma^2I)$, and $\\|x_i\\|\le R$. Suppose the prior of $w$ is $\mathcal{N}(0, \tau^2I)$, then $V(w)=\frac{\\|w\\|^2}{2\tau^2}+\frac{\\|y-X^\top w\\|^2}{2\sigma^2}$, and $H=\frac{I}{\tau^2}+\frac{XX^\top}{\sigma^2}$. Similar to the ridge-separable example, we can construct examples such that $\beta d=(\tau^{-2}+nR^2\sigma^{-2})d$. In contrast, $tr(H)=d\tau^{-2}+nR^2\sigma^{-2}$. Despite not completely dimension-free, $tr(H)$ saves a factor of $d$ in the term with respect to $n$, which is the dominant term with high signal-to-noise ratio $R/\sigma$. The bound of $tr(H)$ can be further improved with anisotropic prior, but $\beta$ does not.
>
> ---
>
> **Q3.** "Dimension-independence" suggests the bounds provided are dimension free for many canonical examples (which it isn't). I would suggest the phrasing "does not explicitly depend on dimension".
>
> **A3.**
> We thank the reviewer for the suggestion. We will emphasize that dimension-independence means the bound does not explicitly depend on the dimension in the revision. However, we would like to clarify that the term "dimension-free/independent" is used in the standard sense, adopted throughout this line of research (e.g., [4, 5]): The bound does not explicitly depend on the ambient dimension $d$, but instead on spectral quantities of the Hessian upper bound $H$ such as $\text{tr}(H)$.
>
> We agree that for certain canonical examples (e.g., $V(x) = \frac{1}{2}\\|x\\|^2$ where $\text{tr}(H) = d$), the bound does not improve over the dimension-dependent one. However, as demonstrated in response **A2**, the improvement is substantial in practically relevant settings where the Hessian has low effective rank.
>
> ---
>
> [1] Altschuler et al. Shifted composition IV: toward ballistic acceleration for log-concave sampling. 2025.
>
> [2] Bietti & Mairal. On the Inductive Bias of Neural Tangent Kernels. 2019
>
> [3] Bordelon et al. Spectrum Dependent Learning Curves in Kernel Regression and Wide Neural Networks. 2020.
>
> [4] Freund et al. When is the convergence time of Langevin algorithms dimension-independent? A composite optimization viewpoint. 2022.
>
> [5] Liu et al. Double randomized underdamped Langevin with dimension-independent convergence guarantee. 2023.

---

> > ### Author Rebuttal · Reviewer_x1sQ · 2026-04-01
> >
> > Thanks to the authors for the detailed response. My questions were addressed. In light of the response my original opinion of the paper is not changed. I understand that there are new technical steps to obtain the result that formally depend on $tr(H)$ rather than $\beta d$, and that this obtains improvement in some examples where $tr(H)$ is small, but frankly I don't think these steps are particularly surprising or yield major new insights into the field of sampling. I also still would strongly prefer the more accurate phrasing "does not explicitly depend on dimension".
> >
> > Despite these flaws, the paper does make a contribution and I think it is at the bar for ICML. As such I will keep my current (positive) score.

---

> > > ### Author Response · Authors · 2026-04-02
> > >
> > > We are glad that our response has addressed your questions.
> > >
> > > We would like to emphasize that the improvement from $\beta d$ to $\mathrm{tr}(H)$, without modifying the ULMC algorithm, is not conceptually trivial, especially given the presence of isotropic noise in the dynamics. Indeed, such noise can potentially induce dimension-dependent effects, which might appear unavoidable under a naive analysis, but our results show that this is not the case.
> > >
> > > In prior works that achieve similar bounds, modifications to the algorithm are typically required. For example, [5] mitigates this dependence by introducing a doubly randomized technique. In contrast, our result shows that such dimension dependence can be improved purely at the analysis level, **without altering the underlying algorithm**.
> > >
> > > We will also adopt more precise phrasing in the revision, as suggested.

---

### Decision · Program_Chairs · 2026-04-30

**Decision:**

Accept (regular)

**Comment:**

In this paper, the authors present an analysis to establish dimension-independent bounds of the Kullback-Leibler divergence for discretized underdamped Langevin diffusion. The manuscript is mostly clearly written, with detailed explanations and strong theoretical foundations.

The reviewers raised several concerns, in particular about the contribution which could seem marginal relative to the existing literature in the field. The proposed results rely on strong assumptions in a general convex setting  and the assumptions and techniques rely on recent works. They questioned the significance and novelty of the results.

During the rebuttal, the authors acknowledged the connections with previous works but also clarified the technical novelties and the position of the paper about the dimension-free setting. They also included discussions on the assumptions and results with  examples where dimension-free bounds are explicit. They outlined research perspectives based on functional inequalities (Log-Sobolev and Poincaré inequalities), although they do not claim that their current techniques directly generalize to these settings.

I believe that this work provide a strong theoretical foundation and they mainly answered to the concerns raised by reviewers. A complete extensions to the stochastic gradient and non-convex settings would require substantial additional work but I encourage the authors to clarify precisely how they work could be adapted in such settings to improve the scope of this paper and makes it a contribution for a larger Machine Learning audience.